# TRADING QUALITY FOR EFFICIENCY OF GRAPH PARTITIONING: AN INDUCTIVE METHOD ACROSS GRAPHS

## ABSTRACT

Many applications of network systems can be formulated as several NP-hard combinatorial optimization problems regarding graph partitioning (GP), e.g., modularity maximization and NCut minimization. Due to the NP-hardness, to balance the quality and efficiency of GP remains a challenge. Existing methods use advanced machine learning techniques to obtain high-quality solutions but usually have high complexity. Some fast GP methods use heuristic strategies to ensure low runtime but suffer from quality degradation. In contrast to conventional *transductive* GP methods applied to a single graph, we propose an *inductive* graph partitioning (IGP) framework across multiple associated graphs of a system or scenario to alleviate the NP-hard challenge. IGP first conducts the *offline* training of a dual graph neural network on historical graph snapshots to capture properties of the system. The trained model is then generalized to newly generated graphs for fast high-quality *online* GP without additional optimization, where a better trade-off between quality and efficiency is achieved. IGP is also a generic framework that can capture the permutation invariant GP ground-truth of historical snapshots in the *offline* training and tackle the *online* GP on graphs with non-fixed number of nodes and clusters. Experiments on a set of benchmarks demonstrate that IGP achieves competitive quality and efficiency over various state-of-the-art baselines.

## 1 INTRODUCTION

For various complex systems, e.g., communication networks, graph is a generic model to describe the entities and their relations using a set of nodes and edges. Graph partitioning (GP) is a classic inference task that aims to partition the nodes of a graph into several groups (i.e., clusters) with dense linkage distinct from other groups. Since the extracted clusters are believed to correspond to several real-world substructures of a system, e.g., cells in wireless networks (Dai & Bai, 2017), many network applications are formulated as GP tasks (Qin et al., 2019; Patil & Kulkarni, 2021).

Mathematically, GP can be described as some NP-hard combinatorial optimization problems, e.g., modularity maximization (Newman, 2006) and normalized cut (NCut) minimization (Von Luxburg, 2007). Due to the NP-hardness, to balance the quality and efficiency of GP remains a challenge but some real applications have both requirements of high quality and low runtime, e.g., accurate GP on a wireless cellular network with several thousand nodes in a few seconds (Dai & Bai, 2017).

On the one hand, prior work has demonstrated the ability of machine learning (ML) techniques to achieve high GP quality. Typical ML-based methods include non-negative matrix factorization (Wang et al., 2011) and generative probabilistic models (Karrer & Newman, 2011). Graph embedding emerges as a promising technique for GP in recent studies. These methods first learn low-dimensional node representations via random walk (Perozzi et al., 2014) or deep learning (Wang et al., 2016; Yang et al., 2016) to capture high-order proximities and nonlinear characteristics of a graph. The GP result is derived by feeding the learned embedding to a downstream clustering module (e.g, $K$Means). However, most ML-based methods have to use iterative optimization algorithms (e.g., gradient descent) to learn large-scale parameters for each graph with high complexity.

On the other hand, to reduce the overall runtime and satisfy real-time constraints of some applications is another major focus. Some fast GP methods apply heuristic strategies to approximate conventional objectives, e.g., greedy modularity maximization (Blondel et al., 2008) and multi-level coarsening for graph-cut minimization (Dhillon et al., 2007). Several graph embedding approaches

also adopt fast approximation to reduce the inference time, e.g., random projection for high-order proximities (Zhang et al., 2018a). Despite their high efficiency (e.g., low runtime), they may suffer from unexpected quality decline due to the inherent information loss of heuristic approximation.

The aforementioned state-of-the-art (SOTA) GP methods are inherently *transductive*, which independently optimize the model on each static graph and can only tackle GP on such a unique graph. We try to **achieve a better trade-off between quality and efficiency from a new *inductive* perspective across multiple associated graphs of a system or scenario**. An inductive graph partitioning (IGP) framework is proposed based on the fact that *most real-world complex systems generate a set of graphs via common knowledge in terms of several underlying distributions*, e.g., power-law distributions. This hypothesis is also adopted in the simulation of various network systems (Wehrle et al., 2010). The multiple graphs can be (i) associated snapshots evolving over time or (ii) independent (sub)graphs of a scenario without temporal dependency e.g., ego-nets in social media. IGP first trains a high-quality GP model on historical snapshots in an *offline* way, which aims to fully capture properties of the system or scenario regardless of time cost. The trained model is then generalized to newly generated graphs for *online* GP without additional optimization, which significantly reduces the runtime on these new graphs and is believed to have high-quality GP results. For instance, we can conduct the *offline* training of IGP on existing known ego-nets of a social network and generalize it to new unseen ego-nets. In real applications, it is usually assumed that we have enough time to prepare a high-quality model using historical known data in an *offline* way, which is also one-time-effort only. Our focus is to achieve a better trade-off between quality and efficiency of *online* GP on new graphs after deploying the trained model to a system with model parameters fixed.

Technically, IGP follows a graph embedding scheme for unsupervised node-level tasks (i.e., GP) across graphs using the *inductive* nature of graph neural networks (GNN). Although some inductive GNNs have the potential to be generalized to new unseen nodes and graphs (Hamilton et al., 2017; Veličković et al., 2018), they still suffer from the following limitations for GP.

In this study, we consider GP on multiple graphs where topology is the only available information source, i.e., without graph attributes. As some systems allow the addition and deletion of entities, we assume that the number of nodes $N$ is non-fixed for multiple graphs. Although existing inductive GNNs can be applied to graphs with non-fixed $N$, they were originally designed for attributed graphs and rely on fixed dimensionality of node features. Our preliminary experiments indicate that some classic settings of GNN for the case without node features (e.g., using a constant matrix with fixed dimensionality as the feature input) may suffer from poor GP quality. Some methods extract features via dimensionality reduction (e.g., PCA (Nazi et al., 2019)) to map the topology (e.g., adjacency matrix) with non-fixed $N$ to a fixed-dimensional feature space, which is still time-consuming.

Moreover, different system snapshots can be assigned with different number of clusters $K$. Some inductive GNN based approaches adopt an end-to-end (E2E) framework to achieve a better approximation to classic GP objectives (e.g., GAP (Nazi et al., 2019) for NCut minimization and ClusNet (Wilder et al., 2019) for modularity maximization), with partitioning results derived via a fully-connected (FC) output layer. Despite their high quality, they can only tackle the *inductive* GP across graphs with fixed $K$, due to fixed dimensionality of the output layer. These E2E methods still need to be trained from scratch for new graphs with non-fixed $K$, which is time-consuming.

Although prior studies have demonstrated the ability of inductive GNNs to tackle (semi-)supervised tasks (e.g., node classification) on new unseen nodes and graphs (Hamilton et al., 2017; Veličković et al., 2018), few of them consider unsupervised node-level tasks (e.g., GP) across graphs. Our experiments also indicate that some standard settings of inductive GNNs for unsupervised tasks (e.g., using classic unsupervised training loss of graph embedding for GNN (Hamilton et al., 2017)) may still lack robustness for the *online* GP on newly generated graphs.

In addition to obtaining a better trade-off between quality and efficiency over conventional *transductive* GP methods via a novel ***inductive* framework across graphs**, we also make the following contributions to address the aforementioned limitations of *inductive* GNNs. **(i)** To tackle the *online* GP with non-fixed $N$, we develop an **efficient feature extraction module for *inductive* GNNs** via graph coarsening. **(ii)** In contrast to E2E methods, IGP adopts an ***inductive* graph embedding scheme across graphs** which can tackle the *online* GP with non-fixed $K$. **(iii)** Note that GP is an unsupervised node-level task, where the cluster labels are permutation invariant, e.g., label assignments $(l_1, l_2, l_3) = (1, 2, 2)$ and $(l_1, l_2, l_3) = (2, 1, 1)$ are the same in terms of GP with $l_i$ as the

cluster label of node $v_i$. IGP can further incorporate such permuttion invariant label information (e.g., GP ground-truth) of historical graphs to the *offline* training by **combining several GP objectives** (e.g., modularity maximization and NCut minimization) **with a novel dual GNN structure** which ensures strong robustness for the *online* GP on new graphs.

## 2 PROBLEM STATEMENTS AND PRELIMINARIES

In this study, we consider GP on a set of graphs $S = \{\mathcal{G}_1, \mathcal{G}_2, \cdots, \mathcal{G}_T\}$ from a common system or scenario. Each graph $\mathcal{G}_t \in S$ can be represented as $\mathcal{G}_t = (\mathcal{V}_t, \mathcal{E}_t)$ with $\mathcal{V}_t = \{v_1^t, \ldots, v_{N_t}^t\}$ and $\mathcal{E}_t = \{(v_i^t, v_j^t) \big| v_i^t, v_j^t \in \mathcal{V}_t\}$ as the sets of nodes and edges. For each $\mathcal{G}_t$, topology is the only available information source without graph attributes. We use an adjacency matrix $\mathbf{A}_t \in \Re^{N_t \times N_t}$ to describe its topology with $N_t$ nodes, where $(\mathbf{A}_t)_{ij} = (\mathbf{A}_t)_{ji} = 1$ if $(v_i^t, v_j^t) \in \mathcal{E}_t$ and $(\mathbf{A}_t)_{ij} = (\mathbf{A}_t)_{ji} = 0$ otherwise. Since some systems allow addition and deletion of entities, we assume that different snapshots in $S$ can have different node sets, i.e., $\exists \mathcal{G}_t, \mathcal{G}_s \in S$ s.t. $\mathcal{V}_t \neq \mathcal{V}_s$. Since there may be no temporal dependency among $\{\mathcal{G}_1, \cdots, \mathcal{G}_T\}$ in some cases (e.g., multiple ego-nets in social media), we do not consider the node correspondence among $\{\mathcal{V}_1, \cdots, \mathcal{V}_T\}$. We follow the hypothesis adopted in the simulation of various network systems (Wehrle et al., 2010) that *graphs in $S$ are independently generated via several underlying distributions of a common system or scenario.*

**Graph Partitioning.** Given a graph $\mathcal{G}_t$, GP aims to partition the node set $\mathcal{V}_t$ into $K_t$ subsets (i.e., clusters) $C_t = \{C_1^t, \cdots, C_{K_t}^t\}$ so that (i) within each cluster the linkage is dense but (ii) between different clusters the linkage is relatively loose. The result $C_t$ also satisfies $C_r^t \cap C_s^t = \emptyset$ for $\forall r \neq s$. Mathematically, GP can be formulated as several NP-hard combinatorial optimization problems, e.g., NCut minimization (Von Luxburg, 2007) and modularity maximization (Newman, 2006).

Given a graph $\mathcal{G}_t$, NCut minimization aims to get the GP result $C_t$ that minimizes the NCut metric:

$$\arg\min_{C_t} \text{NCut}(C_t) = \frac{1}{2} \sum_{r=1}^{K_t} [\text{cut}(C_r^t, \bar{C}_r^t)/\text{vol}(C_r^t)] \tag{1}$$

where $\bar{C}_r^t = \mathcal{V}_t - C_r^t$ is the complementary set of $C_r^t$; $\text{cut}(C_r^t, \bar{C}_r^t) = \sum_{v_i \in C_r^t, v_j \in \bar{C}_r^t} (\mathbf{A}_t)_{ij}$ is the cut between $C_r^t$ and $\bar{C}_r^t$; $\text{vol}(C_r^t) = \sum_{v_i, v_j \in C_r^t} (\mathbf{A}_t)_{ij}$ is the volume of $C_r^t$. The objective (1) can be equivalently expressed in the following matrix form:

$$\arg\min_{\mathbf{H}_t} \text{tr}(\mathbf{H}_t^T \mathbf{L}_t \mathbf{H}_t) \text{ s.t. } \mathbf{H}_t^T \mathbf{H}_t = \mathbf{I}_{K_t}, \tag{2}$$

where $\mathbf{L}_t = \mathbf{I}_{N_t} - \mathbf{D}_t^{-0.5} \mathbf{A}_t \mathbf{D}_t^{-0.5}$ is the Laplacian matrix of $\mathbf{A}_t$; $\mathbf{D}_t = \text{diag}(d_1^t, \cdots, d_{N_t}^t)$ is a diagonal matrix with $d_i^t = \sum_j (\mathbf{A}_t)_{ij}$; $\mathbf{I}_N$ is an $N$-dimensional identity matrix. $\mathbf{H}_t \in \Re^{N_t \times K_t}$ is the membership indicator, where $(\mathbf{H}_t)_{ir} = [d_i^t \cdot \text{vol}(C_r^t)^{-1}]^{0.5}$ if $v_i^t \in C_r^t$ and $(\mathbf{H}_t)_{ir} = 0$ otherwise.

Modularity maximization is another conventional NP-hard objective of GP. Given a graph $\mathcal{G}_t$, it aims to find a partition $C_t$ that maximizes the modularity metric:

$$\arg\min_{C_t} \text{Mod}(C_t) = \frac{1}{2e} \sum_r \sum_{v_i^t, v_j^t \in C_r^t} [(\mathbf{A}_t)_{ij} - d_i^t d_j^t/(2e)], \tag{3}$$

where $e = \sum_i d_i^t/2$ is the number of edges. The objective can also be rewritten in a matrix form:

$$\arg\min_{\mathbf{H}_t} -\text{tr}(\mathbf{H}_t^T \mathbf{Q}_t \mathbf{H}_t) \text{ s.t. } \text{tr}(\mathbf{H}_t^T \mathbf{H}_t) = N_t, \tag{4}$$

where $\mathbf{Q}_t \in \Re^{N_t \times N_t}$ is the modularity matrix with $(\mathbf{Q}_t)_{ij} = (\mathbf{A}_t)_{ij} - d_i^t d_j^t/(2e)$. $\mathbf{H}_t \in \Re^{N_t \times K_t}$ is the membership indicator, where $(\mathbf{H}_t)_{ir} = 1$ if $v_i^t \in C_r^t$ and $(\mathbf{H}_t)_{ir} = 0$ otherwise.

**Inductive Graph Embedding.** Given a single graph $\mathcal{G}_t$, conventional *transductive* graph embedding aims to learn a function $f : \{v_i^t\} \mapsto \{\mathbf{u}_i^t \in \Re^{1 \times k}\}$ that maps each node $v_i^t$ to a $k$-dimensional vector $\mathbf{u}_i^t$. The key characteristics of $\mathcal{G}_t$ (e.g., clustering structure) should be preserved in the embedding space, where a node pair $(v_i^t, v_j^t)$ with similar properties (e.g., in the same cluster) should have similar representations $(\mathbf{u}_i^t, \mathbf{u}_j^t)$ (e.g., with close distance in the embedding space). The learned representations $\{\mathbf{u}_i^t\}$ are used as the input of several downstream tasks on $\mathcal{G}_t$ including GP.

To alleviate the NP-hard challenge of GP, we consider a novel *inductive* graph embedding scheme, where model parameters of $f$ are shared by all the graphs in $S$. We divide $S$ into a training set

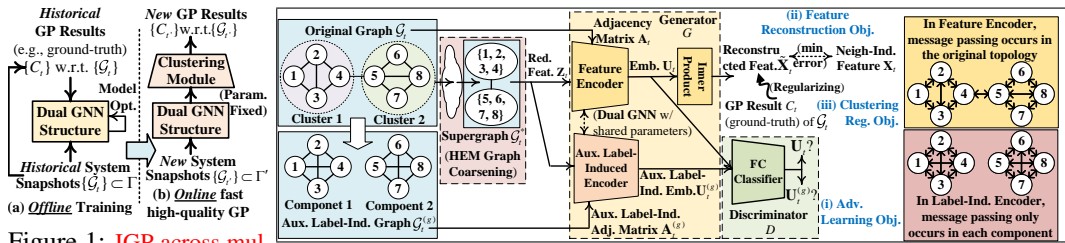

Figure 1: IGP across multiple associated graphs.

Figure 2: A running example of the *offline* training of our dual GNN structure.

$\Gamma \subset S$ and a test set $\Gamma' = S - \Gamma$, which represent the sets of historical known and newly generated graphs. We first train $f$ on $\Gamma$ in an *offline* way to fully capture structural properties of historical graphs. Although graph embedding and GP are unsupervised procedures, we assume the *permutation invariant* GP result $C_t$ (i.e., label information) of each historical graph $\mathcal{G}_t \in \Gamma$ is available in the *offline* training, which can be used to enhance the *offline* embedding optimization. In general, the training label information $\{C_t | \mathcal{G}_t \in \Gamma\}$ can be from the (i) ground-truth of system applications (e.g., cells in wireless cellular network decomposition) or (ii) results of some high-quality (but usually time-consuming) GP baselines. It indicates another application that we can use GP results of some strong baselines to regularize the *offline* training of IGP. After the training, we directly obtain the embedding $\{\mathbf{u}_i^{t'}\}$ from $f$ for each new graph $\mathcal{G}_{t'} \in \Gamma'$ with model parameters fixed. The *online* GP on a new graph $\mathcal{G}_{t'}$ is the downstream task, where we assume the number of clusters $K_{t'}$ is given. The GP result of $\mathcal{G}_{t'}$ is derived by applying a clustering algorithm (e.g., $K$Means) to $\{\mathbf{u}_i^{t'}\}$.

## 3 METHODOLOGY

We propose an IGP framework with the overview shown in Fig. 1, including the (i) *offline* training of a dual GNN structure on historical graphs and (ii) *online* generalization to new graphs via an *inductive* graph embedding scheme. Fig. 2 gives a running example of the *offline* training in IGP.

**Model Architecture.** Inspired by the adversarial auto-encoder (AAE) (Makhzani et al., 2015), we introduce a novel dual GNN structure, including a *generator G* and a *discriminator D* that are jointly optimized via an adversarial process. To the best of our knowledge, **the dual GNN structure is our original design, which is different from SOTA AAE-based graph embedding methods** (Pan et al., 2018) that regularize the embedding via the adversarial process between a probability distribution. In contrast, our dual GNN structure incorporates the permutation invariant label information of historical graphs $\{\mathcal{G}_t \in \Gamma\}$ to the learned embedding via (i) a parameter sharing strategy and (ii) an adversarial process between $\{\mathcal{G}_t\}$ and *auxiliary label-induced graphs* $\{\mathcal{G}_t^{(g)}\}$.

Concretely, in the *offline* training, we assume the topology and GP result (used as the 'ground-truth') of each historical graph $\mathcal{G}_t$ are available, which are described by an adjacency matrix $\mathbf{A}_t \in \Re^{N_t \times N_t}$ and an *indicator matrix* $\mathbf{R}_t \in \Re^{N_t \times K_t}$. We define $(\mathbf{R}_t)_{ir} = 1$ if node $v_i^t$ is in cluster $C_r^t$ by the 'ground-truth' and $(\mathbf{R}_t)_{ir} = 0$ otherwise. Due to the disjoint constraint of GP, only one entry in each row of $\mathbf{R}_t$ is 1 with other entries in the same row set to 0. In addition to the original graph $\mathcal{G}_t$, an *auxiliary label-induced graph* $\mathcal{G}_t^{(g)}$, with $\mathbf{A}_t^{(g)} = \mathbf{R}_t \mathbf{R}_t^T$ as the adjacency matrix describing its topology, is introduced to encode the structure of 'ground-truth' based on the following **Fact 1**.

**(Fact 1)** For a historical graph $\mathcal{G}_t$ with $K_t$ clusters, its *auxiliary label-induced graph* $\mathcal{G}_t^{(g)}$ has $K_t$ fully connected components with each component corresponding to one unique cluster of $\mathcal{G}_t$. Namely, there is an edge between nodes $v_i^t$ and $v_j^t$ in $\mathcal{G}_t^{(g)}$ only when they are in the same cluster, thus encoding the label information of 'ground-truth' (see Appendix A for the proof).

In the example of Fig. 2 with 8 nodes $\{1, 2, \cdots, 8\}$, $\mathcal{G}_t^{(g)}$ has 2 fully connected components $\{1, 2, 3, 4\}$ and $\{5, 6, 7, 8\}$ w.r.t. the 2 clusters in the GP result (i.e., 'ground-truth') of $\mathcal{G}_t$.

The generator $G$ includes a *feature encoder* and a *label-induced encoder*, forming a *dual GNN structure* with shared parameters $\delta_G$. The *feature encoder* takes $\mathbf{A}_t$ and a feature matrix $\mathbf{X}_t$ as inputs, and derives graph embedding $\mathbf{U}_t$, i.e., $\mathbf{U}_t = G(\mathbf{A}_t, \mathbf{X}_t; \delta_G)$. The *label-induced encoder* takes $\mathbf{A}_t^{(g)}$ and $\mathbf{X}_t$ as inputs, and outputs *auxiliary embedding* $\mathbf{U}_t^{(g)}$, i.e., $\mathbf{U}_t^{(g)} = G(\mathbf{A}_t^{(g)}, \mathbf{X}_t; \delta_G)$.

GNN is originally designed for attributed graphs, where each node $v_i^t$ must have a feature vector described by the $i$-th row of a feature matrix $\mathbf{X}_t$. In this study, we consider GP without node attributes. Instead of using conventional settings of GNN for graphs without attributes (e.g., use a constant matrix for the input features), we extract additional structural feature $\mathbf{X}_t$ based on the neighbor similarity encoded in $\mathbf{A}_t$. IGP captures the permutation invariant label information of historical graphs by combining the dual GNN structure with classic GP objectives. We adopt the objectives of (i) modularity maximization and (ii) NCut minimization as two examples, corresponding to two settings of $\mathbf{X}_t$. For modularity maximization in (4), the modularity matrix $\mathbf{Q}_t$ encodes the reweighted neighbor similarity based on $\mathbf{A}_t$, so we let $\mathbf{X}_t = \mathbf{Q}_t$. For NCut minimization in (2), the Laplacian matrix $\mathbf{L}_t$ gives the primary characteristics of graph structure, where $\mathbf{M}_t = \mathbf{D}_t^{-0.5}\mathbf{A}_t\mathbf{D}_t^{-0.5}$ is the key component regarding neighbor similarity. Hence, we let $\mathbf{X}_t = \mathbf{M}_t$. Both $\mathbf{Q}_t$ and $\mathbf{M}_t$ are the reweighting of $\mathbf{A}_t$, where the pair of nodes $(v_i^t, v_j^t)$ with similar neighbor-induced features $((\mathbf{X}_t)_{i,:}, (\mathbf{X}_t)_{j,:})$ are more likely to be partitioned into the same cluster. Our experiments further demonstrate that the extraction of $\mathbf{X}_t$ is essential to ensuring the high GP quality of IGP.

However, the feature dimensionality of $\mathbf{X}_t$ is $N_t$ (i.e., the number of nodes) which may not be fixed over $t$. We introduce an efficient feature extraction module via heavy-edge matching (HEM) graph coarsening (Hendrickson & Leland, 1995) to map $\mathbf{X}_t \in \Re^{N_t \times N_t}$ to a matrix $\mathbf{Z}_t \in \Re^{N_t \times L}$ with fixed dimensionality $L$. For $N_t > L$, we extract reweighted edges $\mathcal{E}_t^{(w)} = \{w(v_i^t, v_j^t)|w(v_i^t, v_j^t) = (\mathbf{X}_t)_{ij}, (\mathbf{A}_t)_{ij} = 1\}$ and apply HEM to $\mathcal{E}_t^{(w)}$. It merges the original graph $\mathcal{G}_t$ with $N_t$ nodes into a supergraph $\mathcal{G}_t^*$ with $L$ supernodes via a greedy strategy of continuously merging the node pair with largest weight in $\mathcal{E}_t^{(w)}$ into a supernode, e.g., merging a graph with 8 nodes into 2 supernodes in Fig. 2. HEM outputs a *coarsening matrix* $\mathbf{C}_t \in \Re^{N_t \times L}$, where $(\mathbf{C}_t)_{ij} = |v_j^{t*}|^{-0.5}$ if node $v_i^t$ is merged into supernode $v_j^{t*}$ and $(\mathbf{C}_t)_{ij} = 0$ otherwise. We then let $\mathbf{Z}_t = \mathbf{X}_t\mathbf{C}_t$ be the reduced features. Since $\mathbf{C}_t$ is a sparse matrix, one can obtain $\mathbf{Z}_t$ by setting its $j$-th column to $(\mathbf{Z}_t)_{:,j} = \sum_{v_i \in v_j^{t*}} (\mathbf{C}_t)_{ij}(\mathbf{X}_t)_{:,i}$. For $N_t < L$, we let $\mathbf{Z}_t = [\mathbf{X}_t, \mathbf{0}_{N_t \times (L-N_t)}]$. In contrast to existing methods using HEM to reduce topology complexities (Dhillon et al., 2007), we are the first to extract feature input $\mathbf{Z}_t$ for inductive GNNs (with non-fixed $N_t$) via HEM. It is more efficient than dimensionality reduction (e.g., PCA of $\mathbf{A}_t$) in SOTA GNNs (Nazi et al., 2019), since HEM is widely used in fast GP methods (Liang et al., 2021). The derived $\mathbf{Z}_t$ is also more informative than the input features in some classic settings of inductive GNNs (e.g., let $\mathbf{Z}_t$ be a constant matrix (Xu et al., 2019)).

Let $\mathbf{F}_t^{(l-1)}$ and $\mathbf{F}_t^{(l)}$ be the input and output of the $l$-th layer of *feature encoder* or *label-induced encoder* in $G$ with $\mathbf{F}_t^{(0)} = \mathbf{Z}_t$. The $l$-th GNN encoder layer is defined as

$$\mathbf{F}_t^{(l)} = \tanh(\hat{\mathbf{D}}_t^{-0.5}\hat{\mathbf{A}}_t\hat{\mathbf{D}}_t^{-0.5}\mathbf{F}_t^{(l-1)}\mathbf{W}_G^{(l-1)}), \tag{5}$$

where $\hat{\mathbf{A}}_t = \mathbf{A}'_t + \mathbf{I}_{N_t}$ ($\mathbf{A}'_t \in \{\mathbf{A}_t, \mathbf{A}_t^{(g)}\}$) is the adjacency matrix with self-edges; $\hat{\mathbf{D}}_t = \mathrm{diag}(\hat{d}_1^t, \cdots, \hat{d}_{N_t}^t)$ is the diagonal degree matrix; $\mathbf{W}_G^{(l-1)}$ is the trainable parameter matrix shared by the two encoders; $(\mathbf{F}_t^{(l)})_{i,:}$ is the latent feature vector of node $v_i^t$. In (5), $(\mathbf{F}_t^{(l)})_{i,:}$ is the nonlinear aggregation of the features of $\{v_i^t\} \cup n(v_i^t)$ in the previous layer, where $n(v_i^t)$ is the neighbor set of $v_i^t$. It is also known as the *message passing* of GNN, where neighbors of $v_i^t$ propagate their features to $v_i^t$ for aggregation. By **Fact 1**, $\mathbf{A}_t^{(g)}$ ensures that ***message passing* of the *label-induced encoder* only occurs in each connected component w.r.t. each cluster in the 'ground-truth'** (e.g., the 2 components in Fig. 2), while ***message passing* of the *feature encoder* occurs in the original topology of** $\mathcal{G}_t$. The last layers of the *feature* and *label-induced encoders* output the graph embedding $\mathbf{U}_t \in \Re^{N_t \times k}$ and *auxiliary label-induced embedding* $\mathbf{U}_t^{(g)} \in \Re^{N_t \times k}$, where $\mathbf{U}_t^{(g)} = G(\mathbf{A}_t^{(g)}, \mathbf{X}_t)$ preserves more informative label-induced properties than $\mathbf{U}_t = G(\mathbf{A}_t, \mathbf{X}_t)$. We further use the non-linear inner product of $\mathbf{U}_t$ to reconstruct the neighbor-induced features $\mathbf{X}_t$:

$$\tilde{\mathbf{X}}_t = \tanh(\mathbf{U}_t\mathbf{U}_t^T). \tag{6}$$

The discriminator $D$ is an auxiliary classifier to distinguish $\mathbf{U}_t^{(g)}$ from $\mathbf{U}_t$, while the generator $G$ tries to generate plausible embedding $\mathbf{U}_t$ to fool $D$. Such an adversarial process helps $G$ to generate embedding $\mathbf{U}_t$ close to $\mathbf{U}_t^{(g)}$. We denote $D$ as $\mathbf{y}_t = D(\mathbf{S}_t; \delta_D)$, where $\mathbf{S}_t \in \{\mathbf{U}_t, \mathbf{U}_t^{(g)}\}$ and $\delta_D$ are the input and set of model parameters; $\mathbf{y}_t \in \Re^{N_t}$ is a column vector with $(\mathbf{y}_t)_i$ as the probability that $(\mathbf{S}_t)_{i,:} = (\mathbf{U}_t^{(g)})_{i,:}$ rather than $(\mathbf{S}_t)_{i,:} = (\mathbf{U}_t)_{i,:}$. Let $\mathbf{P}_t^{(l-1)}$ and $\mathbf{P}_t^{(l)}$ be the input and output of

the $l$-th layer in $D$ with $\mathbf{P}_t^{(0)} = \mathbf{S}_t$. The $l$-th layer of $D$ is defined as

$$\mathbf{P}_t^{(l)} = \text{ReLU}(\mathbf{P}_t^{(l-1)}\mathbf{W}_D^{(l-1)} + \mathbf{b}_D^{(l-1)}), \tag{7}$$

where $\{\mathbf{W}_D^{(l-1)}, \mathbf{b}_D^{(l-1)}\}$ are trainable model parameters. In particular, we use $\text{sigmoid}$ as the activation function of the last layer instead of ReLU.

**Model Optimization.** As in Fig. 2, the *offline* training of IGP is based on three objectives: (i) adversarial learning (AL), (ii) feature reconstruction (FR), and (iii) clustering regularization (CR).

The AL objective helps incorporate the permutation invariant label information of historical graphs to the unsupervised embedding learning via an adversarial process between $D$ and $G$. On the one hand, $D$ tries to distinguish $\mathbf{U}_t^{(g)}$ from $\mathbf{U}_t$. The objective of $D$ w.r.t. a graph $\mathcal{G}_t$ is

$$\arg\min_{\delta_D} L_D(\mathcal{G}_t) = -\left[\sum\nolimits_i \log(1 - D(\mathbf{U}_t)_i) + \sum\nolimits_i \log D(\mathbf{U}_t^{(g)})_i\right]/N_t. \tag{8}$$

On the other hand, $G$ tries to fool $D$ by minimizing the following loss w.r.t. $\mathcal{G}_t$:

$$L_{\text{AL}}(\mathcal{G}_t) = -\left[\sum\nolimits_i \log D(\mathbf{U}_t)_i\right]/N_t. \tag{9}$$

Such an adversarial process directs $G$ to output the embedding $\mathbf{U}_t$ that is close to $\mathbf{U}_t^{(g)}$, thus enabling $G$ to capture the permutation invariant label information of historical training graphs.

The FR loss forces $G$ to derive the embedding $\mathbf{U}_t$ that encodes the key properties of the neighbor-induced features $\mathbf{X}_t$ w.r.t. $\mathcal{G}_t$ by minimizing the reconstruction error between $\mathbf{X}_t$ and $\tilde{\mathbf{X}}_t$:

$$L_{\text{FR}}(\mathcal{G}_t) = ||\tilde{\mathbf{X}}_t - \mathbf{X}_t||_F^2. \tag{10}$$

In addition to the AL loss, IGP can also capture the permutation invariant label information via the regularization of several GP objectives, e.g., modularity maximization (4) and NCut minimization (2). We introduce the CR objective for a graph $\mathcal{G}_t$ that minimizes the following loss:

$$L_{\text{CR}}(\mathcal{G}_t) = -\text{tr}(\mathbf{H}_t^T\tilde{\mathbf{X}}_t\mathbf{H}_t), \tag{11}$$

where $\mathbf{H}_t$ is the membership indicator encoding the 'ground-truth' with the same definitions in (4) and (2). Since the constraints on $\mathbf{H}_t$, e.g., $\text{tr}(\mathbf{H}_t^T\mathbf{H}_t) = N_t$ and $\mathbf{H}_t^T\mathbf{H}_t = \mathbf{I}_{K_t}$, are always satisfied for the given 'ground-truth', we do not need to consider the discrete constraints on $\mathbf{H}_t$. To the best of our knowledge, we are the first to use the CR objective to incorporate the permutation invariant label information (encoded by $\mathbf{H}_t$ of historical training data) to the unsupervised embedding learning, whereas $\mathbf{H}_t$ is the parameter to be optimized or output in existing GP methods (Wilder et al., 2019).

Finally, we derive the objective of $G$ w.r.t. $\mathcal{G}_t$ by combining (9), (10), and (11):

$$\arg\min_{\delta_G} L_G(\mathcal{G}_t) = L_{\text{AL}}(\mathcal{G}_t) + \alpha L_{\text{FR}}(\mathcal{G}_t) + \beta L_{\text{CR}}(\mathcal{G}_t), \tag{12}$$

where $\{\alpha, \beta\}$ are hyper-parameters to balance $L_{\text{FR}}$ and $L_{\text{CR}}$.

In the joint *offline* optimization of $D$ and $G$, we use the Xavier method to initialize model parameters $\{\delta_D, \delta_G\}$. The Adam optimizer is applied to iteratively update $\delta_D$ and $\delta_G$ based on the gradients of loss (8) and (12). In each epoch, we randomly sample a certain number of historical graphs $p$ from the training set to update $\{\delta_D, \delta_G\}$. Finally, we save the model parameters $\{\delta_D^*, \delta_G^*\}$ that result in the best average GP quality on the validation set within a certain number of epochs $n$.

After *offline* training, we generalize IGP to new graphs for fast *online* GP. As in Fig. 1, for each new graph $\mathcal{G}_{t'}$, we derive its embedding $\mathbf{U}_{t'}$ by passing $\{\mathbf{A}_{t'}, \mathbf{X}_{t'}\}$ forward the *feature encoder* of $G$ with its parameters $\delta_G$ fixed. As we assume that the number of clusters $K_{t'}$ is given, we apply $K$Means to $\mathbf{U}_{t'}$, which outputs the GP result w.r.t. the given $K_{t'}$. Therefore, the runtime of *online* GP on $\mathcal{G}_{t'}$ includes (i) the feature extraction of $\mathbf{Z}_t$, (ii) one feedforward propagation through the *feature encoder*, and (iii) the downstream clustering. Due to the space limit, we conclude the *offline* training and *online* generalization procedures of IGP in Appendix B.

## 4 EXPERIMENTS

**Datasets.** We evaluate IGP on 2 synthetic benchmarks (with 7 settings) and 4 real datasets. Statistics of the datasets are depicted in Table 1, where $T$, $N$, $|\mathcal{E}|$, and $K$ are the number of graphs, nodes,

edges, and clusters. The 11 datasets cover the cases with (i) fixed $N$ and $K$, (ii) fixed $N$ but non-fixed $K$, as well as (iii) non-fixed $N$ and $K$. *GN-Net* (Girvan & Newman, 2002) and *LFR-Net* (Lancichinetti et al., 2008) are synthetic benchmarks that can simulate properties of real-world network systems. In each benchmark, we generated a set of graphs and their GP ground-truth via several distributions. *GN-Net* and *LFR-Net* use $p_{\text{in}}$ and $\mu$ to control clustering structures of each graph. We denote the *GN-Net* with a setting of $p_{\text{in}}$ as *GN-$p_{\text{in}}$* ($p_{\text{in}} \in \{0.5, 0.4, 0.3\}$). The 3 datasets of *GN-Net* are with fixed $N$ and $K$. We use $L(c, \mu)$ ($c \in \{f, n\}$, $\mu \in \{0.3, 0.6\}$) to represent the *LFR-Net* with a setting of $\mu$, where $f$ and $n$ denote cases with f̲ixed and n̲on-fixed $N$. *As $p_{\text{in}}$ decreases and $\mu$ increases, the clustering structures become increasingly difficult to identify.* *Taxi* (Piorkowski et al., 2009), *Reddit* (Yanardag & Vishwanathan, 2015), *Enron* (Rossi & Ahmed, 2015), and *AS* (Leskovec et al., 2005) are real datasets from a vehicle network, an email network, a social network, and an autonomous system. As the 4 real datasets do not provide ground-truth regarding $K$, to ensure the fairness of comparison, where *each graph is with a common $K$ for all the methods to be evaluated*, we adopted a widely-used strategy that uses the auxiliary *Louvain* algorithm (Blondel et al., 2008) to estimate $K$ for each graph. Details of the datasets are given in Appendix C.2.

**Baselines.** We compared IGP over 15 baselines with 4 categories. **(i)** *SNMF* (Wang et al., 2011) and *spectral clustering* (*SC*) (Von Luxburg, 2007) are classic *transductive* GP methods. **(ii)** *VERSE* (Tsitsulin et al., 2018), *MILE* (Liang et al., 2021), and *PhUSION* (*PhN*) (Zhu et al., 2021) are graph embedding approaches for high-quality representations, while *NPR* (Yang et al., 2020), *RandNE* (Zhang et al., 2018a), and *ProNE* (Zhang et al., 2019) are efficient embedding methods with fast approximation. All these embedding methods are *transductive*. **(iii)** *Metis* (Hendrickson & Leland, 1995), *hMetis* (Selvakkumaran & Karypis, 2006), and *GraClus* (Dhillon et al., 2007) are fast *transductive* GP baselines with heuristic approximation. **(iv)** *GraSAGE* (*GSAGE*) (Hamilton et al., 2017) and *GAT* (Veličković et al., 2018) are SOTA *inductive* GNNs, while *GAP* (Nazi et al., 2019) and *ClusNet* (Wilder et al., 2019) are E2E methods based on *inductive* GNNs. Besides the 15 methods, we also include other strong baselines (e.g., *node2vec*) with details given in Appendix C.3.

Both variants of IGP were evaluated. For simplicity, we use IGP-M and IGP-C to denote the variants adopting the objectives of modularity maximization and NCut minimization, respectively.

**Quantitative Evaluation.** In our quantitative evaluation, we consider GP on graphs where $K$ is given. On all the datasets, we adopted quality metrics of **modularity** and **NCut** as in (3) and (1). On datasets with ground-truth (i.e., *GN-Net* and *LFR-Net*), we also used quality metrics of **NMI** and accuracy (**AC**) (Cui et al., 2018) to measure the correspondence between GP results and ground-truth. Usually, larger **modularity**, **NMI** and **AC**, as well as smaller **NCut** imply better quality. Moreover, the total **runtime** of a method to derive its final GP result was used as the efficiency metric with lower **runtime** indicating higher efficiency.

Most conventional baselines (e.g., *SC*) are *transductive*, which can only be applied to a static graph, but IGP alleviates the NP-hard challenge of GP in an *inductive* framework across graphs. To illustrate such superiority of IGP beyond *transductive* baselines, for each dataset, we used the first $80\%$ of the graphs as the training set $\Gamma_{\text{T}}$ with the remaining $10\%$ and $10\%$ deployed as the validation set $\Gamma_{\text{V}}$ and test set $\Gamma'$. In this setting, $\Gamma_{\text{T}} \cup \Gamma_{\text{V}}$ and $\Gamma'$ represent the sets of historical and newly generated graphs, respectively. We conducted the *offline* training of IGP on $\Gamma_{\text{T}}$ and generalized it to $\Gamma'$ for *online* GP. In the *offline* training, we save the model parameters of IGP with the best average quality on $\Gamma_{\text{V}}$, where we use NMI and modularity as the validation quality metric for datasets with and without ground-truth, respectively. Since we focus on the trade-off between quality and efficiency on new graphs, the evaluation was conducted on $\Gamma'$, where we had to apply *transductive* baselines to each test graph in $\Gamma'$ from scratch due to their *transductive* nature.

For *inductive* GNN baselines (i.e., *GraSAGE* and *GAT*), we use the same training and evaluation settings with IGP, i.e., *offline* training on $\Gamma_{\text{T}}$ and *online* generalization to $\Gamma'$. Since we consider GP on graphs without attributes, we use a constant matrix $\mathbf{1}_{N_t \times L}$ (with fixed feature dimensionality $L$), where each entry is set to 1, as the input features of these GNNs, which is a widely-used setting of inductive GNNs for the case without attributes (Xu et al., 2019). In the *offline* training of *inductive* GNN baselines, we adopted the widely-used unsupervised loss of *GraSAGE* (Hamilton et al., 2017) to train them on $\Gamma_{\text{T}}$, since most existing inductive GNNs (even with a supervised loss like cross-entropy) cannot directly integrate the permutation invariant label information from $\Gamma_{\text{T}}$.

Table 1: Statistics of the datasets.

| | $T$ | $N$ | $|\mathcal{E}|$ | $K$ |
|---|---|---|---|---|
| $GN$-0.5 | 2,000 | 5,000 | 48,065-49,314 | 250 |
| $GN$-0.4 | 2,000 | 5,000 | 48,335-49,698 | 250 |
| $GN$-0.3 | 2,000 | 5,000 | 48,488-49,981 | 250 |
| $L(f,0.3)$ | 2,000 | 5,000 | 22,539-25,707 | 57-104 |
| $L(f,0.6)$ | 2,000 | 5,000 | 22,585-25,639 | 58 - 107 |
| $L(n,0.3)$ | 2,000 | 5,000-5,999 | 22,999-29,670 | 59-117 |
| $L(n,0.6)$ | 2,000 | 5,000-6,000 | 22,680-29,931 | 61-118 |
| $Taxi$ | 3,000 | 1,279 | 38,554-40,171 | 7-12 |
| $Reddit$ | 1,870 | 501-3,648 | 525-4,780 | 14-121 |
| $Enron$ | 410 | 502-3,261 | 554-5,097 | 10-31 |
| $AS$ | 733 | 103-6,474 | 487-26,467 | 7-39 |

Table 2: Ablation study on $L(n,.6)$.

| | NMI(%↑) | AC(%↑) | Mod(%↑) | Ncut(↓) |
|---|---|---|---|---|
| **IGP-M** | **56.48** | **53.42** | **34.22** | **82.76** |
| w/o AL | 52.13 | 49.30 | 33.35 | 84.05 |
| w/o FR | 11.19 | 6.28 | 10.08 | 1.01e3 |
| w/o CR | 49.97 | 46.70 | 32.70 | 86.08 |
| w/o $\mathbf{Z}_t$ | 16.39 | 6.14 | 1.27 | 3.01e4 |
| w/o GNN | 1.14 | 3.92 | 0.01 | 1.08e4 |
| **IGP-C** | **52.71** | **47.47** | **31.95** | **92.40** |
| w/o AL | 35.12 | 26.15 | 26.70 | 113.10 |
| w/o FR | 15.46 | 5.46 | 0.45 | 2.08e4 |
| w/o CR | 35.16 | 26.22 | 26.78 | 111.62 |
| w/o $\mathbf{Z}_t$ | 13.88 | 5.57 | 0.28 | 2.88e4 |
| w/o GNN | 8.45 | 4.78 | 0.32 | 5.12e4 |

For the E2E GNN methods (i.e., *GAP* and *ClusNet*), we could only conduct the *offline* training and fast *online* generalization on datasets with fixed $K$ (i.e., on *GN-Net*), but we had to train them from scratch on the test set of other datasets with non-fixed $K$, due to the fixed dimensionality of their E2E output layers. As recommended in (Nazi et al., 2019) and (Wilder et al., 2019), we used *PCA* and *node2vec* to map the adjacency matrices $\{\mathbf{A}_t\}$ to a feature space with fixed dimensionality for the input features of *GAP* and *ClusNet*. Moreover, we used their own unsupervised training loss, which approximates the objectives of NCut minimization and modularity maximization, respectively.

For all the embedding-based methods (including the *transductive* embedding baselines, *inductive* GNNs, and IGP), we used *KMeans* as the downstream clustering module. We ran *KMeans* 10 times to avoid its limitation of getting locally optimal solutions, which is included in the total runtime. To ensure the fairness of comparison, the embedding dimensionality of all the embedding-based methods was set to be the same on each dataset. For other E2E baselines (e.g., *Metis*, *GAP*, and *ClusNet*), we directly derived the GP results from their outputs. Since *Taxi*, *Reddit*, *Enron*, and *AS* do not provide the GP ground-truth (in terms of the membership $C_t$) on each historical training snapshot of these datasets, we used the GP result of the baseline with best average modularity on $\Gamma_V$ to derive the membership indicator $\mathbf{H}_t$ for IGP, which is the second source of historical label information defined in Section 2. Note that we do not need to derive $\mathbf{H}_t$ in the evaluation phase when generalizing the trained model to test set $\Gamma'$.

We tuned parameters of all the methods on $\Gamma_V$ of each dataset to report best quality metrics. Both the mean and standard deviation of all the quality and efficiency metrics on $\Gamma'$ of each dataset were recorded. Evaluation results in terms of average **NMI**, **modularity**, and **runtime** are visualized in Fig. 3. Due to the space limit, we give (i) the visualization results w.r.t. **AC** and **NCut**, (ii) mean and standard deviation of all the metrics, and (iii) parameter settings in Appendix C.5. In most cases of Fig. 3, IGP-M (red circle) and IGP-C (blue circle) are in the top group with best quality, even though we did not train them on test set $\Gamma'$, where other *transductive* baselines were fully trained. Surprisingly, IGP even has the best or second-best quality metrics in some cases. Moreover, the runtime of IGP-M and IGP-C is also competitive to some fast GP baselines (e.g., *Metis*). Although the runtime of *inductive* GNN baselines (with fast *online* generalization) is slightly faster than IGP, they still suffer from poor quality, which implies that the feature extraction module and hybrid training loss of IGP are essential to ensure its high quality. From the view of alleviating the NP-hard challenge of *online* GP on newly generated graphs, IGP achieves a better trade-off between quality and efficiency. According to the visualization results in Fig. 3, we introduce the **trade-off score** in Appendix C.5, which can quantitatively evaluate the trade-off between quality and efficiency by computing the area covered by an induced rectangle of each data point in the visualized 2D space.

**Ablation Study.** For IGP-M and IGP-C, we also validated the effectiveness of (i) AL loss, (ii) FR loss, (iii) CR loss, (iv) feature input $\mathbf{Z}_t$, and (v) GNN by respectively excluding the corresponding components from the original model. In case (iv), we used a constant matrix $\mathbf{1}_{N_t \times L}$ with all entries set to 1 to replace $\mathbf{Z}_t$ while we use an FC network that only takes $\mathbf{Z}_t$ as input to replace GNNs in $G$ with same layer configurations. Example results on $L(n, .6)$ in terms of **NMI**, **AC**, **modularity**, and **NCut** are shown in Table 2. According to Table 2, the FR loss, $\mathbf{Z}_t$, and GNNs are key components to ensure the high quality of IGP, since there is significant quality decline for case (ii), (iv), and (v). The AL and CR losses further enhance the quality by incorporating permutation invariant label information of historical graphs. Other detailed ablation studies are given in Appendix C.5.

**Extended Applications.** Besides the aforementioned experiments, we consider two extended applications. First, when the number of clusters $K$ of a new graph is not given, IGP also has the potential

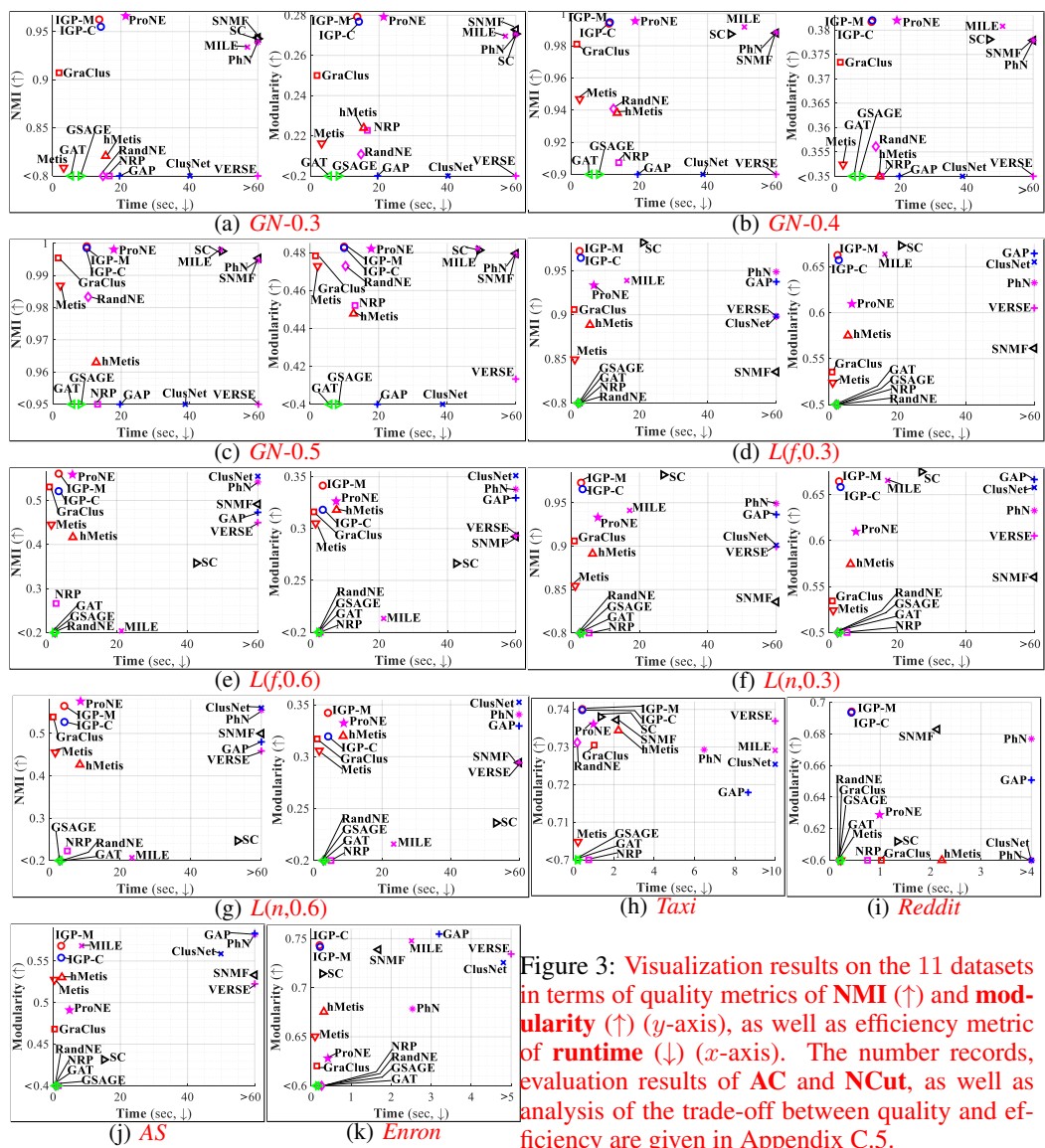

Figure 3: Visualization results on the 11 datasets in terms of quality metrics of **NMI** (↑) and **modularity** (↑) ($y$-axis), as well as efficiency metric of **runtime** (↓) ($x$-axis). The number records, evaluation results of **AC** and **NCut**, as well as analysis of the trade-off between quality and efficiency are given in Appendix C.5.

to estimate $K$, where we apply an auxiliary model selection method (e.g., XMeans) to the derived embedding $\mathbf{U}_t$. Second, in contrast to the hypothesis that *all the given snapshots are from a common system or scenario following several underlying distributions*, it is promising that one conducts the *offline* training on historical data of one scenario but generalizes the model to other scenarios for fast *online* GP. Especially, we consider the application where we train IGP on synthetic graphs but generalize it to real datasets. The two extended applications are elaborated in Appendix D.

## 5 CONCLUSION

In this paper, we proposed a novel IGP framework to alleviate the NP-hard challenge of GP, obtaining a better trade-off between quality and efficiency. In IGP, we first conduct *offline* training of a novel dual GNN structure on historical known graphs and generalize the model to newly generated graphs for fast high-quality *online* GP. IGP is a generic framework that can incorporate the permutation invariant label information of GP to the *offline* training by combining unsupervised GP objectives (e.g., modularity maximization and NCut minimization) with the dual GNN structure. Since we introduced an *inductive* graph embedding scheme and an efficient feature extraction module, IGP can also tackle *online* GP on graphs with non-fixed number of nodes and clusters. In Appendix E, we give discussions of possible future work, e.g., further reducing the runtime, scaling IGP up to graphs with large number of nodes, and transferring IGP across scenarios.

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

## A  PROOF OF FACT 1

By the disjoint constraint of GP, i.e., $C_r^t \cap C_s^t = \emptyset$ for $\forall r \neq s$, each node can only belong to one unique cluster. For the *membership indicator matrix* $\mathbf{R}_t$, only one entry in each row of $\mathbf{R}_t$ must be 1 with the rest entries in the same row set to 0. For the *auxiliary adjacency matrix* $\mathbf{A}_t^{(g)} = \mathbf{R}_t \mathbf{R}_t^T$, consider each entry $(\mathbf{A}_t^{(g)})_{ij} = \sum_{r=1}^{K_t} [(\mathbf{R}_t)_{ir}(\mathbf{R}_t)_{jr}]$ with respect to the pair of nodes $(v_i^t, v_j^t)$.

If nodes $(v_i^t, v_j^t)$ are in the same cluster $C_s^t$, i.e., $v_i^t, v_j^t \in C_s^t$, we have $\mathbf{R}_{is} = \mathbf{R}_{js} = 1$. We can further derive $(\mathbf{A}_t^{(g)})_{ij} = (\mathbf{R}_t)_{is}(\mathbf{R}_t)_{js} = 1$. It indicates that there is an edge between $(v_i^t, v_j^t)$ with weight 1, when they are in the same cluster.

If nodes $(v_i^t, v_j^t)$ are not in the same cluster, i.e., $v_i^t \in C_r^t$ and $v_j^t \in C_s^t$ $(r \neq s)$, we have $\mathbf{R}_{ir} = 1$ and $\mathbf{R}_{js} = 1$, but $(\mathbf{A}_t^{(g)})_{ij} = 0$. Namely, there is no edge between nodes $(v_i^t, v_j^t)$ when they are in different clusters.

In summary, there is an edge between $(v_i^t, v_j^t)$ in the *auxiliary label-induced graph* $\mathcal{G}_t^{(g)}$ only if they are partitioned into the same cluster according to the given partitioning result $C_t = \{C_1^t, \cdots, C_{K_t}^t\}$ (used as the 'ground-truth'). Especially, each node $v_i^t \in C_r^t$ (i) must have edges connected to all the other nodes in the same cluster $C_r^t$ and (ii) do not have edges connected to nodes in different clusters $C_s^t$ $(r \neq s)$, thus forming a fully connected component with respect to cluster $C_r^t$. Hence, for a graph $\mathcal{G}_t$ with $K_t$ clusters, its corresponding *auxiliary label-induced graph* $\mathcal{G}_t^{(g)}$ has $K_t$ fully connected components, with each component corresponding to one unique cluster of $\mathcal{G}_t$.

## B  DETAILED ALGORITHMS OF IGP

In contrast to conventional *transductive* GP methods applied to a single static graph, IGP is an *inductive* framework that explores the structure properties across a set of graphs $S = \{\mathcal{G}_1, \cdots, \mathcal{G}_T\}$. In IGP, we first train a novel dual GNN structure on historical system snapshots in an *offline* way (regardless of the training time cost). We then generalize the trained model to newly generated snapshots of the same system for fast high-quality *online* GP without additional optimization, where a better trade-off between quality and efficiency can be achieved. For simplicity, we divide $S$ into a training set $\Gamma_T$, a validation set $\Gamma_V$, and a test set $\Gamma'$, where $\Gamma_T \cup \Gamma_V$ and $\Gamma'$ represent the set of historical and newly generated system snapshots, respectively. Namely, snapshots in $\Gamma_T \cup \Gamma_V$ should occur before those in $\Gamma'$.

---

**Algorithm 1:** HEM Graph Coarsening

---

**Input:** input graph $\mathcal{G}_t$; neighbor-induced features $\mathbf{X}_t$; number of nodes $N_t$; reduced dimensionality $L$

**Output:** coarsening matrix $\mathbf{C}_t$

1   extract the node set $\mathcal{V}_t^{(0)}$ and reweighted edges $\mathcal{E}_t^{(w)}$ from $\mathcal{G}_t$ and $\mathbf{X}_t$

2   $\mathcal{E}_t^{(0)} \leftarrow \mathcal{E}_t^{(w)}$ //Initialize the edge set of the 1st coarsening level

3   $k \leftarrow 0$ //Initialize the level index

4   $N_t' \leftarrow |\mathcal{V}_t^{(0)}|$ //Initialize the number of (super)nodes

5   **while** $N_t' > L$ **do**

6     $\mathcal{V}_t^{(k+1)} \leftarrow \emptyset$ //Initialize the node set of next level

7     $\mathcal{E}_t^{(k+1)} \leftarrow \emptyset$ //Initialize the edge set of next level

8     sort $\mathcal{E}_t^{(k)}$ based on weights in descending order

9     arrange the sorted result as a queue $q(\mathcal{E}_t^{(k)})$

10     **while** $q(\mathcal{E}_t^{(k)}) \neq \emptyset$ **do**

11       pop $(v_i^{(k)}, v_j^{(k)})$ from $q(\mathcal{E}_t^{(k)})$

12       **if** $v_i^{(k)} \in \mathcal{V}_t^{(k)}$ **and** $v_j^{(k)} \in \mathcal{V}_t^{(k)}$ **then**

13         delete $v_i^{(k)}$ and $v_j^{(k)}$ from $\mathcal{V}_t^{(k)}$

14         merge $v_i^{(k)}$ and $v_j^{(k)}$ into supernode $v_l^{(k+1)}$

15         add $v_l^{(k+1)}$ to $\mathcal{V}_t^{(k+1)}$

16         adjust $\mathcal{E}_t^{(k)}$ and $\mathcal{E}_t^{(k+1)}$ w.r.t. $\mathcal{V}_t^{(k)}$ and $\mathcal{V}_t^{(k+1)}$

17         $N_t' \leftarrow N_t' - 1$ //Update the number of supernodes

18       **if** $N_t' = L$ **then**

19         $\mathcal{V}_t^{(k+1)} \leftarrow \mathcal{V}_t^{(k+1)} \cup \mathcal{V}_t^{(k)}$

20         **return** $\mathbf{C}_t$ based on the merging membership of $\mathcal{V}_t^{(k+1)}$

21     $\mathcal{V}_t^{(k+1)} \leftarrow \mathcal{V}_t^{(k+1)} \cup \mathcal{V}_t^{(k)}$

22     $\mathcal{E}_t^{(k+1)} \leftarrow \mathcal{E}_t^{(k+1)} \cup \mathcal{E}_t^{(k)}$

23     $k \leftarrow k + 1$ //Update the level index $k$

24   **return** $\mathbf{C}_t$ based on the merging membership of $\mathcal{V}_t^{(k)}$

---

### B.1 FEATURE EXTRACTION

For a set of graph snapshots $S = \{\mathcal{G}_1, \ldots \mathcal{G}_T\}$ with non-fixed node set $\{\mathcal{V}_1, \ldots, \mathcal{V}_T\}$ (i.e., with non-fixed number of nodes $N_t$), we introduce an efficient feature extraction strategy via HEM graph coarsening, which maps the neighbor-induced feature $\mathbf{X}_t \in \Re^{N_t \times N_t}$ ($\mathbf{X}_t = \mathbf{Q}_t$ or $\mathbf{X}_t = \mathbf{M}_t$) to a reduced feature matrix $\mathbf{Z}_t \in \Re^{N_t \times L}$ with fixed dimensionality $L$. Algorithm 1 and Algorithm 2 summarize the procedures of HEM graph coarsening and the feature extraction module.

For each graph $\mathcal{G}_t \in S$, if $L < N_t$, Algorithm 1 first merges $\mathcal{G}_t$ into a supergraph $\mathcal{G}_t^*$ (with $L$ supernodes) based on reweighted edges $\mathcal{E}_t^{(w)} = \{w(v_i^t, v_j^t) | w(v_i^t, v_j^t) = (\mathbf{X}_t)_{ij}, (\mathbf{A}_t)_{ij} = 1\}$ and outputs a coarsening matrix $\mathbf{C}_t \in \Re^{N_t \times L}$ that describes the merging membership. Concretely, $(\mathbf{C}_t)_{ij} = |v_j^{t*}|^{-0.5}$ if node $v_i^t$ is merged into supernode $v_j^{t*}$ (i.e., $v_i^t \in v_j^{t*}$) and $(\mathbf{C}_t)_{ij} = 0$, otherwise. Based on $\mathbf{C}_t$ given by Algorithm 1, Algorithm 2 further derives a reduced feature matrix $\mathbf{Z}_t$ via $\mathbf{Z}_t = \mathbf{X}_t \mathbf{C}_t$. By the sparsity of $\mathbf{C}_t$, we can set its $j$-th column to $(\mathbf{Z}_t)_{:,j} = \sum_{v_i \in v_j^{t*}} (\mathbf{C}_t)_{ij} (\mathbf{X}_t)_{:,i}$. In contrast, when $L > N_t$, Algorithm 2 directly derives $\mathbf{Z}_t = [\mathbf{X}_t, \mathbf{0}_{N_t \times (L - N_t)}]$ via a simple padding strategy.

### B.2 OFFLINE TRAINING ON HISTORICAL SYSTEM SNAPSHOTS

In the *offline* training of IGP, we assume that the partitioning result $C_t = \{C_1^t, \cdots, C_{K_t}^t\}$ (i.e., 'ground-truth') of each historical graph $\mathcal{G}_t \in \Gamma_T \cup \Gamma_V$ is available, which is permutation invariant in terms of cluster labels. For simplicity, we correlate each historical graph $\mathcal{G}_t \in \Gamma_T \cup \Gamma_V$ with a 2-tuple $(\mathbf{A}_t, C_t)$. The *offline* training procedure of IGP is summarized in Algorithm 3, where $L$ is fixed dimensionality for the reduced features $\mathbf{Z}_t$; $\{\alpha, \beta\}$ are hyper-parameters of the training loss

---

**Algorithm 2:** Feature Extraction of IGP

---

**Input:** input graph $\mathcal{G}_t$; neighbor-induced features $\mathbf{X}_t$; number of nodes $N_t$; reduced dimensionality $L$
**Output:** reduced features $\mathbf{Z}_t$

1 **if** $L \geq N_t$ **then**
2    let $\mathbf{Z}_t = [\mathbf{X}_t, \mathbf{0}_{N_t \times (L-N_t)}]$ //Pad $\mathbf{Z}_t$ with zeros
3 **else if** $L < N_t$ **then**
4    derive the coarsening matrix $\mathbf{C}_t \in \Re^{N_t \times L}$ via Algorithm 1
5    derive the merging membership $\mathcal{V}_t^*$ based on $\mathbf{C}_t$
6    **for each** *supernode* $v_j^* \in \mathcal{V}_t^*$ **do**
7      $(\mathbf{Z}_t)_{:,j} \leftarrow \mathbf{0}_{N_t \times 1}$//Initialize the $j$-th column of $\mathbf{Z}_t$
8      **for each** *node* $v_i \in v_j^*$ **do**
9        $(\mathbf{Z}_t)_{:,j} \leftarrow (\mathbf{Z}_t)_{:,j} + (\mathbf{C}_t)_{ij}(\mathbf{X}_t)_{:,i}$

---

**Algorithm 3:** *Offline* Training of IGP (on Historical System Snapshots)

---

**Input:** training set $\Gamma_\mathrm{T}$; validation set $\Gamma_\mathrm{V}$; reduced feature dimensionality $L$; hyper-parameters of training loss $\{\alpha, \beta\}$; number of historical snapshots sampled in each epoch $p$; repetitions to update model parameters (w.r.t. each sampled snapshot in a epoch) $m$; learning rates $\{\eta_D, \eta_G\}$; maximum number of epochs $n$
**Output:** best model parameters $\{\delta_G^*, \delta_D^*\}$

1 initialize the variable $\bar{q}^*$ that saves the best average quality metric
2 initialize model parameters $\{\delta_G, \delta_D\}$ via the Xavier algorithm
3 **for** *epoch* **from** 1 **to** $n$ **do**
4    **for** *sample_count* **from** 1 **to** $p$ **do**
5      randomly sample a graph $\mathcal{G}_t \in \Gamma_\mathrm{T}$
6      get the associated 2-tuple $(\mathbf{A}_t, C_t)$ of $\mathcal{G}_t$
7      derive the neighbor-induced features $\mathbf{X}_t$ based on adjacency matrix $\mathbf{A}_t$
8      derive the reduced features $\mathbf{Z}_t$ from $\{\mathbf{A}_t, \mathbf{X}_t\}$ via Algorithm 2
9      derive the membership indicator $\mathbf{H}_t$ based on the 'ground-truth' $C_t$
10      **for** *update_count* **from** 1 **to** $m$ **do**
11        derive the loss function $L_D$ via equation (8)
12        update $\delta_D$ via $\delta_D \leftarrow \text{Optimizer}(\eta_D, \delta_D, \frac{\partial L_D}{\partial \delta_D})$ with $\delta_G$ fixed
13        derive the loss function $L_G$ via equation (12)
14        update $\delta_G$ via $\delta_G \leftarrow \text{Optimizer}(\eta_G, \delta_G, \frac{\partial L_G}{\partial \delta_G})$ with $\delta_D$ fixed
15    $\bar{q} \leftarrow 0$ //Initialize the variable that saves the average quality metric of current epoch
16    **for each** $\mathcal{G}_{t'} \in \Gamma_\mathrm{V}$ **do**
17      get the associated 2-tuple $(\mathbf{A}_{t'}, C_{t'})$ of $\mathcal{G}_{t'}$
18      derive the neighbor-induced features $\mathbf{X}_{t'}$ based on adjacency matrix $\mathbf{A}_{t'}$
19      derive the reduced features $\mathbf{Z}_{t'}$ from $\{\mathbf{A}_{t'}, \mathbf{X}_{t'}\}$ via Algorithm 2
20      derive the graph embedding $\mathbf{U}_{t'}$ from the *feature encoder* in G
21      apply $K$Means to $\mathbf{U}_{t'}$ to derive partitioning result $\tilde{C}_{t'}$
22      evaluate partitioning quality $q$ based on $\tilde{C}_{t'}$ (and $C_{t'}$)
23      $\bar{q} \leftarrow \bar{q} + q$ //Update $\bar{q}$
24    $\bar{q} \leftarrow \bar{q}/|\Gamma_V|$ //Compute the average quality metric
25    **if** $\bar{q}$ better than $\bar{q}^*$ **then**
26      $\{\delta_D^*, \delta_G^*\} \leftarrow \{\delta_D, \delta_G\}$ //Update the best model parameters
27      $\bar{q}^* \leftarrow \bar{q}$ //Update the best average partitioning quality

---

$L_G$; $p$ is the number of historical snapshots sampled in each epoch; $\{\eta_D, \eta_G\}$ are learning rates of the optimizers updating $\{\delta_D, \delta_G\}$; $n$ is the maximum number of epochs. For a selected training sample in each epoch, repeatedly updating model parameters $m$ ($m \geq 1$) times can sometimes lead to better partitioning quality (on the validation and test set). Algorithm 3 finally saves the model parameters $\{\delta_D^*, \delta_G^*\}$ with the best quality metric on the validation set $\Gamma_\mathrm{V}$.

---

**Algorithm 4:** *Online* Generalization of IGP (to Newly Generated System Snapshots)

---

**Input:** newly generated graph $\mathcal{G}_{t'} \in \Gamma'$, adjacency matrix $\mathbf{A}_{t'}$, reduced feature dimensionality $L$, number of clusters $K_{t'}$ (w.r.t. $\mathcal{G}_{t'}$)

**Output:** partitioning result $\tilde{C}_{t'}$ w.r.t. $\mathcal{G}_{t'}$

1 derive the neighbor-induced feature $\mathbf{X}_{t'}$ based on adjacency matrix $\mathbf{A}_{t'}$
2 derive the reduced features $\mathbf{Z}_{t'}$ from $\{\mathbf{A}_{t'}, \mathbf{X}_{t'}\}$ via Algorithm 2
3 derive the graph embedding $\mathbf{U}_{t'}$ from the *feature encoder* in $G$
4 apply $K$Means to $\mathbf{U}_{t'}$ to derive the partitioning result $\tilde{C}_{t'}$ with $K_{t'}$ clusters

---

### B.3 Online Generalization to Newly Generated System Snapshots

In the *online* generalization of IGP, we fix the learned model parameters $\{\delta_D^*, \delta_G^*\}$ and directly derive graph embedding $\mathbf{U}_{t'}$ with respect to each newly generated graph $\mathcal{G}_{t'} \in \Gamma'$ from *feature encoder* of $G$. Since we assume that the number of clusters $K_{t'}$ of each graph $\mathcal{G}_{t'} \in \Gamma'$ is given in the *online* GP, we apply $K$Means to $\mathbf{U}_{t'}$ to derive the partitioning result $C'_{t'}$ with $K_{t'}$ clusters. In this case, the total runtime of *online* GP on each newly generated graph $\mathcal{G}_{t'} \in \Gamma'$ includes (i) the efficient feature extraction of $\mathbf{Z}_{t'}$, (ii) one feedforward propagation of $\{\mathbf{A}_{t'}, \mathbf{Z}_{t'}\}$ through *feature encoder*, and (iii) the downstream ($K$Means) clustering. In conclusion, we summarize the *online* generalization procedure of IGP in Algorithm 4.

### B.4 Complexity Analysis

For a given graph $\mathcal{G}_t$, we use the efficient matrix multiplication to derive the neighbor-induced features $\mathbf{Q}_t$ and $\mathbf{M}_t$. Concretely, we set $\mathbf{Q}_t = \mathbf{A}_t - \mathbf{d}_t^T \mathbf{d}_t / (2w)$ and $\mathbf{M}_t = \mathbf{D}^{-1/2} \mathbf{A}_t \mathbf{D}^{-1/2}$, where $\mathbf{d}_t = [d_1^t, \cdots, d_{N_t}^t]$ and $\mathbf{D}_t = \mathrm{diag}(d_1^t, \cdots, d_{N_t}^t)$ are the degree vector and degree diagonal matrix of $\mathcal{G}_t$. Although the complexities of computing $\mathbf{Q}_t$ and $\mathbf{M}_t$ are both at most $O(N_t^2)$, they can be easily paralleled via GPUs, which significantly reduces the runtime. Moreover, the complexity of the HEM graph coarsening (i.e., Algorithm 1) is no more than $O(N_t|\mathcal{E}_t| \log |\mathcal{E}_t|)$, with $|\mathcal{E}_t|$ denoting the number of edges in $\mathcal{G}_t$ and $|\mathcal{E}_t| \ll N_t^2$. The complexity to derive $\mathbf{Z}_t$ via $\mathbf{X}_t$ and $\mathbf{C}_t$ (i.e., Algorithm 2) is no more than $O(N_t L n^*)$, where $n^*$ is the maximum number of nodes merged into a supernode; $L$ is the number of supernodes (which is also the reduced feature dimensionality of $\mathbf{Z}_t$). Hence, the complexity to derive the reduced feature matrix $\mathbf{Z}_t$ based on $\mathbf{X}_t$ is no more than $O(N_t|\mathcal{E}_t| \log |\mathcal{E}_t| + N_t L n^*)$.

Let $L^{(l-1)}$ be the feature dimensionality of the $l$-th GNN layer of the *feature encoder* or *auxiliary label-induced encoder* in $G$, where we have $L^{(0)} = L$. We used the efficient sparse-dense matrix multiplication to implement the graph convolutional operation described in equation (5), where the sparsity of $\mathbf{A}_t$ is considered. The complexity of one feedforward propagation through the *feature encoder* is no more than $O(|\mathcal{E}_t|L^{(0)}L^{(1)} + |\mathcal{E}_t|L^{(1)}L^{(2)} + \cdots) = O(|\mathcal{E}_t|LL')$, where $L' = L^{(1)}$ and $L^{(l)} < L^{(l-1)}$. Hence, the overall time complexity of one feedforward propagation through the *feature encoder* is linear in the number of edges of the input graph $\mathcal{G}_t$.

Let $k$ be the dimensionality of graph embedding. Usually, we have $k \ll N_t$. For a graph with $K_t$ clusters, the complexity of $K$Means is no more than $O(N_t K_t k I)$ for $I$ iterations. In particular, the runtime of the downstream clustering (i.e., $K$Means algorithm) can also be significantly reduced via parallel implementations.

## C Experiment Details

### C.1 Evaluation Metrics

In our experiments, we used the (unsupervised) metrics of modularity and NCut to evaluate the partitioning quality of each method on all the datasets. The formal definitions of modularity and NCut are given in equation (3) and equation (1) (see Section 2).

For datasets with partitioning ground-truth (i.e., *GN-Net* and *LFR-Net*), we also used the metrics of normalized mutual information (NMI) and accuracy (AC) (Cui et al., 2018), which can measure the

Table 3: Detailed statistics of the datasets.

| | $T$ | Min $N$ | Max $N$ | Avg $N$ | Min $|\mathcal{E}|$ | Max $|\mathcal{E}|$ | Avg $|\mathcal{E}|$ | Min $K$ | Max $K$ | Avg $K$ |
|---|---|---|---|---|---|---|---|---|---|---|
| GN-0.5 | | | | | 48,065 | 49,314 | 48,752 | | | |
| GN-0.4 | 2,000 | 5,000 | 5,000 | 5,000 | 48,335 | 49,698 | 48,999 | 250 | 250 | 250 |
| GN-0.3 | | | | | 48,488 | 49,981 | 49,246 | | | |
| L(f,0.3) | 2,000 | 5,000 | 5,000 | 5,000 | 22,539 | 25,707 | 23,925 | 57 | 104 | 79 |
| L(f,0.6) | | | | | 22,585 | 25,639 | 23,927 | 58 | 107 | 79 |
| L(f,0.3) | 2,000 | 5,000 | 5,999 | 5,503 | 22,999 | 29,670 | 26,311 | 59 | 117 | 87 |
| L(f,0.6) | | 5,000 | 6,000 | 5,493 | 22,680 | 29,931 | 26,289 | 61 | 118 | 87 |
| Taxi | 3,000 | 1,279 | 1,279 | 1,279 | 38,554 | 40,171 | 39,291 | 7 | 12 | 9 |
| Reddit | 1,870 | 501 | 3,648 | 946 | 525 | 4,780 | 1,133 | 14 | 121 | 32 |
| Enron | 410 | 502 | 3,261 | 1,048 | 554 | 5,097 | 1,376 | 10 | 31 | 17 |
| AS | 733 | 103 | 6,474 | 4,183 | 487 | 26,467 | 16,324 | 7 | 39 | 29 |

correspondence between the partitioning result (given by the method to be evaluated) and ground-truth (given by the dataset). In general, better correspondence indicates better partitioning quality. For a graph $\mathcal{G}_t$ with $K_t$ clusters, we use $H_t = \{H_1^t, \cdots, H_{K_t}^t\}$ and $C_t = \{C_1^t, \cdots, C_{K_t}^t\}$ to denote the ground-truth and partitioning result, where $H_r^t \subseteq \mathcal{V}_t$ and $C_r^t \subseteq \mathcal{V}_t$ represent node member sets of the $r$-th cluster. Given $H_t$ and $C_t$, the definition of NMI is as follow:

$$\text{NMI}(H_t, C_t) = \frac{-2\sum_{H_r^t \in H_t}\sum_{C_s^t \in C_t}\frac{n_{r,s}}{n}\log\frac{n \times n_{r,s}}{n_r \times n_s}}{\sum_{H_r^t \in H_t}\frac{n_r}{n}\log\frac{n_r}{n} + \sum_{C_s^t \in C_t}\frac{n_s}{n}\log\frac{n_s}{n}}, \tag{13}$$

where $n_r = |H_r^t|$ is the number of nodes in the $r$-th cluster of ground-truth; $n_s = |C_s^t|$ is the number of nodes in the $s$-th cluster of partitioning result; $n_{r,s} = |H_r^t \cap C_s^t|$ is the number of nodes that are simultaneously partitioned into the $r$-th cluster of ground-truth and the $s$-th cluster of partitioning result. Let $L_t = (l_1^t, \cdots, l_{N_t}^t)$ and $R_t = (r_1^t, \cdots, r_{N_t}^t)$ be label sequences of the partitioning result and ground-truth with respect to node sequence $(v_1^t, \cdots, v_{N_t}^t)$, where $l_i^t$ and $r_i^t$ are cluster labels of node $v_i^t$. The definition of AC is as follow:

$$\text{AC}(R_t, L_t) = \frac{\sum_{i=1}^{N_t}\delta(r_i^t, \text{map}(l_i^t))}{N_t}, \tag{14}$$

where $\text{map}(\cdot)$ is the Kuhn-Munkres mapping function that gives the best membership map from $L_t$ to $R_t$; $\delta(a,b)$ is the Kronecker delta function with $\delta(a,b) = 1$ when $a = b$ and $\delta(a,b) = 0$ otherwise. Usually, larger modularity, NMI, and AC, as well as smaller NCut indicate better partitioning quality of a given GP method.

### C.2  DATASET DETAILS

The detailed statistics of the 11 datasets (with 7 synthetic benchmarks and 4 real datasets) used in our experiments are depicted in Table 3, where $T$, $N$, $|\mathcal{E}|$, and $K$ are the number of snapshots, nodes, edges, and clusters, respectively.

*GN-Net* (Girvan & Newman, 2002) is a widely-used synthetic benchmark for GP, in which the graph topology and GP ground-truth of each graph can be generated via the stochastic block model (Abbe, 2017). We fixed $N = 5,000$ and evenly partitioned the nodes into $K = 250$ clusters. For two uniformly selected nodes $v_i$ and $v_j$, if they're assigned in the same cluster (i.e., $v_i, v_j \in C_r$), the probability to generate an edge $(v_i, v_j)$ is $p_{\text{in}}$. Otherwise (i.e., $v_i \in C_r$ and $v_j \in C_s$ ($r \neq s$)), the probability to generate an edge $(v_i, v_j)$ is $(1 - p_{\text{in}})/(K - 1)$. In particular, we set $p_{in} \in \{0.5, 0.4, 0.3\}$ to generate 3 datasets. For each dataset, we independently generated 2,000 graphs. In each graph, we also randomly shuffled the node indices to ensure that there is no index correspondence between any two graphs in the dataset (as we defined in Section 2). Especially, *with the decrease of $p_{\text{in}}$, the clustering structures become increasingly difficult to identify*. For simplicity, we denote the *GN-Net* with a specific setting of $p_{\text{in}}$ as *GN-$p_{\text{in}}$* (e.g., *GN-0.5*, *GN-0.4*, and *GN-0.3*). Note that all the datasets of *GN-Net* are with fixed $N$ and $K$.

*LFR-Net* (Lancichinetti et al., 2008) is a more challenging synthetic benchmark that can simulate power-law properties of node degree and cluster size of real-world network systems. It uses $(k, k_{\max}, c_{\min}, c_{\max}, \mu)$ to independently generate a single graph, where $k$ and $k_{\max}$ are the average and maximum node degree; $c_{\min}$ and $c_{\max}$ are the minimum and maximum cluster size; $\mu$ is

the mixing ratio between the external degree and total degree of each node $v_i$ with respect to the cluster $v_i$ belongs to. In particular, we fixed $(k, k_{\max}, c_{\min}, c_{\max}) = (10, 100, 10, 200)$ and set $\mu \in \{0.3, 0.6\}$. *With the increase of $\mu$, the clustering structure is increasingly difficult to identify.* We consider both cases with fixed and non-fixed $N$, where we respectively fixed $N = 5,000$ and set $N$ to a random integer within $[5000, 6000]$ to generate $2,000$ graphs for each case. Similar to *GN-Net*, we also independently generated each graph in a dataset and ensure that there is no index correspondence between any two graphs via the random shuffle of node indices. For simplicity, we use $L(c, \mu)$ ($c \in \{f, n\}$) to represent the dataset with a specific setting of $\mu$, where $f$ and $n$ denote the case with fixed and non-fixed $N$ (e.g., $L(f, 0.3)$, $L(f, 0.6)$, $L(n, 0.3)$, and $L(n, 0.6)$). Moreover, $K$ is non-fixed in *LFR-Net*.

*Taxi*[1] (Piorkowski et al., 2009) is a real GPS dataset including trajectories of $1,279$ taxis in Beijing, which forms a mobile ad hoc network system. We sampled $3,000$ graph snapshots from the dataset in chronological order. In each graph, we treated each taxi as a node and constructed the topology based on the top-50 neighbors with the closest distance for each node. Since *Taxi* does not provide ground-truth regarding $K$ for GP, we used *Louvain* algorithm (Blondel et al., 2008) to estimate the $K$ value for each graph in this dataset. Hence, it is with fixed $N$ and non-fixed $K$.

*Reddit*[2] (Yanardag & Vishwanathan, 2015) is a social network dataset extracted from Reddit. Each graph snapshot in the dataset corresponds to an online discussion thread. We extracted $1,870$ graphs with $N \geq 500$ from the dataset. Different graphs are with different number of users (i.e., nodes), varying from $501$ to $3,648$. The comment interactions between users in each discussion thread are described as topology of the corresponding graph. Since *Reddit* does not provide ground-truth regarding $K$, we also used *Louvain* to estimate $K$ for each graph. Hence, *Reddit* is with non-fixed $N$ and $K$. Note that different snapshots have different node sets (i.e., with non-fixed $N$) and node indices in each snapshot start from 0. It indicates that there is no correspondence between node indices of any two snapshots in the dataset.

*Enron*[3] (Rossi & Ahmed, 2015) is a real interaction network extracted from the email system of Enron company, which contains email records from 1980-01-01 to 2004-02-04. We extracted 410 snapshots with $N \geq 500$ from the records in chronological order, in which we treated each user as a unique node and constructed the topology based on email interactions. Note that different graphs have different sets of users that have interaction behaviors. Moreover, we also used *Louvain* algorithm to estimate $K$ for each graph, since *Enron* does not provide ground-truth regarding $K$. Therefore, it is with non-fixed $N$ and $K$. We also ensure that there is no node index correspondence between any two graphs by permuting node indices of each graph from 0.

*AS*[4] (Leskovec et al., 2005) is a real dataset extracted from the border gateway protocol (BGP) logs of a communication network, which describe the topology among a set of autonomous systems (AS). It contains 733 daily instances spanning from 1997-08-11 to 2000-01-02. Each BGP router is abstracted as a unique node with communication between routers extracted as the graph topology. As the dataset allows addition and deletion of routers, different graphs are with different set of nodes. Since *AS* does not provide the ground-truth $K$, we also applied *Louvain* to determine $K$ for each snapshot. Therefore, it is with non-fixed $N$ and $K$, where there is no index correspondence between any two graphs.

In summary, the 3 datasets of *GN-Net* are with fixed $N$ and $K$. The 4 datasets of *LFR-Net* cover both the cases of (i) fixed $N$ but non-fixed $K$ as well as (ii) non-fixed $N$ and $K$. *Taxi* is a real dataset with fixed $N$ but non-fixed $K$, while *Reddit*, *Enron*, and *AS* are with non-fixed $N$ and $K$. Note that both *GN-Net* and *LFR-Net* are synthetic benchmarks that can simultaneously generate graph topology and GP ground-truth of each snapshot. *Taxi*, *Reddit*, *Enron*, and *AS* are real datasets without ground-truth (regarding $K$), so we applied *Louvain* algorithm to these datasets to determine the $K$ value of each single snapshot, which ensures the fairness of comparison, i.e., each snapshot shoule be assigned with a common $K$ for all the methods to be evaluated.

---

[1]https://www.microsoft.com/en-us/research/publication/t-drive-driving-directions-based-on-taxi-trajectories/

[2]https://github.com/weihua916/powerful-gnns/blob/master/dataset.zip

[3]http://networkrepository.com/ia-enron-email-dynamic.php

[4]http://snap.stanford.edu/data/as-733.html

## C.3 BASELINE METHODS

In addition to the 15 baseline methods introduced in Section 4, we also evaluated partitioning quality and runtime of other 5 strong baselines, which include degree-corrected stochastic block model (*DCSBM*), *IncSNA* (Su et al., 2020), (Karrer & Newman, 2011), *DeepWalk* (Perozzi et al., 2014), *node2vec* (Grover & Leskovec, 2016), *LINE* (Tang et al., 2015), *TIMERS* (Zhang et al., 2018b) and *GIN* (Xu et al., 2019). In summary, we have in total evaluated the quality and efficiency of 22 baselines with 4 categories.

- First, *SNMF*, *DCSBM*, and *SC* are classic *transductive* GP methods based on several GP objectives (e.g., *SC* for NCut minimization and *DCSBM* for modularity maximization), which aim to achieve high-quality partitioning results (but may have high complexity). Moreover, *IncSNA* is an incrmental GP baselines that can utilize the node correspondence between graphs to incrementally update the GP results for multiple associated graphs.

- Second, *DeepWalk*, *node2vec*, *LINE*, *VERSE*, *MILE*, *PhUSION*, *NPR*, *RandNE*, and *ProNE* are unsupervised *transductive* graph embedding baselines. In particular, *LINE*, *NPR*, *RandNE*, and *ProNE* are efficient embedding methods with several fast approximation strategies for the embedding learning (but may have low partitioning quality). In contrast, *DeepWalk*, *node2vec*, *VERSE*, *MILE*, and *PhUSION* are baselines that aim to derive high-quality representations (but may have high complexity) Besides, *TIMERS* is a typical incremental graph embedding method that can incrementally update the learned embedding for graphs with node correspondence.

- Third, *Metis*, *hMetis*, and *GraClus* are classic fast *transductive* GP baselines that aim to achieve low runtime with the heuristic approximation of classic GP objectives (which may suffer from quality degradation due to the information loss of heuristic approximation).

- Fourth, *GraSAGE*, *GAT*, and *GIN*, *GAP*, and *ClusNet* are *inductive* GNN baselines. In particular, *GraSAGE*, *GAT*, and *GIN* are SOTA *inductive* GNN following the graph embedding scheme, i.e., learning node representations to support downstream tasks including GP. Note that we do not consider supervised training loss of these *inductive* GNNs (e.g., using the cross-entropy loss for a softmax output layer combined with the GNN), because GP is a typical unsupervised task and the cluster labels are permutation invariant. In contrast, *GAP* and *ClusNet* are E2E GP baselines based on *inductive* nature of GNN, where an FC output layer is integrated into the GNN to directly output the partitioning results.

Note that the first three types of methods are inherently *transductive*, which only focus on GP on a single static graph but cannot be generalized to other new graphs. Hence, we had to apply these *transductive* baselines to each newly generated system snapshot (i.e., each snapshot in test set $\Gamma'$) from scratch. For the fourth type of baselines (i.e., *inductive* GNN approaches), we used the same settings with IGP (i.e., *offline* training on training set $\Gamma_T$ and *online* generalization to test set $\Gamma'$). In particular, we can only use the *inductive* nature of E2E GNN methods (i.e., *GAP* and *ClusNet*) on datasets with fixed number of clusters $K$ (i.e., on *GN-Net*). However, we have to use the same settings with *transductive* baselines (i.e., training *GAP* and *ClusNet* from scratch for each new test graph) on rest datasets with non-fixed $K$, due to the fixed dimensionality of their output layers. Note that not all the datasets have node correspondence between graphs. The incremental GP and graph embedding baselines (i.e., *IncNSA* and *TIMERS*) can only be applied to datasets with node correspondence. To include *IncNSA* and *TIMERS* in our evaluation, we applied them to the original version of *Taxi* and *AS* that have node correspondence. Some other experiment settings regarding the baseline methods have been already introduced in Section 4.

## C.4 IMPLEMENTATION DETAILS

We used PyTorch to implement the dual GNN structure of IGP. The feature extraction module (i.e., Algorithm 1 and Algorithm 2) was implemented via NumPy and PyTorch, while we used the implementation of $K$Means of scikit-learn as the downstream clustering module. All the experiments were conducted on a server with Intel Xeon CPU (E5-2650v4@2.20GHz, 48 cores), 1 Tesla V100 GPU, 61GB memory, as well as the Ubuntu Linux operating system. In this setting, modules implemented by PyTorch were speeded up via GPU.

Table 4: Parameter settings and layer configurations of IGP.

| | IGP-M | | IGP-C | | Layer Configurations | |
|---|---|---|---|---|---|---|
| | $(\alpha, \beta, m, p)$ | $(\eta_G, \eta_D)$ | $(\alpha, \beta, m, p)$ | $(\eta_G, \eta_D)$ | $G$ | $D$ |
| *GN*-0.5 | (1,1,1,1000) | (5e-4,5e-4) | (1,1,1,1000) | (1e-4,1e-4) | | |
| *GN*-0.4 | (1,1,1,1000) | (5e-4,5e-4) | (1,1,1,1000) | (1e-4,1e-4) | 2048-1024-512 | 512-128-64-16-1 |
| *GN*-0.3 | (1,5,1,1000) | (5e-4,5e-4) | (1,1,1,1000) | (1e-4,1e-4) | | |
| *L(f,*0.3) | (1,1,1,1000) | (5e-4,5e-4) | (1,1e2,5,1000) | (1e-4,1e-4) | 4096-2048-512-256 | 256-128-64-16-1 |
| *L(f,*0.6) | (1,3,1,1000) | (1e-4,1e-4) | (1,1e3,2,1000) | (5e-5,5e-5) | | |
| *L(n,*0.3) | (1,1,1,1000) | (5e-4,5e-4) | (1,1e2,5,1000) | (1e-4,1e-4) | 4096-2048-512-256 | 256-128-64-16-1 |
| *L(n,*0.6) | (1,3,1,1000) | (1e-4,1e-4) | (1,1e3,2,1000) | (5e-5,5e-5) | | |
| *Taxi* | (1,0.1,1,2000) | (5e-5,5e-5) | (1,1e2,1,2000) | (5e-5,5e-5) | 1024-512-256 | 256-64-32-16-1 |
| *Reddit* | (1,0.1,1,1000) | (5e-5,5e-5) | (1,0.1,1,1000) | (5e-4,5e-4) | 2048-1024-512-256 | 256-64-16-1 |
| *AS* | (1,0.1,1,500) | (5e-5,5e-5) | (1,1e2,1,500) | (5e-5,5e-5) | 6000-4096-2048-1024-256 | 256-128-64-16-1 |
| *Enron* | (1,5,1,300) | (5e-5,5e-5) | (1,1e2,1,300) | (5e-5,5e-5) | 1024-512-256 | 256-128-64-16-1 |

For all the embedding based methods, including *transductive* embedding baselines (i.e., second type of the baseline), *inductive* GNNs (i.e., *GraSAGE*, *GAT*, and *GIN*), and IGP, we used $K$Means as the downstream clustering module. In contrast, we can directly obtain the partitioning results from outputs of the rest baselines. Note that $K$Means for graph embedding based methods and optimization algorithms of several classic GP baselines (i.e., *SNMF*, *DCSBM*, *Metis*, *hMetis*, and *GraClus*) have the limitation of obtaining locally optimal solutions. We adopted the strategy to run $K$Means of embedding based methods and optimization algorithms of classic baselines 10 times, which was included in the total runtime of corresponding methods. The partitioning membership with minimum loss function value (e.g., the sum of squared distance between data and cluster center for $K$Means) was reported as the final partitioning result for evaluation. When testing the runtime of each method, we ensured that only the method to be evaluated was running on the experiment server, i.e., there were no multiple methods simultaneously running on the test server.

On the validation set $\Gamma_V$ of each dataset, we tuned parameters and determine proper layer configurations for the two variants of IGP. Concretely, we adjusted $\alpha \in \{1, 10\}$ and $m \in \{1, 2, \cdots 5\}$ for both IGP-M and IGP-C, while we tuned $\beta \in \{0.1, 1, 2, \cdots 5\}$ and $\beta \in \{0.1, 1, 100, 1000\}$ for IGP-M and IGP-C, respectively. The recommended parameter settings and layer configurations of IGP-M and IGP-C on each dataset are depicted in Table 4, where dimensionality of the first and last layer of $G$ are set to be $L$ (i.e., dimensionality of reduced feature) and $k$ (i.e., dimensionality of graph embedding), respectively. Note that we use the same layer configurations for IGP-M and IGP-C on each dataset. Moreover, we set the maximum number of epochs $n = 100$ for all the datasets.

## C.5 DETAILED EVALUATION RESULTS AND DISCUSSIONS

### C.5.1 QUANTITATIVE EVALUATION

In addition to the visualization results of **NMI**, **Modularity**, and **runtime** in Fig. 3, we also visualized the evaluation results in terms of average **AC**, **NCut**, and **runtime** on test sets of the 11 datasets in Fig. 4, where one can have the observation consistent with Fig. 3. In most cases, both IGP-M and IGP-C are in the top groups with best quality metrics. The total runtime of IGP-M and IGP-C is also competitive to the fast *transductive* GP baselines.

In the experiments, we recorded the mean value $\mu$ and standard deviation $\sigma$ of all the evaluation metrics and runtime for all the methods (on test set $\Gamma'$ of each dataset). In particular, we use format '$\mu$ $(\sigma)$' to represent each evaluation result. The quantitative evaluation results on *GN-Net* and *LFR-Net* in terms of NMI and AC are depicted in Table 5 and Table 6. The evaluation results on all the datasets in terms of modularity and NCut are illustrated in Table 7, Table 8, Table 9, and Table 10. Moreover, Table 11 and Table 12 give the runtime of all the methods. For both IGP-M and IGP-C, we also recorded the time of (i) feature extraction (i.e., Algorithm 2), (ii) one feedforward propagation (through the *feature encoder* of $G$), and (iii) downstream clustering (i.e., 10 independent runs of $K$Means), which together contribute to the total runtime of IGP. Runtime of the three modules is denoted as 'Feat', 'Prop', and 'Clus', respectively.

According to evaluation results in Table 5, Table 6, Table 7, Table 8, Table 9, Table 10 Table 11, and Table 12, which are consistent with the visualization results in shown Fig. 3 (see Section 4), we have the following major observations.

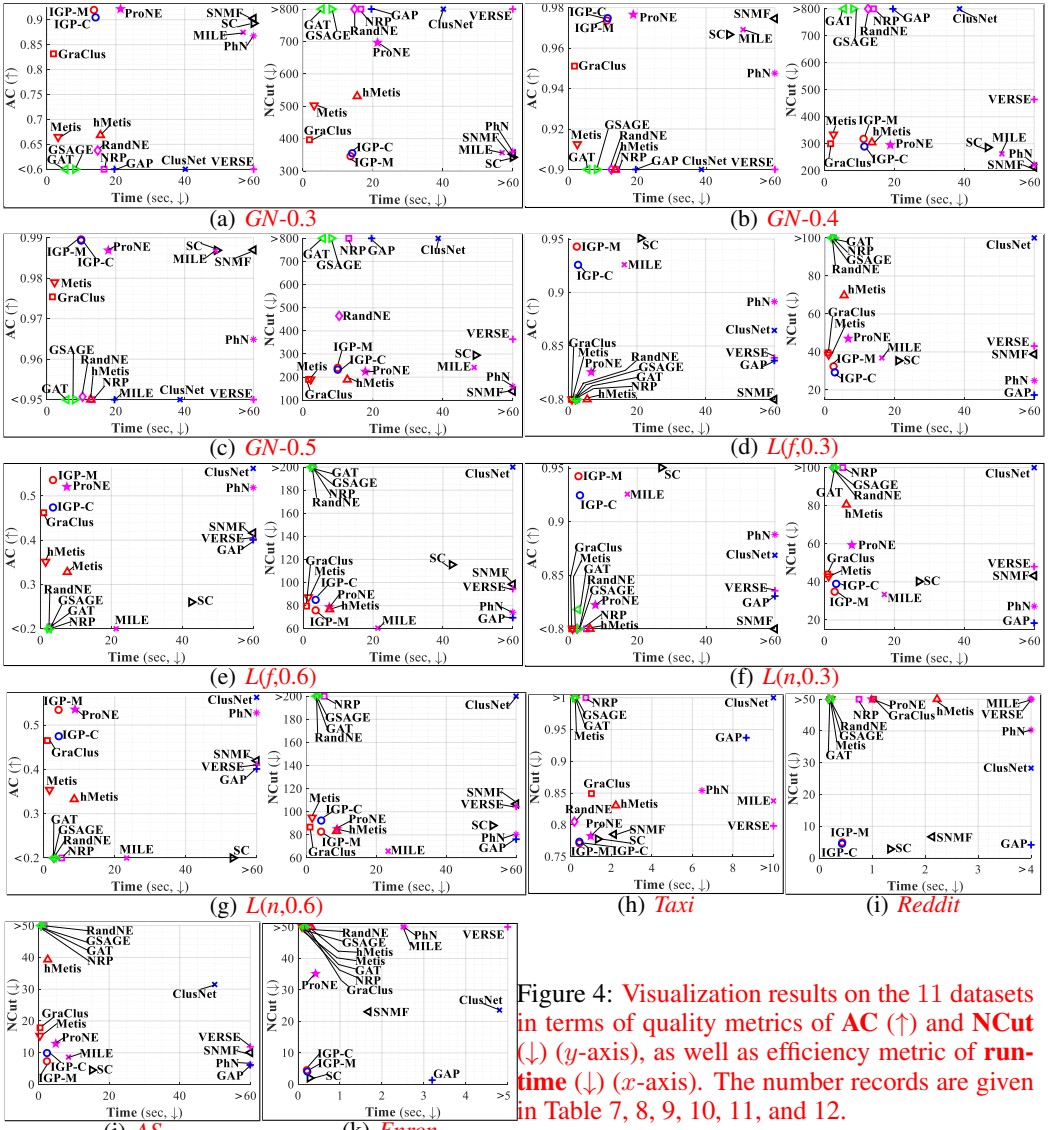

Figure 4: Visualization results on the 11 datasets in terms of quality metrics of **AC** (↑) and **NCut** (↓) (*y*-axis), as well as efficiency metric of **run-time** (↓) (*x*-axis). The number records are given in Table 7, 8, 9, 10, 11, and 12.

- For all the datasets, IGP-M and IGP-C are in the top group with best quality metrics. More-over, their runtime is also competitive to fast GP baselines (e.g., *Metis*, *hMetis*, *GraClus*, *RandNE*, etc.) and other *inductive* GNNs with fast *online* generalization (e.g., *GraSAGE*, *GAT*, and *GIN*). From the perspective of alleviating the NP-hard challenge of GP, IGP can achieve a significant trade-off between quality and efficiency of *online* GP on newly gener-ated system snapshots. We further quantitatively evaluate the significant trade-off obtained by IGP via a novel trade-off score in Appendix C.5.2.

- Surprisingly, in some cases, IGP even has the best or second-best quality metrics. Note that IGP is a novel *inductive* framework, in which we do not conduct additional optimization on the test set of each dataset, but other conventional *transductive* baselines (e.g., *SP*, *Deep-Walk*, etc.) can fully explore the structural properties of each test graph. It indicates the potential of IGP to capture additional characteristics shared by multiple historical snapshots beyond conventional *transductive* approaches applied to one single new graph.

- On *GN-Net* and *LFR-Net*, partitioning quality of all the methods declines with the decrease of $p_{in}$ and increase of $\mu$, where both IGP-M and IGP-C are always in the top group with best quality metrics. It demonstrates the strong robustness of IGP, especially when the intrinsic clustering structure of the given graph is not distinct.

Table 5: Quantitative evaluation results in terms of NMI (%) (↑).

| | GN-0.5 | GN-0.4 | GN-0.3 | L(f,0.3) | L(f,0.6) | L(n,0.3) | L(n,0.6) |
|---|---|---|---|---|---|---|---|
| SNMF | 99.53 (0.21) | 98.79 (0.36) | 94.37 (0.69) | 83.53 (1.60) | 49.15 (2.56) | 83.59 (1.59) | 49.98 (2.19) |
| DCSBM | 67.92 (10.49) | 57.68 (10.42) | 49.76 (6.04) | 45.64 (6.25) | 27.27 (2.89) | 44.73 (7.48) | 27.32 (2.42) |
| SC | 99.75 (0.10) | 98.73 (0.24) | 94.26 (0.41) | 98.19 (0.39) | 35.78 (3.65) | 98.21 (0.35) | 24.66 (2.76) |
| DeepWalk | 99.40 (0.11) | 97.60 (0.30) | 82.81 (0.85) | 93.27 (0.77) | 55.71 (2.19) | 93.36 (0.73) | 56.40 (2.03) |
| node2vec | 99.44 (0.09) | 98.89 (0.19) | 88.26 (0.77) | 95.15 (0.73) | 59.11 (2.21) | 95.22 (0.68) | 59.88 (2.07) |
| LINE | 93.73 (0.54) | 68.70 (0.75) | 53.25 (0.31) | 80.72 (1.44) | 42.69 (2.59) | 79.89 (1.56) | 38.78 (3.29) |
| VERSE | 92.09 (0.99) | 84.75 (1.57) | 61.15 (1.50) | 89.80 (0.89) | 44.97 (3.16) | 89.94 (0.97) | 45.87 (3.05) |
| MILE | 99.79 (0.04) | 99.17 (0.13) | 93.39 (0.48) | 93.87 (0.95) | 20.37 (2.32) | 94.11 (0.97) | 20.67 (2.04) |
| PhUSION | 99.46 (0.08) | 98.81 (0.20) | 93.89 (0.57) | 94.85 (0.75) | 54.18 (2.64) | 94.90 (0.72) | 55.52 (2.38) |
| NRP | 94.98 (0.78) | 90.73 (0.86) | 77.90 (1.24) | 55.78 (3.02) | 26.60 (2.39) | 51.90 (3.05) | 22.26 (2.52) |
| RandNE | 98.33 (0.26) | 94.08 (0.47) | 79.18 (1.01) | 57.63 (2.06) | 18.57 (2.16) | 58.08 (2.08) | 18.87 (1.95) |
| ProNE | 99.80 (0.05) | 99.54 (0.08) | 96.62 (0.36) | 93.33 (0.81) | 55.88 (3.04) | 93.31 (0.89) | 57.61 (2.63) |
| Metis | 98.68 (0.33) | 94.68 (0.41) | 80.86 (0.77) | 84.94 (0.74) | 44.52 (2.48) | 85.43 (0.71) | 45.53 (2.28) |
| hMetis | 96.30 (0.16) | 93.82 (0.30) | 82.08 (0.68) | 88.85 (1.27) | 41.65 (3.28) | 89.13 (1.17) | 42.64 (2.99) |
| GraClus | 99.54 (0.19) | 98.10 (0.30) | 90.72 (0.76) | 90.58 (1.14) | 53.06 (3.00) | 90.60 (1.09) | 53.90 (2.75) |
| GraSAGE | 45.76 (0.13) | 45.98 (0.09) | 43.21 (0.21) | 22.52 (2.21) | 15.79 (1.89) | 24.67 (2.03) | 16.66 (1.77) |
| GAT | 46.09 (0.24) | 45.03 (0.25) | 44.55 (0.20) | 22.99 (1.88) | 15.49 (1.78) | 25.91 (1.94) | 15.83 (1.70) |
| GIN | 43.29 (0.19) | 43.26 (0.19) | 43.49 (0.17) | 18.49 (1.83) | 13.32 (1.47) | 17.61 (1.71) | 13.43 (1.22) |
| GAP | 72.69 (0.52) | 64.37 (0.39) | 56.13 (0.31) | 93.73 (1.76) | 47.30 (3.23) | 93.64 (1.65) | 48.01 (3.11) |
| ClusNet | 26.30 (1.85) | 17.65 (2.10) | 13.10 (1.44) | 89.84 (1.31) | 55.49 (2.04) | 90.10 (1.27) | 56.04 (1.78) |
| **IGP-M** | 99.85 (0.05) | 99.38 (0.11) | 96.26 (0.32) | 97.21 (0.49) | 56.07 (2.33) | 97.30 (0.44) | 56.48 (2.08) |
| **IGP-C** | 99.85 (0.04) | 99.46 (0.09) | 95.50 (0.43) | 96.44 (0.57) | 52.11 (2.79) | 96.56 (0.59) | 52.71 (2.52) |

Table 6: Quantitative evaluation results in terms of AC (%) (↑).

| | GN-0.5 | GN-0.4 | GN-0.3 | L(f,0.3) | L(f,0.6) | L(n,0.3) | L(n,0.6) |
|---|---|---|---|---|---|---|---|
| SNMF | 98.70 (0.50) | 97.46 (0.78) | 90.25 (1.32) | 73.37 (3.20) | 41.61 (2.56) | 72.96 (2.97) | 41.95 (2.40) |
| DCSBM | 38.08 (13.89) | 25.83 (9.88) | 16.40 (3.93) | 22.35 (4.39) | 13.65 (2.02) | 21.37 (4.96) | 13.04 (1.75) |
| SC | 98.69 (0.39) | 96.67 (0.54) | 89.29 (0.85) | 95.09 (0.73) | 26.01 (3.62) | 97.27 (0.96) | 12.26 (1.59) |
| DeepWalk | 96.40 (0.58) | 93.15 (0.91) | 69.03 (1.52) | 87.91 (2.01) | 54.97 (2.10) | 87.65 (2.13) | 55.12 (1.97) |
| node2vec | 96.48 (0.51) | 95.53 (0.64) | 79.30 (1.47) | 90.01 (2.10) | 58.40 (2.07) | 89.96 (2.05) | 58.79 (1.96) |
| LINE | 86.17 (1.06) | 46.90 (1.34) | 20.36 (0.66) | 72.14 (3.07) | 38.22 (2.24) | 70.55 (3.17) | 33.63 (3.50) |
| VERSE | 80.76 (1.89) | 70.02 (2.53) | 35.39 (2.16) | 83.89 (2.07) | 40.25 (3.23) | 83.56 (2.14) | 41.03 (3.11) |
| MILE | 98.67 (0.31) | 96.91 (0.50) | 87.48 (0.91) | 92.60 (1.60) | 10.19 (0.93) | 92.57 (1.55) | 9.94 (0.86) |
| PhUSION | 96.49 (0.54) | 94.76 (0.66) | 86.77 (1.11) | 89.14 (2.06) | 51.83 (2.49) | 88.80 (1.75) | 52.75 (2.25) |
| NRP | 81.61 (1.36) | 72.90 (1.26) | 56.39 (1.78) | 42.27 (2.46) | 20.42 (1.74) | 38.75 (2.64) | 16.43 (1.70) |
| RandNE | 95.07 (0.68) | 86.56 (1.03) | 63.77 (1.49) | 40.67 (1.77) | 12.91 (1.25) | 40.64 (1.73) | 12.80 (1.17) |
| ProNE | 98.69 (0.36) | 97.66 (0.42) | 92.16 (0.80) | 82.56 (2.55) | 52.04 (2.95) | 82.23 (2.52) | 53.51 (2.76) |
| Metis | 97.90 (0.62) | 91.25 (0.82) | 66.48 (1.55) | 58.70 (2.28) | 35.13 (1.77) | 58.58 (2.57) | 35.37 (1.55) |
| hMetis | 88.01 (0.44) | 84.44 (0.72) | 66.84 (1.20) | 66.62 (3.16) | 32.80 (2.91) | 65.99 (2.49) | 33.26 (2.63) |
| GraClus | 97.54 (0.84) | 95.11 (0.94) | 83.16 (1.54) | 72.00 (3.40) | 46.19 (3.02) | 71.81 (3.33) | 46.50 (2.78) |
| GraSAGE | 7.83 (0.13) | 7.66 (0.09) | 7.44 (0.12) | 7.55 (0.25) | 5.79 (0.15) | 7.58 (0.34) | 5.66 (0.19) |
| GAT | 9.20 (0.17) | 8.66 (0.16) | 8.00 (0.13) | 8.52 (0.36) | 6.22 (0.16) | 8.78 (0.40) | 5.69 (0.19) |
| GIN | 7.61 (0.12) | 7.51 (0.11) | 7.46 (0.11) | 6.97 (0.23) | 5.83 (0.15) | 6.08 (0.30) | 5.37 (0.21) |
| GAP | 38.90 (1.04) | 29.71 (0.76) | 20.30 (0.57) | 83.63 (4.16) | 40.06 (3.34) | 83.06 (3.92) | 40.08 (3.41) |
| ClusNet | 4.51 (0.30) | 2.94 (0.28) | 3.13 (0.23) | 86.45 (2.47) | 56.20 (1.82) | 86.87 (2.36) | 56.25 (1.65) |
| **IGP-M** | 98.96 (0.28) | 97.36 (0.46) | 91.96 (0.73) | 94.29 (1.41) | 53.57 (2.21) | 94.24 (1.38) | 53.42 (2.04) |
| **IGP-C** | 98.93 (0.31) | 97.48 (0.46) | 90.47 (0.85) | 92.58 (1.49) | 47.38 (2.68) | 92.46 (1.46) | 47.47 (2.59) |

- Similar to IGP, the E2E GNN baselines (i.e., *GAP* and *ClusNet*) can also apply fast *online* generalization to datasets with fixed $K$ (i.e., on *GN-Net*). Namely, we also trained *GAP* and *ClusNet* on training set of *GN-Net* and directly generalized them to corresponding test sets for *online* GP, in which we directly derived the partitioning results from their output layers. However, the *online* GP of *GAP* and *ClusNet* is with poor quality, which indicates that the *online* generalization of the graph embedding scheme in IGP is more robust than that of the E2E approaches, especially when clustering structure of a dataset is not distinct. Furthermore, we still need to train *GAP* and *ClusNet* from scratch for each new test graph on datasets with non-fixed $K$ due to the fixed dimensionality of their output layers, which is time-consuming. In contrast, the graph embedding scheme of IGP can still tackle fast *online* GP with non-fixed $K$.

- In contrast to the reduced structural feature $\mathbf{Z}_t$ in IGP, we used a constant matrix $\mathbf{1}_{N_t \times L}$ as feature inputs of inductive GNN baselines (i.e., *GraSAGE*, *GAT*, adn *GIN*), which is a standard setting of most *inductive* GNNs for graphs without available node attributes (Xu et al., 2019). The widely-used unsupervised training loss of *GraSAGE* (Hamilton et al., 2017) was also adopted in the *offline* training of these inductive GNNs. Although the inductive GNN baselines have slightly faster runtime over IGP, they still suffer from poor partitioning quality on all the datasets. The results further validate that the extracted

Table 7: Quantitative evaluation results in terms of Modularity (%) (↑) on datasets with ground-truth.

| | GN-0.5 | GN-0.4 | GN-0.3 | L(f,0.3) | L(f,0.6) | L(n,0.3) | L(n,0.6) |
|---|---|---|---|---|---|---|---|
| SNMF | 47.95 (0.23) | 37.79 (0.27) | 27.28 (0.29) | 56.12 (1.58) | 29.22 (0.63) | 56.03 (1.50) | 29.43 (0.60) |
| DCSBM | 17.31 (5.55) | 9.33 (2.40) | 3.86 (0.72) | 11.37 (2.46) | 4.77 (0.96) | 11.00 (2.68) | 4.71 (0.74) |
| SC | 48.13 (0.20) | 37.81 (0.23) | 27.07 (0.23) | 67.32 (0.20) | 26.64 (1.06) | 67.50 (0.26) | 23.62 (2.83) |
| DeepWalk | 47.88 (0.20) | 37.27 (0.28) | 21.97 (0.46) | 62.31 (1.11) | 33.92 (0.54) | 62.35 (1.14) | 34.06 (0.48) |
| node2vec | 47.89 (0.19) | 37.91 (0.21) | 24.62 (0.45) | 63.23 (1.14) | 34.47 (0.54) | 63.30 (1.14) | 34.64 (0.47) |
| LINE | 42.47 (0.59) | 15.96 (0.61) | 5.14 (0.17) | 55.88 (1.84) | 23.34 (0.98) | 55.21 (1.74) | 20.76 (1.93) |
| VERSE | 41.33 (0.91) | 28.62 (1.05) | 12.79 (0.69) | 60.50 (1.18) | 29.34 (1.13) | 60.52 (1.19) | 29.40 (1.08) |
| MILE | 48.21 (0.18) | 38.08 (0.20) | 26.96 (0.27) | 66.37 (0.34) | 21.33 (1.25) | 66.55 (0.42) | 21.59 (1.14) |
| PhUSION | 47.89 (0.20) | 37.79 (0.22) | 27.03 (0.29) | 63.25 (1.00) | 33.80 (0.50) | 63.26 (0.91) | 34.06 (0.48) |
| NRP | 45.21 (0.41) | 34.31 (0.39) | 22.27 (0.46) | 24.32 (1.29) | 7.07 (0.72) | 25.00 (1.79) | 5.69 (0.80) |
| RandNE | 47.29 (0.24) | 35.61 (0.34) | 21.08 (0.47) | 18.44 (1.20) | 3.26 (0.34) | 18.48 (1.21) | 3.38 (0.32) |
| ProNE | 48.20 (0.18) | 38.20 (0.19) | 27.92 (0.24) | 60.96 (1.19) | 32.64 (0.81) | 60.98 (1.17) | 33.24 (0.70) |
| Metis | 47.29 (0.34) | 35.24 (0.30) | 21.62 (0.36) | 52.38 (1.12) | 30.51 (0.44) | 52.41 (1.17) | 30.58 (0.39) |
| hMetis | 44.78 (0.22) | 34.62 (0.28) | 22.39 (0.31) | 57.51 (1.14) | 31.80 (0.60) | 57.44 (0.96) | 31.99 (0.49) |
| GraClus | 47.83 (0.26) | 37.34 (0.30) | 25.00 (0.48) | 53.54 (1.98) | 31.61 (0.92) | 53.43 (1.87) | 31.71 (0.83) |
| GraSAGE | 0.08 (0.03) | 0.04 (0.03) | 0.01 (0.01) | 0.28 (0.17) | 0.18 (0.11) | 0.56 (0.27) | 0.17 (0.16) |
| GAT | 0.48 (0.05) | 0.27 (0.05) | 0.08 (0.03) | 0.99 (0.17) | 0.08 (0.12) | 1.71 (0.22) | 0.02 (0.11) |
| GIN | 0.05 (0.03) | 0.03 (0.03) | 0.01 (0.03) | 0.50 (0.11) | 0.06 (0.08) | 4.73 (0.29) | 4.19 (0.18) |
| GAP | 19.35 (0.63) | 9.95 (0.36) | 5.24 (0.19) | 66.45 (0.59) | 32.96 (0.73) | 66.65 (0.57) | 32.96 (0.77) |
| ClusNet | 0.03 (0.10) | 0.02 (0.12) | 0.01 (0.13) | 65.52 (0.53) | 35.10 (0.51) | 65.77 (0.54) | 35.25 (0.44) |
| **IGP-M** | 48.25 (0.18) | 38.17 (00.20) | 27.92 (0.22) | 66.26 (0.54) | 34.14 (0.53) | 66.45 (0.56) | 34.22 (0.47) |
| **IGP-C** | 48.25 (0.18) | 38.19 (00.19) | 27.68 (0.25) | 65.72 (0.59) | 31.81 (0.65) | 65.83 (0.64) | 31.95 (0.62) |

Table 8: Quantitative evaluation results in terms of Modularity (%) (↑) on datasets w/o ground-truth.

| | Taxi | Reddit | AS | Enron |
|---|---|---|---|---|
| SNMF | 73.72 (0.41) | 68.29 (7.16) | 53.32 (2.55) | 73.88 (14.55) |
| DCSBM | 73.71 (0.60) | 2.81 (5.17) | 4.97 (2.16) | 4.36 (7.09) |
| SC | 73.80 (0.59) | 61.19 (7.68) | 43.14 (6.60) | 71.43 (19.05) |
| IncSNA | 71.96 (0.79) | - | 4.13 (1.13) | - |
| DeepWalk | 73.72 (0.54) | 65.66 (8.91) | 56.48 (2.90) | 74.72 (17.57) |
| node2vec | 73.64 (0.54) | 69.31 (7.90) | 58.19 (2.80) | 76.86 (13.31) |
| LINE | 73.95 (0.54) | 57.25 (12.92) | 45.02 (4.87) | 70.94 (14.86) |
| VERSE | 73.69 (0.51) | 51.70 (13.77) | 52.25 (10.59) | 73.43 (18.95) |
| MILE | 72.91 (1.21) | 59.09 (17.94) | 56.83 (3.20) | 74.78 (18.98) |
| PhUSION | 72.93 (0.59) | 67.69 (6.18) | 58.18 (2.78) | 67.84 (12.37) |
| NRP | 6.07 (9.55) | 4.01 (5.99) | 4.61 (3.39) | 0.24 (6.53) |
| RandNE | 73.12 (0.72) | 5.52 (5.04) | 5.30 (0.54) | 21.53 (12.32) |
| ProNE | 73.61 (0.64) | 62.88 (5.27) | 49.10 (3.87) | 62.80 (9.98) |
| TIMERS | 30.80 (2.56) | - | 4.89 (1.29) | - |
| Metis | 70.48 (0.51) | 52.34 (10.68) | 52.75 (3.86) | 65.04 (17.07) |
| hMetis | 73.44 (0.80) | 54.27 (11.02) | 56.03 (4.33) | 67.54 (18.16) |
| GraClus | 73.05 (0.64) | 45.27 (11.59) | 46.81 (4.83) | 62.02 (21.93) |
| GraSAGE | 21.85 (1.11) | 6.94 (3.72) | 5.69 (1.47) | 9.41 (6.55) |
| GAT | 1.11 (0.42) | 8.69 (11.82) | 4.60 (0.84) | 4.20 (4.89) |
| GIN | 8.09 (0.44) | 4.82 (7.30) | 5.62 (7.60) | 10.82 (3.61) |
| GAP | 71.79 (1.30) | 65.08 (6.24) | 58.31 (2.68) | 75.47 (12.57) |
| ClusNet | 72.54 (1.27) | 55.31 (10.37) | 55.87 (2.42) | 72.56 (18.75) |
| **IGP-M** | 74.01 (0.51) | 69.38 (5.49) | 56.82 (2.71) | 74.33 (13.51) |
| **IGP-C** | 74.00 (0.42) | 69.37 (5.47) | 55.40 (2.26) | 74.15 (11.78) |

feature $\mathbf{Z}_t$ as well as hybrid training loss of IGP (which can incorporate the permutation invariant label information of historical training snapshots) are essential to ensure its high partitioning quality.

- For the total runtime of IGP on all the datasets, the downstream clustering (e.g., $K$Means with 10 independent runs) is a major bottleneck. However, the downstream clustering module is also the key component that enables IGP to deal with *online* GP with non-fixed $K$. In our future work, we intend to further reduce the runtime of IGP by replacing the downstream clustering module with a generic E2E module that can directly derive the partitioning result with respect to a specific (non-fixed) $K$ via an output layer.

- In most cases, IGP-M and IGP-C have similar quality metrics. On datasets in which the clustering structures are more difficult to identify (e.g., $L(f, 0.6)$ and $L(n, 0.6)$), IGP-M has better partitioning quality than that of IGP-C. It implies that the modularity matrix $\mathbf{Q}_t$ is more informative than the normalized adjacency matrix $\mathbf{M}_t$ to be the neighbor similarity feature $\mathbf{X}_t$ in IGP. Moreover, the neighbor-induced feature $\mathbf{X}_t$ is also the key component to ensure the strong robustness of IGP, especially when clustering structures of a given graph are indistinct.

Table 9: Quantitative evaluation results in terms of NCut (↓) on datasets with ground-truth.

| | GN-0.5 | GN-0.4 | GN-0.3 | L(f,0.3) | L(f,0.6) | L(n,0.3) | L(n,0.6) |
|---|---|---|---|---|---|---|---|
| SNMF | 139.14 (10.90) | 213.23 (14.28) | 352.04 (8.94) | 38.80 (5.62) | 98.49 (10.95) | 43.22 (6.33) | 106.72 (11.89) |
| DCSBM | 3.74e3 (2.33e3) | 7.64e3 (4.37e3) | 1.73e4 (6.21e3) | 3.26e3 (1.72e3) | 2.25e3 (1.63e3) | 4.10e3 (2.14e3) | 2.51e3 (1.70e3) |
| SC | 294.67 (388.31) | 286.04 (118.40) | 342.36 (34.29) | 35.38 (35.34) | 115.48 (29.30) | 40.20 (45.35) | 88.11 (43.17) |
| DeepWalk | 283.44 (114.49) | 488.62 (202.14) | 1.05e3 (298.59) | 36.29 (39.00) | 80.17 (16.20) | 42.39 (50.88) | 89.76 (35.83) |
| node2vec | 224.85 (84.62) | 346.07 (142.89) | 664.81 (193.29) | 55.11 (94.33) | 80.50 (37.93) | 50.57 (63.56) | 87.03 (34.39) |
| LINE | 374.65 (251.98) | 1.25e3 (331.83) | 3.04e3 (532.29) | 2.56e3 (1.21e3) | 2.20e3 (1.30e3) | 2.83e3 (1.33e3) | 1.93e3 (1.12e3) |
| VERSE | 363.71 (97.62) | 463.89 (64.89) | 939.25 (56.17) | 43.07 (44.96) | 94.54 (11.16) | 47.99 (70.43) | 1.04e2 (11.05) |
| MILE | 241.91 (86.40) | 262.43 (81.67) | 356.62 (7.27) | 36.87 (62.47) | 60.60 (7.52) | 33.36 (32.68) | 65.94 (9.75) |
| PhUSION | 160.17 (36.11) | 222.93 (5.29) | 360.98 (7.67) | 24.87 (4.02) | 74.25 (7.70) | 27.21 (3.59) | 80.54 (8.46) |
| NRP | 806.64 (190.88) | 2.18e3 (374.90) | 5.82e3 (621.38) | 1.91e3 (660.70) | 7.48e3 (1.06e3) | 3.84e3 (848.78) | 9.63e3 (1.36e3) |
| RandNE | 464.84 (199.62) | 1.36e3 (372.75) | 5.07e3 (877.58) | 2.24e3 (1.50e3) | 1.31e4 (2.30e3) | 2.83e3 (1.83e3) | 1.41e4 (2.56e3) |
| ProNE | 224.35 (77.47) | 295.37 (92.08) | 696.42 (314.10) | 47.03 (125.66) | 78.78 (8.51) | 59.34 (168.78) | 85.04 (9.65) |
| Metis | 141.14 (3.88) | 242.91 (6.56) | 502.63 (12.16) | 38.60 (5.76) | 87.21 (9.53) | 42.88 (7.72) | 94.96 (10.46) |
| hMetis | 189.03 (4.85) | 304.87 (11.92) | 531.05 (13.98) | 69.77 (97.05) | 76.78 (7.44) | 80.56 (129.09) | 83.35 (8.72) |
| GraClus | 150.26 (37.63) | 229.87 (74.71) | 396.54 (24.44) | 39.55 (7.91) | 79.57 (9.25) | 44.04 (7.73) | 86.74 (10.02) |
| GraSAGE | 1.87e5 (1.23e4) | 2.05e5 (1.29e4) | 1.42e5 (1.17e4) | 3.42e4 (4.58e3) | 3.71e4 (5.51e3) | 4.79e4 (6.36e3) | 5.50e4 (6.79e3) |
| GAT | 7.70e4 (9.07e3) | 9.49e4 (9.97e3) | 1.36e5 (1.27e4) | 1.81e4 (5.16e3) | 3.17e4 (5.66e3) | 3.26e4 (5.31e3) | 3.40e4 (6.86e3) |
| GIN | 1.36e5 (1.054e4) | 1.45e5 (1.10e4) | 1.53e5 (1.11e4) | 2.42e4 (6.691e3) | 2.77e4 (7.50e3) | 4.45e4 (8.52e3) | 4.05e4 (9.36e3) |
| GAP | 4.38e3 (1.21e3) | 7.99e3 (2.00e3) | 1.28e4 (2.70e3) | 17.33 (2.74) | 69.5463 (7.8034) | 18.35 (2.51) | 76.2432 (9.5541) |
| ClusNet | 5.58e4 (8.38e3) | 2.80e4 (6.12e3) | 2.91e4 (5.12e3) | 2.97e2 (1.66e2) | 1.11e3 (4.18e2) | 2.91e2 (1.56e2) | 1.19e3 (4.22e2) |
| **IGP-M** | 240.14 (71.27) | 317.82 (91.99) | 345.82 (19.15) | 32.39 (38.31) | 75.92 (8.29) | 34.84 (35.44) | 82.76 (8.97) |
| **IGP-C** | 232.12 (73.70) | 289.53 (88.19) | 354.95 (29.97) | 29.22 (21.72) | 85.00 (9.02) | 38.96 (47.41) | 92.40 (10.05) |

Table 10: Quantitative evaluation results in terms of NCut (↓) on datasets without ground-truth.

| | Taxi | Reddit | AS | Enron |
|---|---|---|---|---|
| SNMF | 0.7851 (0.1299) | 6.7052 (8.4353) | 10.0464 (1.6011) | 23.0720 (1.34e2) |
| DCSBM | 0.7908 (0.1338) | 7.22e3 (3.89e3) | 4.76e2 (3.74e2) | 3.05e3 (2.10e3) |
| SC | 0.7677 (0.1294) | 2.8457 (1.5721) | 2.2442 (0.2929) | 0.7977 (0.5563) |
| IncNSA | 0.5393 (0.0877) | - | 8.26e3 (3.84e3) | - |
| DeepWalk | 0.7999 (0.1198) | 56.7501 (3.51e2) | 6.9109 (1.1831) | 54.3684 (2.48e2) |
| node2vec | 0.8069 (0.1201) | 4.4548 (4.0488) | 6.1055 (1.0540) | 14.9095 (87.0920) |
| LINE | 0.7685 (0.1260) | 9.31e2 (1.45e3) | 2.98e2 (5.46e2) | 1.04e2 (4.30e2) |
| VERSE | 0.7983 (0.1247) | 1.26e3 (1.53e4) | 11.8885 (9.2661) | 1.48e2 (6.50e2) |
| MILE | 0.8378 (0.1261) | 9.98e2 (1.56e3) | 8.6199 (2.3743) | 1.37e2 (6.10e2) |
| PhUSION | 0.8541 (0.1372) | 40.3041 (69.9684) | 6.2456 (1.0237) | 91.8462 (1.35e2) |
| NRP | 51.7825 (57.8389) | 6.84e2 (7.13e2) | 1.72e3 (1.15e3) | 4.66e2 (5.48e2) |
| RandNE | 0.8050 (0.1239) | 5.33e3 (3.50e3) | 6.62e3 (3.08e3) | 3.26e3 (2.35e3) |
| ProNE | 0.7824 (0.1251) | 52.6408 (4.52e2) | 12.8754 (4.0178) | 35.1600 (1.09e2) |
| TIMERS | 3.4756 (0.5099) | - | 1.38e3 (8.06e2) | - |
| Metis | 1.0695 (0.1264) | 2.94e2 (4.45e2) | 15.3312 (3.7055) | 2.95e2 (4.39e2) |
| hMetis | 0.8305 (0.1157) | 6.76e2 (7.88e2) | 39.3087 (1.59e2) | 4.26e2 (6.40e2) |
| GraClus | 0.8493 (0.1195) | 2.20e3 (1.75e3) | 17.8574 (6.7540) | 3.08e2 (7.82e2) |
| GraSAGE | 12.4076 (2.1775) | 6.59e3 (3.97e3) | 1.05e4 (3.85e3) | 4.59e3 (1.68e3) |
| GAT | 1.20e3 (4.26e2) | 5.21e3 (2.86e3) | 5.27e3 (1.84e3) | 2.17e3 (9.53e2) |
| GIN | 25.31 (4.63) | 6.16e3 (3.74e3) | 1.18e4 (3.01e3) | 6.34e3 (2.60e3) |
| GAP | 0.9371 (0.1529) | 4.1535 (4.4771) | 6.1307 (1.2355) | 1.3524 (0.4140) |
| ClusNet | 6.9216 (42.5113) | 28.2975 (53.4356) | 31.4675 (49.0738) | 23.6067 (99.4427) |
| **IGP-M** | 0.7711 (0.1241) | 4.7408 (2.1934) | 7.2568 (1.2706) | 4.5338 (16.9436) |
| **IGP-C** | 0.7736 (0.1253) | 4.7084 (2.1537) | 9.6586 (3.4559) | 4.3581 (16.5047) |

### C.5.2 TRADE-OFF ANALYSIS

In this paper, we focus on the trade-off between quality and efficiency of *online* GP on newly generated graphs from a system. To the best of our knowledge, to evaluate the trade-off between two aspects that conflict with each other using a comprehensive metric remains an open issue. We propose a *trade-off score* (TOS) to quantitatively evaluate the trade-off between quality and efficiency of a given GP method. Concretely, the calculation of TOS is based on the normalized area covered by an induced rectangle of a data point in the quality-efficiency visualization result (like Fig. 3). Fig. 5 gives two running examples regarding how to calculate TOS for a given GP method. Our goal is to let larger TOS indicate a better trade-off between quality and efficiency.

In our evaluation, we use NMI, AC, modularity, and NCut as the quality metrics, while we evaluate the efficiency of a specific method via its total runtime. Let $Q$ and $E$ be the value of a quality metric and the runtime with respect to a GP method to be evaluated. Given the tuple $(E, Q)$, each method used in our experiments can be visualized as a data point in a 2-D space as illustrated in Fig. 3. When we use NMI or AC as the quality metric $Q$, we further derive the normalized metric via

$$\widehat{Q} = Q/1, \tag{15}$$

Table 11: Quantitative evaluation results in terms of runtime (sec) (↓) on datasets w/ ground-truth.

| | GN-0.5 | GN-0.4 | GN-0.3 | L(f,0.3) | L(f,0.6) | L(n,0.3) | L(n,0.6) |
|---|---|---|---|---|---|---|---|
| SNMF | 216.25 (63.49) | 232.32 (51.10) | 326.90 (63.67) | 129.15 (32.03) | 136.21 (0.64) | 155.84 (43.87) | 194.13 (54.34) |
| DCSBM | 143.91 (14.92) | 143.47 (16.88) | 155.99 (19.97) | 123.05 (7.95) | 127.26 (11.40) | 161.94 (24.38) | 165.57 (28.21) |
| SC | 49.38 (0.60) | 46.73 (0.69) | 62.14 (1.08) | 21.04 (1.28) | 42.54 (2.21) | 27.00 (3.85) | 53.26 (6.95) |
| DeepWalk | 35.70 (0.57) | 36.63 (0.67) | 42.92 (0.95) | 14.90 (0.64) | 16.87 (1.04) | 19.45 (1.19) |
| node2vec | 54.36 (0.58) | 54.63 (0.71) | 59.75 (1.01) | 30.97 (0.62) | 33.55 (0.61) | 34.81 (2.03) | 35.86 (2.00) |
| LINE | 34.29 (1.72) | 29.38 (1.99) | 24.16 (1.74) | 8.15 (0.71) | 8.51 (0.62) | 10.84 (1.87) | 13.28 (1.52) |
| VERSE | 185.28 (1.00) | 185.95 (1.04) | 184.87 (1.17) | 112.85 (0.92) | 114.20 (0.97) | 123.72 (6.53) | 125.10 (6.80) |
| MILE | 48.95 (3.76) | 50.77 (3.76) | 56.96 (3.67) | 16.31 (11.62) | 21.36 (3.45) | 17.14 (3.56) | 23.38 (3.60) |
| PhUSION | 112.53 (0.83) | 113.65 (0.51) | 115.98 (0.58) | 85.62 (0.49) | 86.59 (0.45) | 105.61 (11.14) | 105.61 (10.95) |
| NRP | 13.11 (0.28) | 14.03 (0.57) | 16.58 (1.34) | 2.44 (0.23) | 2.82 (0.24) | 5.24 (0.49) | 5.07 (0.57) |
| RandNE | 10.34 (0.09) | 12.48 (0.14) | 14.73 (0.22) | 2.12 (0.18) | 2.49 (0.22) | 2.54 (0.30) | 2.91 (0.33) |
| ProNE | 17.83 (0.10) | 18.85 (0.13) | 21.36 (0.30) | 6.67 (0.23) | 7.46 (0.33) | 7.85 (0.70) | 8.81 (0.71) |
| Metis | 2.21 (0.06) | 2.60 (0.04) | 3.23 (0.08) | 1.07 (0.07) | 1.47 (0.05) | 1.21 (0.10) | 1.64 (0.09) |
| hMetis | 12.68 (0.43) | 13.59 (0.48) | 15.53 (0.58) | 5.51 (0.21) | 7.58 (0.26) | 6.29 (0.50) | 8.62 (0.67) |
| GraClus | 1.59 (0.08) | 1.77 (0.07) | 1.88 (0.10) | 0.88 (0.08) | 0.94 (0.04) | 0.95 (0.04) | 1.03 (0.05) |
| GraSAGE | 8.12 (0.27) | 8.24 (0.31) | 8.18 (0.30) | 2.58 (0.17) | 2.50 (0.16) | 3.12 (0.33) | 3.38 (0.36) |
| GAT | 5.70 (0.12) | 5.56 (0.15) | 5.37 (0.12) | 2.03 (0.16) | 2.37 (0.19) | 2.77 (0.31) | 2.81 (0.29) |
| GIN | 4.49 (0.07) | 4.58 (0.10) | 4.56 (0.06) | 1.86 (0.14) | 1.93 (0.15) | 1.83 (0.22) | 1.85 (0.23) |
| GAP | 19.63 (0.66) | 19.59 (0.59) | 19.59 (0.61) | 66.45 (1.96) | 66.94 (1.75) | 76.79 (6.73) | 77.66 (6.57) |
| ClusNet | 38.66 (0.23) | 39.21 (0.21) | 40.16 (0.24) | 73.28 (4.97) | 76.96 (4.27) | 88.50 (7.63) | 90.07 (8.14) |
| **IGP-M** | 9.93 (0.25) | 11.19 (0.32) | 13.63 (0.55) | 2.48 (0.18) | 3.56 (0.26) | 2.93 (0.32) | 4.19 (0.46) |
| Feat | 0.42 (0.01) | 0.42 (0.01) | 0.42 (0.01) | 0.20 (0.01) | 0.20 (0.01) | 0.23 (0.02) | 0.23 (0.02) |
| Prop | 0.02 (0.001) | 0.02 (0.001) | 0.02 (0.0005) | 0.03 (0.0006) | 0.03 (0.0005) | 0.03 (0.002) | 0.03 (0.002) |
| Clus | 9.49 (0.25) | 10.76 (0.32) | 13.18 (0.55) | 2.25 (0.18) | 3.33 (0.26) | 2.67 (0.31) | 3.93 (0.45) |
| **IGP-C** | 9.93 (0.28) | 11.42 (0.30) | 14.11 (0.50) | 2.84 (0.20) | 3.55 (0.25) | 3.36 (0.37) | 4.24 (0.44) |
| Feat | 0.46 (0.01) | 0.46 (0.01) | 0.471 (0.01) | 0.24 (0.01) | 0.25 (0.01) | 0.28 (0.03) | 0.28 (0.03) |
| Prop | 0.02 (0.002) | 0.01 (0.0003) | 0.02 (0.0003) | 0.03 (0.0004) | 0.03 (0.0004) | 0.03 (0.002) | 0.03 (0.002) |
| Clus | 9.45 (0.28) | 10.95 (0.30) | 13.62 (0.50) | 2.57 (0.20) | 3.28 (0.25) | 3.05 (0.35) | 3.93 (0.42) |

Table 12: Quantitative evaluation results in terms of runtime (sec) (↓) on datasets w/o ground-truth.

| | Taxi | Reddit | AS | Enron |
|---|---|---|---|---|
| SNMF | 2.13 (1.40) | 217.19 (372.52) | 112.31 (83.53) | 1.68 (1.25) |
| DCSBM | 6.15 (0.15) | 9.34 (12.63) | 197.17 (135.89) | 3.17 (1.54) |
| SC | 1.34 (0.08) | 0.35 (0.52) | 15.13 (9.66) | 0.28 (0.33) |
| IncSNA | 0.99 (0.24) | - | 1.43 (0.81) | - |
| DeepWalk | 8.97 (0.09) | 9.99 (6.05) | 34.33 (14.71) | 5.70 (1.74) |
| node2vec | 19.70 (0.31) | 25.37 (17.83) | 29.90 (14.21) | 4.27 (1.49) |
| LINE | 1.00 (0.08) | 2.40 (0.73) | 3.76 (1.12) | 1.38 (0.14) |
| VERSE | 20.17 (0.16) | 25.39 (11.43) | 98.61 (40.19) | 20.18 (0.17) |
| MILE | 11.39 (5.48) | 6.56 (2.79) | 8.49 (2.36) | 2.51 (0.66) |
| PhUSION | 6.47 (0.07) | 4.92 (5.54) | 77.74 (49.35) | 2.53 (1.24) |
| NRP | 0.75 (0.06) | 0.36 (0.20) | 1.58 (0.63) | 0.20 (0.19) |
| RandNE | 0.18 (0.02) | 0.48 (0.27) | 0.74 (0.31) | 0.26 (0.07) |
| ProNE | 0.99 (0.03) | 0.76 (0.51) | 4.89 (2.65) | 0.42 (0.13) |
| TIMERS | 20.20 (2.94) | - | 19.00 (4.66) | - |
| Metis | 0.22 (0.01) | 0.14 (0.06) | 0.37 (0.11) | 0.10 (0.02) |
| hMetis | 2.22 (0.12) | 0.42 (0.24) | 2.60 (1.25) | 0.32 (0.13) |
| GraClus | 1.02 (0.01) | 0.18 (0.05) | 0.42 (0.15) | 0.14 (0.03) |
| GraSAGE | 0.21 (0.02) | 0.35 (0.19) | 1.29 (0.62) | 0.17 (0.04) |
| GAT | 0.21 (0.02) | 0.33 (0.16) | 0.69 (0.30) | 0.14 (0.03) |
| GIN | 0.16 (0.01) | 0.24 (0.10) | 0.67 (0.29) | 0.13 (0.03) |
| GAP | 8.66 (2.59) | 6.32 (4.48) | 70.53 (45.17) | 3.20 (0.89) |
| ClusNet | 19.10 (0.28) | 6.55 (3.92) | 49.93 (29.66) | 4.80 (1.15) |
| **IGP-M** | 0.43 (0.02) | 0.29 (0.14) | 2.37 (1.25) | 0.21 (0.06) |
| Feat | 0.24 (0.004) | 0.007 (0.008) | 1.12 (0.73) | 0.03 (0.01) |
| Prop | 0.003 (0.0001) | 0.01 (0.009) | 0.13 (0.05) | 0.003 (0.001) |
| Clus | 0.19 (0.02) | 0.27 (0.12) | 1.12 (0.48) | 0.18 (0.05) |
| **IGP-C** | 0.43 (0.02) | 0.30 (0.14) | 2.38 (1.28) | 0.22 (0.06) |
| Feat | 0.24 (0.005) | 0.008 (0.009) | 1.16 (0.75) | 0.03 (0.01) |
| Prop | 0.003 (0.0001) | 0.01 (0.009) | 0.13 (0.05) | 0.003 (0.001) |
| Clus | 0.19 (0.01) | 0.28 (0.12) | 1.10 (0.49) | 0.19 (0.05) |

since the value range of NMI or AC is $Q \in [0,1]$ and larger NMI or AC indicate better partitioning quality. Furthermore, when we use modularity as the quality metric, the value range is $Q \in [-1,1]$, in which larger modularity indicates better quality. Hence, we derive the normalized metric as

$$\widehat{Q} = (Q+1)/2. \tag{16}$$

In contrast, smaller NCut metric implies better partitioning quality with value range $Q \in [0, +\infty)$. Therefore, when using NCut as the quality metric $Q$, we normalize it via

$$\widehat{Q} = (Q_m - Q)/Q_m \tag{17}$$

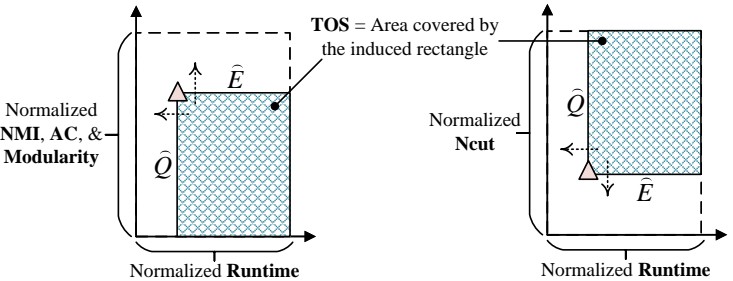

Figure 5: Running examples of the calculation of *trade-off score* (TOS).

Table 13: Trade-off scores (↑) between NMI and runtime on datasets with ground-truth.

|  | GN-0.3 | GN-0.4 | GN-0.5 | L(f,0.3) | L(f,0.6) | L(n,0.3) | L(n,0.6) |
|---|---|---|---|---|---|---|---|
| SNMF | 0 | 0 | 0 | 0 | 0 | 0.0315 | 0 |
| DCSBM | 0.2602 | 0.2206 | 0.2272 | 0.0216 | 0.0179 | 0 | 0.0402 |
| SC | 0.7634 | 0.7887 | 0.7697 | 0.8219 | 0.2460 | 0.8184 | 0.1789 |
| DeepWalk | 0.7194 | 0.8221 | 0.8299 | 0.8251 | 0.4833 | 0.8363 | 0.5075 |
| node2vec | 0.7213 | 0.7564 | 0.7444 | 0.7233 | 0.4455 | 0.7475 | 0.4882 |
| LINE | 0.4931 | 0.6001 | 0.7887 | 0.7562 | 0.4002 | 0.7454 | 0.3613 |
| VERSE | 0.2657 | 0.1691 | 0.1319 | 0.1134 | 0.0726 | 0.2123 | 0.1631 |
| MILE | 0.7712 | 0.7750 | 0.7720 | 0.8201 | 0.1718 | 0.8415 | 0.1818 |
| PhUSION | 0.6058 | 0.5047 | 0.4770 | 0.3197 | 0.1974 | 0.3301 | 0.2532 |
| NRP | 0.7395 | 0.8525 | 0.8922 | 0.5472 | 0.2605 | 0.5022 | 0.2168 |
| RandNE | 0.7561 | 0.8902 | 0.9363 | 0.5669 | 0.1823 | 0.5717 | 0.1859 |
| ProNE | 0.9031 | 0.9146 | 0.9157 | 0.8851 | 0.5282 | 0.8879 | 0.5500 |
| Metis | 0.8006 | 0.9362 | 0.9767 | 0.8423 | 0.4404 | 0.8479 | 0.4515 |
| hMetis | 0.7818 | 0.8833 | 0.9065 | 0.8506 | 0.3933 | 0.8567 | 0.4075 |
| GraClus | 0.9020 | 0.9735 | 0.9881 | 0.8996 | 0.5269 | 0.9007 | 0.5361 |
| GraSAGE | 0.4213 | 0.4435 | 0.4404 | 0.2207 | 0.1550 | 0.2419 | 0.1637 |
| GAT | 0.4382 | 0.4395 | 0.4488 | 0.2263 | 0.1522 | 0.2547 | 0.1560 |
| GIN | 0.4288 | 0.4241 | 0.4239 | 0.1822 | 0.1313 | 0.1741 | 0.1330 |
| GAP | 0.5277 | 0.5894 | 0.6609 | 0.4551 | 0.2406 | 0.4924 | 0.2880 |
| ClusNet | 0.1149 | 0.1467 | 0.2160 | 0.3887 | 0.2414 | 0.4086 | 0.3004 |
| **IGP-M** | 0.9225 | 0.9459 | 0.9526 | 0.9535 | 0.5461 | 0.9554 | 0.5526 |
| **Ranking** | 1 | 2 | 4 | 1 | 1 | 1 | 1 |
| **IGP-C** | 0.9138 | 0.9457 | 0.9527 | 0.9432 | 0.5075 | 0.9456 | 0.5156 |
| **Ranking** | 2 | 3 | 3 | 2 | 4 | 2 | 4 |

with $Q_m$ as the maximum NCut value through all the methods to be evaluated. For efficiency metric $E$ (i.e., runtime), smaller value of $E$ indicates higher efficiency with value range $E \in [0, +\infty)$, so we further normalize $E$ via

$$\widehat{E} = (E_m - E)/E_m, \tag{18}$$

where $E_m$ denotes the maximum runtime through all the methods to be evaluated. Finally, we define TOS with respect to the given quality and efficiency metrics $(Q, E)$ as the (normalized) area covered by the induced rectangle illustrated in Fig. 5. Namely, we have

$$\text{TOS}(Q, E) = \widehat{Q} \times \widehat{E}, \tag{19}$$

in which larger TOS implies a better trade-off between quality and efficiency.

The TOS values with respect to the quality metrics of NMI, AC, modularity, and NCut are depicted in Table 13, Table 14, Table 15, and Table 16, respectively. In our experiments, *we saved model parameters of IGP with best metrics in terms of NMI and modularity on validation sets of datasets with and without ground-truth, respectively*. Namely, *we used NMI and modularity as validation metrics for datasets with and without ground-truth*. For TOS metrics with respect to NMI and modularity (see Table 13 and Table 15), both IGP-M and IGP-C have the best and second-best TOS values in most cases, indicating that IGP can achieve the best trade-off when using NMI and modularity as the (validation) quality metrics. Furthermore, IGP-M and IGP-C can also achieve the top-5 TOS metrics with respect to AC and NCut on all the datasets. In summary, IGP can achieve a better trade-off between quality and efficiency of the *online* GP over various baseline methods.

Table 14: Trade-off scores (↑) between AC and runtime on datasets with ground-truth.

| | GN-0.3 | GN-0.4 | GN-0.5 | L(f,0.3) | L(f,0.6) | L(n,0.3) | L(n,0.6) |
|---|---|---|---|---|---|---|---|
| SNMF | 0 | 0 | 0 | 0 | 0 | 0.0275 | 0 |
| DCSBM | 0.1274 | 0.0988 | 0.1274 | 0.0106 | 0.0090 | 0 | 0.0192 |
| SC | 0.7616 | 0.7723 | 0.7616 | 0.7960 | 0.1789 | 0.8105 | 0.0890 |
| DeepWalk | 0.8049 | 0.7846 | 0.8049 | 0.7777 | 0.4769 | 0.7852 | 0.4960 |
| node2vec | 0.7223 | 0.7307 | 0.7223 | 0.6843 | 0.4402 | 0.7062 | 0.4793 |
| LINE | 0.7250 | 0.4097 | 0.7250 | 0.6759 | 0.3583 | 0.6583 | 0.3133 |
| VERSE | 0.1157 | 0.1397 | 0.1157 | 0.1059 | 0.0650 | 0.1972 | 0.1459 |
| MILE | 0.7634 | 0.7573 | 0.7634 | 0.8091 | 0.0859 | 0.8277 | 0.0874 |
| PhUSION | 0.4628 | 0.4840 | 0.4628 | 0.3005 | 0.1888 | 0.3089 | 0.2405 |
| NRP | 0.7666 | 0.6850 | 0.7666 | 0.4147 | 0.2000 | 0.3750 | 0.1600 |
| RandNE | 0.9052 | 0.8191 | 0.9052 | 0.4000 | 0.1267 | 0.4000 | 0.1261 |
| ProNE | 0.9055 | 0.8974 | 0.9055 | 0.7830 | 0.4919 | 0.7824 | 0.5108 |
| Metis | 0.9690 | 0.9023 | 0.9690 | 0.5821 | 0.3475 | 0.5814 | 0.3507 |
| hMetis | 0.8285 | 0.7950 | 0.8285 | 0.6378 | 0.3097 | 0.6343 | 0.3178 |
| GraClus | 0.9682 | 0.9439 | 0.9682 | 0.7151 | 0.4587 | 0.7139 | 0.4625 |
| GraSAGE | 0.0754 | 0.0739 | 0.0754 | 0.0740 | 0.0568 | 0.0743 | 0.0556 |
| GAT | 0.0896 | 0.0845 | 0.0896 | 0.0839 | 0.0611 | 0.0863 | 0.0561 |
| GIN | 0.0745 | 0.0736 | 0.0745 | 0.0687 | 0.0575 | 0.0601 | 0.0532 |
| GAP | 0.3537 | 0.2720 | 0.3537 | 0.4060 | 0.2037 | 0.4367 | 0.2405 |
| ClusNet | 0.0370 | 0.0244 | 0.0370 | 0.3740 | 0.2445 | 0.3940 | 0.3015 |
| **IGP-M** | 0.9442 | 0.9267 | 0.9442 | 0.9248 | 0.5217 | 0.9254 | 0.5227 |
| **Ranking** | 3 | 3 | 3 | 1 | 1 | 1 | 1 |
| **IGP-C** | 0.9439 | 0.9269 | 0.9439 | 0.9054 | 0.4614 | 0.9054 | 0.4643 |
| **Ranking** | 4 | 2 | 4 | 2 | 4 | 2 | 5 |

Table 15: Trade-off scores (↑) between modularity and runtime on all the datasets.

| | GN-0.3 | GN-0.4 | GN-0.5 | L(f,0.3) | L(f,0.6) | L(n,0.3) | L(n,0.6) | Taxi | Reddit | AS | Enron |
|---|---|---|---|---|---|---|---|---|---|---|---|
| SNMF | 0 | 0 | 0 | 0 | 0 | 0.0294 | 0 | 0.7769 | 0 | 0.3299 | 0.7970 |
| DCSBM | 0.1962 | 0.2091 | 0.1962 | 0.0263 | 0.0344 | 0 | 0.0770 | 0.6039 | 0.4919 | 0 | 0.4398 |
| SC | 0.5715 | 0.5505 | 0.5715 | 0.7003 | 0.4354 | 0.6979 | 0.4485 | 0.8115 | 0.8047 | 0.6608 | 0.8454 |
| IncSNA | - | - | - | - | - | - | - | - | - | 0.5169 | - |
| DeepWalk | 0.6173 | 0.5781 | 0.6173 | 0.7179 | 0.5809 | 0.7272 | 0.6031 | 0.4830 | 0.7902 | 0.6462 | 0.6271 |
| node2vec | 0.5536 | 0.5274 | 0.5536 | 0.6204 | 0.5068 | 0.6410 | 0.5488 | 0.0211 | 0.7477 | 0.6710 | 0.6972 |
| LINE | 0.5994 | 0.5065 | 0.5994 | 0.7302 | 0.5782 | 0.7241 | 0.5625 | 0.8267 | 0.7776 | 0.7113 | 0.7965 |
| VERSE | 0.1012 | 0.1283 | 0.1012 | 0.1013 | 0.1045 | 0.1894 | 0.2301 | 0.0212 | 0.6698 | 0.3805 | 0 |
| MILE | 0.5733 | 0.5395 | 0.5733 | 0.7268 | 0.5115 | 0.7446 | 0.5347 | 0.3770 | 0.7714 | 0.7504 | 0.7653 |
| PhUSION | 0.3547 | 0.3519 | 0.3547 | 0.2751 | 0.2437 | 0.2840 | 0.3056 | 0.5875 | 0.8194 | 0.4791 | 0.7338 |
| NRP | 0.6820 | 0.6310 | 0.6820 | 0.6098 | 0.5243 | 0.6048 | 0.5146 | 0.5107 | 0.5192 | 0.5189 | 0.4963 |
| RandNE | 0.7012 | 0.6416 | 0.7012 | 0.5825 | 0.5069 | 0.5831 | 0.5092 | 0.8578 | 0.5264 | 0.5245 | 0.5998 |
| ProNE | 0.6799 | 0.6349 | 0.6799 | 0.7633 | 0.6269 | 0.7659 | 0.6360 | 0.8256 | 0.8115 | 0.7270 | 0.7971 |
| TIMERS | - | - | - | - | - | - | - | 0 | - | 0.4739 | - |
| Metis | 0.7289 | 0.6686 | 0.7289 | 0.7556 | 0.6455 | 0.7564 | 0.6474 | 0.8431 | 0.7612 | 0.7623 | 0.8213 |
| hMetis | 0.6814 | 0.6337 | 0.6814 | 0.7540 | 0.6223 | 0.7566 | 0.6307 | 0.7717 | 0.7699 | 0.7699 | 0.8245 |
| GraClus | 0.7337 | 0.6815 | 0.7337 | 0.7625 | 0.6535 | 0.7626 | 0.6551 | 0.8215 | 0.7257 | 0.7325 | 0.8044 |
| GraSAGE | 0.4816 | 0.4825 | 0.4816 | 0.4914 | 0.4917 | 0.4931 | 0.4921 | 0.6030 | 0.5338 | 0.5250 | 0.5423 |
| GAT | 0.4892 | 0.4894 | 0.4892 | 0.4970 | 0.4917 | 0.4998 | 0.4929 | 0.5002 | 0.5426 | 0.5212 | 0.5175 |
| GIN | 0.4899 | 0.4903 | 0.4899 | 0.4953 | 0.4932 | 0.5177 | 0.5160 | 0.5362 | 0.5235 | 0.5263 | 0.5504 |
| GAP | 0.5426 | 0.5034 | 0.5426 | 0.4041 | 0.3381 | 0.4381 | 0.3988 | 0.4907 | 0.8014 | 0.5084 | 0.7383 |
| ClusNet | 0.4107 | 0.4157 | 0.4107 | 0.3581 | 0.2938 | 0.3759 | 0.3625 | 0.0471 | 0.7531 | 0.5820 | 0.6574 |
| **IGP-M** | 0.7072 | 0.6576 | 0.7072 | 0.8154 | 0.6532 | 0.8172 | 0.6566 | 0.8514 | 0.8458 | 0.7747 | 0.8626 |
| **Ranking** | 4 | 3 | 4 | 1 | 2 | 1 | 1 | 3 | 1 | 1 | 1 |
| **IGP-C** | 0.7072 | 0.6570 | 0.7072 | 0.8104 | 0.6419 | 0.8120 | 0.6453 | 0.8516 | 0.8457 | 0.7676 | 0.8612 |
| **Ranking** | 3 | 4 | 3 | 2 | 4 | 2 | 4 | 2 | 2 | 3 | 2 |

### C.5.3 ROBUSTNESS ANALYSIS

We further test the robustness of IGP across graphs using a more challenging setting of the *LFR-Net*. Different from the quantitative evaluation in Appendix C.5.1 where fixed settings of $(k, k_{max,}, c_{min,c_{max}})$ and $\mu$ were used, we randomly set $\mu \in [0.3, 0.6]$, $k \in [10, 20]$, $k_{max} \in [100, 200]$, $c_{min} \in [10, 20]$, $c_{max} \in [100, 200]$, and $N \in [500, 10000]$ to independently generate $1,000$ graphs with GP ground-truth for the robustness analysis. In this case, the difference between synthetic graphs (in terms of their underlying distributions) is much larger than that in Appendix C.5.1. We adopted the same experiment settings and evaluation protocols as in Appendix C.5.1 to compare the quality, efficiency, and trade-off between the two aspects of all the methods on this new dataset.

The robustness analysis results in terms of **NMI**, **AC**, **modularity**, **NCut**, **runtime**, and **TOS** values between each quality metric (from the four metrics) and one efficiency metric are depicted in Table 17. In most cases, IGP-M can achieve the best TOS values. The TOS values of IGP-C are also competitive to those of IGP-M. In summary, the robustness analysis results demonstrate that our

Table 16: Trade-off scores (↑) between NCut and runtime on all the datasets.

| | GN-0.3 | GN-0.4 | GN-0.5 | L(f,0.3) | L(f,0.6) | L(n,0.3) | L(n,0.6) | Taxi | Reddit | AS | Enron |
|---|---|---|---|---|---|---|---|---|---|---|---|
| SNMF | 0 | 0 | 0 | 0 | 0 | 0.037673 | 0 | 0.8939 | 0 | 0.4300 | 0.9134 |
| DCSBM | 0.4637 | 0.3682 | 0.3278 | 0.0427 | 0.0617 | 0 | 0.1404 | 0.6949 | 0 | 0 | 0.4375 |
| SC | 0.8081 | 0.7978 | 0.7705 | 0.8362 | 0.6855 | 0.8326 | 0.7245 | 0.9332 | 0.9980 | 0.9231 | 0.9862 |
| IncSNA | - | - | - | - | - | - | - | 0.9504 | - | 0.2978 | 0.9862 |
| DeepWalk | 0.8627 | 0.8403 | 0.8337 | 0.8837 | 0.8657 | 0.8950 | 0.8983 | 0.5557 | 0.9465 | 0.8254 | 0.7117 |
| node2vec | 0.8137 | 0.7636 | 0.7477 | 0.7590 | 0.7521 | 0.7842 | 0.8140 | 0.0244 | 0.8826 | 0.8479 | 0.7865 |
| LINE | 0.9078 | 0.8682 | 0.8397 | 0.8665 | 0.8818 | 0.8777 | 0.8988 | 0.9499 | 0.8615 | 0.9561 | 0.9165 |
| VERSE | 0.4318 | 0.1991 | 0.1429 | 0.1261 | 0.1611 | 0.2358 | 0.3549 | 0.0012 | 0.7290 | 0.4994 | 0 |
| MILE | 0.8238 | 0.7805 | 0.7727 | 0.8728 | 0.8418 | 0.8935 | 0.8785 | 0.4358 | 0.8358 | 0.9562 | 0.8567 |
| PhUSION | 0.6437 | 0.5103 | 0.4792 | 0.3368 | 0.3636 | 0.3477 | 0.4553 | 0.6790 | 0.9719 | 0.6054 | 0.8618 |
| NRP | 0.9134 | 0.9296 | 0.9354 | 0.9263 | 0.7819 | 0.8899 | 0.8036 | 0.9211 | 0.9039 | 0.8474 | 0.9173 |
| RandNE | 0.9234 | 0.9500 | 0.9498 | 0.9190 | 0.6348 | 0.9260 | 0.7317 | 0.9903 | 0.2610 | 0.4373 | 0.4796 |
| ProNE | 0.9304 | 0.9175 | 0.9165 | 0.9471 | 0.9432 | 0.9503 | 0.9532 | 0.9505 | 0.9892 | 0.9742 | 0.9738 |
| TIMERS | - | - | - | - | - | - | - | 0 | - | 0.7980 | - |
| Metis | 0.9869 | 0.9877 | 0.9890 | 0.9906 | 0.9868 | 0.9917 | 0.9899 | 0.9882 | 0.9586 | 0.9968 | 0.9488 |
| hMetis | 0.9492 | 0.9401 | 0.9404 | 0.9554 | 0.9424 | 0.9596 | 0.9542 | 0.8893 | 0.9046 | 0.9835 | 0.9182 |
| GraClus | 0.9917 | 0.9913 | 0.9919 | 0.9920 | 0.9910 | 0.9932 | 0.9931 | 0.9487 | 0.6939 | 0.9964 | 0.9447 |
| GraSAGE | 0.0740 | 0 | 0 | 0 | 0 | 0 | 0 | 0.9795 | 0.0869 | 0.8882 | 0.2738 |
| GAT | 0.1082 | 0.5257 | 0.5742 | 0.4615 | 0.1426 | 0.3121 | 0.3755 | 0 | 0.2779 | 0.5514 | 0.6528 |
| GIN | 0 | 0.2875 | 0.2674 | 0.2883 | 0.2491 | 0.0695 | 0.2621 | 0.9712 | 0.1465 | 0 | 0 |
| GAP | 0.8617 | 0.8801 | 0.8880 | 0.4852 | 0.5076 | 0.5256 | 0.5991 | 0.5704 | 0.9704 | 0.6419 | 0.8414 |
| ClusNet | 0.7112 | 0.7178 | 0.5768 | 0.4289 | 0.4220 | 0.4508 | 0.5244 | 0.0543 | 0.9660 | 0.7448 | 0.7591 |
| **IGP-M** | 0.9562 | 0.9503 | 0.9529 | 0.9799 | 0.9719 | 0.9812 | 0.9769 | 0.9780 | 0.9980 | 0.9874 | 0.9889 |
| **Ranking** | 3 | 3 | 4 | 3 | 3 | 3 | 3 | 5 | 2 | 3 | 1 |
| **IGP-C** | 0.9546 | 0.9495 | 0.9529 | 0.9772 | 0.9717 | 0.9785 | 0.9765 | 0.9782 | 0.9980 | 0.9871 | 0.9883 |
| **Ranking** | 4 | 4 | 3 | 4 | 4 | 4 | 4 | 4 | 3 | 4 | 2 |

Table 17: Robustness analysis on *LFR-Net* with various parameter settings.

| | NMI | AC | Mod | NCut | Runtime | TOS (4 Quality Metrics vs Runtime) | | | |
|---|---|---|---|---|---|---|---|---|---|
| | | | | | | NMI | AC | Mod | NCut |
| SNMF | 81.17 (12.15) | 74.56 (11.96) | 42.28 (9.64) | 73.46 (54.79) | 221.88 (258.69) | 0.0213 | 0.0195 | 0.0186 | 0.0262 |
| DCSBM | 43.97 (10.50) | 23.44 (12.53) | 8.63 (5.01) | 8.51e3 (1.03e4) | 227.85 (261.93) | 0 | 0 | 0 | 0 |
| SC | 90.77 (16.63) | 88.40 (19.25) | 48.26 (11.42) | 89.91 (118.07) | 11.81 (12.39) | 0.8607 | 0.838 | 0.7029 | 0.9467 |
| DW | 89.73 (11.43) | 87.83 (8.99) | 46.76 (9.93) | 107.92 (162.86) | 16.40 (9.82) | 0.8327 | 0.8151 | 0.6810 | 0.9263 |
| N2V | 92.72 (9.68) | 90.84 (8.20) | 47.53 (9.88) | 161.28 (443.84) | 41.36 (24.57) | 0.7589 | 0.7435 | 0.6038 | 0.8162 |
| LINE | 64.32 (21.41) | 56.32 (25.18) | 32.37 (14.38) | 7.74e3 (9.54e3) | 10.70 (8.14) | 0.6130 | 0.5368 | 0.6308 | 0.8276 |
| VERSE | 85.39 (11.89) | 81.96 (10.44) | 44.23 (9.30) | 86.66 (77.58) | 96.86 (51.15) | 0.4909 | 0.4712 | 0.4146 | 0.5740 |
| MILE | 86.51 (14.70) | 86.19 (12.98) | 47.63 (10.60) | 198.78 (583.55) | 14.29 (10.74) | 0.8108 | 0.8078 | 0.6919 | 0.9341 |
| PhUSION | 91.01 (12.12) | 88.08 (10.99) | 47.29 (9.94) | 57.01 (48.21) | 127.04 (113.76) | 0.4027 | 0.3897 | 0.3258 | 0.4420 |
| NRP | 56.13 (14.61) | 43.03 (9.14) | 16.68 (6.81) | 5.50e3 (6.40e3) | 3.21 (2.68) | 0.5534 | 0.4242 | 0.5752 | 0.8937 |
| RandNE | 43.43 (17.83) | 29.49 (9.96) | 8.34 (5.61) | 1.21e4 (1.54e4) | 2.99 (2.51) | 0.4286 | 0.291 | 0.5346 | 0.7838 |
| ProNE | 90.90 (13.43) | 86.24 (11.52) | 46.36 (9.51) | 373.46 (1.39e3) | 8.35 (6.53) | 0.8757 | 0.8308 | 0.7050 | 0.9572 |
| Metis | 74.77 (7.39) | 65.14 (5.20) | 40.34 (6.87) | 69.46 (48.30) | 1.52 (0.94) | 0.7427 | 0.6471 | 0.6970 | 0.9922 |
| hMetis | 78.53 (14.62) | 68.81 (12.20) | 43.17 (9.50) | 891.54 (3.10e3) | 9.23 (6.57) | 0.7535 | 0.6602 | 0.6869 | 0.9449 |
| GraClus | 88.70 (9.76) | 78.60 (7.70) | 39.87 (7.30) | 370.67 (1.64e3) | 0.78 (0.35) | 0.8840 | 0.7833 | 0.6970 | 0.9903 |
| GraSAGE | 23.38 (6.61) | 9.19 (4.49) | 0.88 (1.35) | 5.88e4 (4.94e4) | 3.11 (2.52) | 0.2306 | 0.0906 | 0.4975 | 0.0000 |
| GAT | 24.55 (6.85) | 9.83 (4.78) | 0.49 (0.96) | 4.57e4 (3.89e4) | 2.98 (2.39) | 0.2423 | 0.097 | 0.4959 | 0.2199 |
| GIN | 16.05 (5.41) | 8.31 (5.07) | 1.04 (1.28) | 5.12e4 (4.23e4) | 2.55 (2.04) | 0.1587 | 0.0822 | 0.4995 | 0.1278 |
| GAP | 86.70 (11.10) | 71.95 (9.78) | 48.01 (9.85) | 42.37 (34.88) | 129.57 (112.81) | 0.3740 | 0.31035 | 0.3192 | 0.4310 |
| ClusNet | 87.84 (9.23) | 84.95 (7.70) | 47.94 (10.14) | 450.70 (415.82) | 76.61 (51.80) | 0.5831 | 0.5639 | 0.4910 | 0.6587 |
| **IGP-M** | 92.41 (10.97) | 91.78 (9.08) | 48.71 (9.97) | 185.17 (643.65) | 3.54 (2.95) | 0.9097 | 0.9035 | 0.7320 | 0.9814 |
| TOS Rank. | - | - | - | - | - | 1 | 1 | 1 | 3 |
| **IGP-C** | 87.24 (15.80) | 87.59 (14.90) | 47.46 (11.07) | 171.96 (546.45) | 3.96 (3.37) | 0.8572 | 0.8607 | 0.7245 | 0.9797 |
| TOS Rank. | - | - | - | - | - | 5 | 2 | 2 | 4 |

IGP method is robust to the large distribution difference of graphs in the same system or scenario, while ensuring its ability to achieve significant trade-off between quality and efficiency.

### C.5.4 CONVERGENCE ANALYSIS

We also analyzed the convergence of the *offline* training procedure of IGP (i.e., Algorithm 3). In each epoch of the *offline* training on a dataset, we recorded the average partitioning quality (in terms of NMI, AC, modularity, and NCut) with respect to the learned embedding $\{\mathbf{U}_t\}$ (i) on training set $\Gamma_T$ and validation set $\Gamma_V$ (i.e., historical system snapshots). Moreover, we also evaluated the average partitioning quality of the generalized embedding $\{\mathbf{U}_t\}$ on test set $\Gamma'$ (i.e., new snapshots) in each epoch. To further validate the effectiveness of *auxiliary label-induced embedding* $\{\mathbf{U}_t^{(g)}\}$,

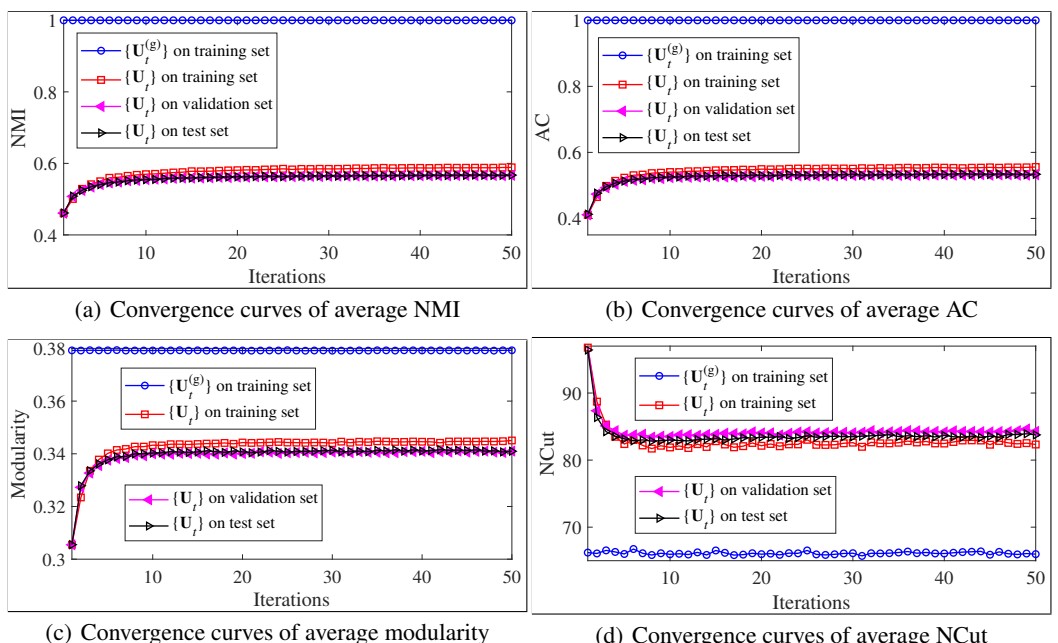

(a) Convergence curves of average NMI

(b) Convergence curves of average AC

(c) Convergence curves of average modularity

(d) Convergence curves of average NCut

Figure 6: Convergence curves of IGP-M on $L(n,.6)$ in terms of NMI, AC, modularity, and NCut.

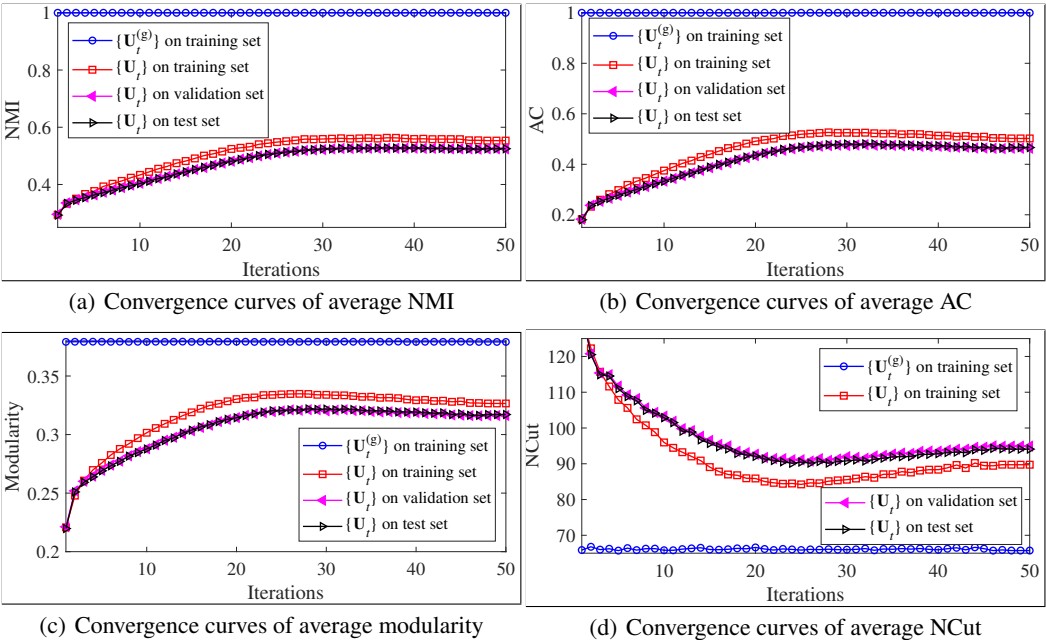

(a) Convergence curves of average NMI

(b) Convergence curves of average AC

(c) Convergence curves of average modularity

(d) Convergence curves of average NCut

Figure 7: Convergence curves of IGP-C on $L(n,.6)$ in terms of NMI, AC, modularity, and NCut.

the average partitioning quality of $\{\mathbf{U}_t^{(g)}\}$ on training set was also recorded in each epoch during the *offline* training. Note that we do not need to derive $\{\mathbf{U}_t^{(g)}\}$ for $\Gamma_V$ and $\Gamma'$.

We use evaluation records on the challenging $L(n, 0.6)$ dataset, whose clustering structures are difficult to identify, as an example. The convergence curves (within the first 50 epochs) of the average NMI, AC, modularity, and NCut with respect to (i) $\{\mathbf{U}_t^{(g)}\}$ on training set $\Gamma_T$, (ii) $\{\mathbf{U}_t\}$ on training set $\Gamma_T$, (iii) $\{\mathbf{U}_t\}$ on validation set $\Gamma_V$, and (iv) $\{\mathbf{U}_t\}$ on test set $\Gamma'$ for IGP-M and IGP-C are illustrated in Fig. 6 and Fig. 7, respectively.

Table 18: Detailed ablation study on $L(f,0.6)$.

| | NMI (%) (↑) | | AC (%) (↑) | | Modularity (%) (↑) | | Ncut (↓) | |
|---|---|---|---|---|---|---|---|---|
| | IGP-M | IGP-C | IGP-M | IGP-C | IGP-M | IGP-C | IGP-M | IGP-C |
| IGP | **56.07** (2.33) | **52.11** (2.79) | **53.57** (2.21) | **47.38** (2.68) | **34.14** (0.53) | **31.81** (0.65) | **75.92** (8.29) | **84.99** (9.02) |
| w/o AL | 51.73 (2.86) | 34.83 (3.29) | 49.45 (2.59) | 26.64 (2.79) | 33.27 (0.63) | 26.88 (0.80) | 77.05 (7.33) | 101.79 (8.61) |
| w/o FR | 11.89 (1.80) | 15.60 (1.75) | 6.57 (0.34) | 8.25 (0.55) | 9.14 (1.38) | 5.73 (0.62) | 6.90e2 (1.59e2) | 1.02e3 (8.22e2) |
| w/o CR | 49.50 (3.06) | 34.84 (3.25) | 46.83 (2.84) | 26.57 (2.65) | 32.61 (0.72) | 26.90 (0.76) | 78.93 (7.19) | 102.31 (8.70) |
| w/o $\mathbf{Z}_t$ | 14.13 (1.65) | 15.39 (1.81) | 5.71 (0.13) | 5.79 (0.14) | 0.24 (0.15) | 0.76 (0.19) | 2.62e4 (4.49e3) | 1.03e4 (4.42e3) |
| w/o GNN | 1.03 (0.20) | 8.75 (0.55) | 4.22 (0.15) | 5.17 (0.15) | 0.01 (0.01) | 0.86 (0.23) | 1.70e3 (4.99e2) | 4.01e4 (7.24e3) |

Table 19: Detailed ablation study on $L(n,0.6)$.

| | NMI (%) (↑) | | AC (%) (↑) | | Modularity (%) (↑) | | NCut (↓) | |
|---|---|---|---|---|---|---|---|---|
| | IGP-M | IGP-C | IGP-M | IGP-C | IGP-M | IGP-C | IGP-M | IGP-C |
| IGP | **56.48** (2.08) | **52.71** (2.52) | **53.42** (2.04) | **47.47** (2.59) | **34.22** (0.47) | **31.95** (0.62) | **82.76** (8.97) | **92.40** (10.05) |
| w/o AL | 52.13 (2.52) | 35.12 (3.01) | 49.30 (2.41) | 26.15 (2.61) | 33.35 (0.58) | 26.70 (0.74) | 84.05 (8.63) | 113.10 (10.23) |
| w/o FR | 11.19 (2.04) | 15.46 (1.68) | 6.28 (0.45) | 5.46 (0.18) | 10.08 (1.77) | 0.45 (0.11) | 1.01e3 (237.66) | 2.08e4 (4.79e3) |
| w/o CR | 49.97 (2.79) | 35.16 (2.92) | 46.70 (2.73) | 26.22 (2.50) | 32.70 (0.68) | 26.78 (0.71) | 86.08 (8.71) | 111.62 (10.39) |
| w/o $\mathbf{Z}_t$ | 16.39 (1.59) | 13.88 (1.39) | 6.14 (0.26) | 5.57 (0.18) | 1.27 (0.47) | 0.28 (0.15) | 3.01e4 (4.76e3) | 2.88e4 (5.25e3) |
| w/o GNN | 1.14 (0.25) | 8.45 (0.45) | 3.92 (0.18) | 4.78 (0.22) | 0.01 (0.02) | 0.32 (0.32) | 1.08e4 (2.86e3) | 5.12e4 (7.94e3) |

One can have similar observations in Fig. 6 and Fig. 7. Concretely, the partitioning quality (in terms of NMI, AC, modularity, and NCut) of $\{\mathbf{U}_t\}$ continuously improves in the first several epochs on the training, validation, and test set. More importantly, the average NMI and AC of $\{\mathbf{U}_t^{(g)}\}$ on training set $\Gamma_T$ are always 1 in all the epochs for both IGP-M and IGP-C. It indicates that the partitioning results derived from the *auxiliary label-induced embedding* $\{\mathbf{U}_t^{(g)}\}$ have perfect mapping to the ground-truth. For both IGP-M and IGP-C, the average modularity and NCut of $\{\mathbf{U}_t^{(g)}\}$ also keep at values that are much better than metrics of $\{\mathbf{U}_t\}$ on the training, validation, and test set. The convergence curves with respect to $\{\mathbf{U}_t^{(g)}\}$ further verify that the introduced *label-induced embedding* $\{\mathbf{U}_t^{(g)}\}$ can capture much more information regarding 'ground-truth' of the training set (i.e., historical system snapshots) than the learned embedding $\{\mathbf{U}_t\}$, which enables IGP to capture the permutation invariant label information during the *offline* training. It also ensures the strong robustness of IGP beyond existing baselines.

In summary, the convergence curves in Fig. 6 and Fig. 7 validate the effectiveness of the *offline* training procedure and the dual GNN structure of IGP.

### C.5.5 ABLATION STUDY

In addition to example results shown in Table 2, we conducted additional ablation studies on *GN*-0.3, $L(f,0.6)$, and $L(n,0.6)$, in which the clustering structures are more difficult to identify compared with other datasets. In each case of the ablation study (see Section 4), the mean value $\mu$ and standard deviation $\sigma$ of all the quality metrics (on test set $\Gamma'$) were recorded. The detailed ablation study results on $L(f,0.6)$, $L(n,0.6)$, and *GN*-0.3 (in the format $\mu(\sigma)$) are illustrated in Table 18, Table 19, and Table 20, respectively.

Note that Table 20, Table 18, and Table 19 have consistent results. The FR loss, feature input $\mathbf{Z}_t$, and GNN are key components to ensure the high quality of IGP, because there are significant quality declines in cases without the three components. Moreover, the AL and CR losses are the components to further enhance the partitioning quality, as they can incorporate the permutation invariant label information of historical snapshots to the *offline* training of IGP.

## D EXTENDED APPLICATIONS

### D.1 MODEL SELECTION

As defined in Section 2, we assume that the number of clusters $K$ is given in the downstream GP. For *online* GP in which $K$ is not given, IGP also has the potential to estimate the $K$ value for a given graph. As a demonstration, we applied XMeans, a model selection method, to the derived embedding $\{\mathbf{U}_t\}$ of both IGP-M and IGP-C to estimate $K$ for each snapshot in the test sets of $L(f,.3)$, $L(f,.6)$, $L(n,.3)$ and $L(n,.6)$. In the 4 datasets, ground-truth regarding the $K$ value is

Table 20: Detailed ablation study on *GN*-0.3.

| | NMI (%) (↑) | | AC (%) (↑) | | Modularity (%) (↑) | | NCut (↓) | |
|---|---|---|---|---|---|---|---|---|
| | IGP-M | IGP-C | IGP-M | IGP-C | IGP-M | IGP-C | IGP-M | IGP-C |
| IGP | **96.26** (0.32) | **95.50** (0.43) | **91.96** (0.73) | **90.47** (0.85) | **27.92** (0.22) | **27.68** (0.25) | **345.82** (19.15) | **354.95** (29.97) |
| w/o AL | 93.31 (0.57) | 95.00 (0.41) | 87.30 (1.02) | 89.75 (0.85) | 26.98 (0.30) | 27.51 (0.25) | 427.44 (97.97) | 363.44 (42.56) |
| w/o FR | 47.29 (0.69) | 72.47 (0.93) | 16.87 (0.57) | 56.06 (1.46) | 7.27 (0.27) | 15.06 (0.43) | 1.95e4 (2.84e3) | 1.34e3 (827.27) |
| w/o CR | 93.75 (0.45) | 95.03 (0.41) | 87.85 (0.83) | 89.86 (0.77) | 27.11 (0.26) | 27.52 (0.24) | 404.90 (85.24) | 371.18 (83.18) |
| w/o $\mathbf{Z}_t$ | 44.20 (0.18) | 44.91 (0.17) | 7.34 (0.12) | 7.73 (0.12) | 0.03 (0.03) | 0.03 (0.03) | 1.68e5 (1.28e4) | 1.65e5 (1.32e4) |
| w/o GNN | 30.08 (0.69) | 56.25 (1.28) | 12.99 (0.50) | 33.66 (1.20) | 4.53 (0.19) | 11.80 (0.32) | 6.11e3 (646.84) | 996.69 (439.41) |

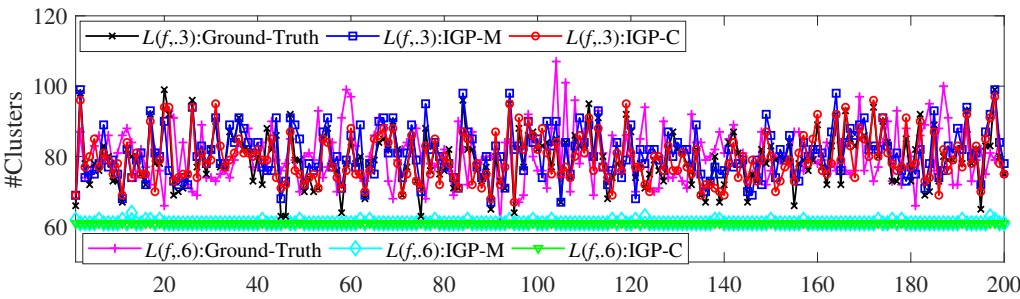

Figure 8: Model selection results of IGP-M and IGP-C on the test set of $L(f,0.3)$ and $L(f,0.6)$.

given by the dataset and $K$ is non-fixed for multiple test graphs. With the increase of $\mu$, the clustering structures of $L(f, \mu)$ and $L(n, \mu)$ are increasingly difficult to identify. In particular, we use the same parameter settings and layer configurations with that in Table 4. The range of $K$ of XMeans for each dataset was set according to the maximum and minimum number of clusters as illustrated in Table 3. The model selection results and ground-truth with respect to the test sets of $L(f, \mu)$ and $L(n, \mu)$ are depicted in Fig. 8 and Fig 9, respectively.

The model selection results in Fig. 8 and Fig 9 are similar. When $\mu = 0.3$ (with *distinct clustering structures*), the model selection results of both IGP-M and IGP-C fit well to ground-truth on most test snapshots of $L(f, 0.3)$ and $L(n, 0.3)$. Note that the potential of model selection of IGP is still largely influenced by the intrinsic clustering structures of a dataset. The estimated $K$ values are still far from ground-truth when $\mu = 0.6$ (with *indistinct clustering structures*) on $L(f, 0.6)$ and $L(n, 0.6)$. In summary, IGP has the potential to estimate the number of clusters $K$ of newly generated graph snapshots with distinct clustering structures.

## D.2 TRANSFERRING TEST

To test the potential of IGP to transfer from one (training) scenario to other different (generalization) scenarios, we consider a challenging application in which we first conduct *offline* training of IGP on completely synthetic graphs and directly generalize the trained model to the real datasets for fast *online* GP (without additional optimization).

In particular, we expect the synthetic graphs used for *offline* training cover various structural properties of real-world network systems. We adopt *LFR-Net* (see Appendix C.2) as the model to generate synthetic training graphs. In contrast to the parameter settings of *LFR-Net* in our quantitative evaluation (see Appendix C.2), we randomly set $N \in [500, 6000]$, $k \in [10, 100]$, $k_{\max} \in [50, 200]$, $c_{\min} \in [5, 100]$, $c_{\max} \in [50, 400]$, and $\mu \in [0.1, 0.7]$ to independently generate $2,000$ synthetic snapshots, where we ensure that $k_{\max} > k$ and $c_{\max} > c_{\min}$ for each snapshot.

In our experiments, we conducted *offline* training on the aforementioned synthetic snapshots and generalized the trained model to test sets of the four real datasets used in our quantitative evaluation (i.e., *Taxi*, *Reddit*, *AS*, and *Enron* as introduced in Appendix C.2). We used the same layer configurations as in Table 4 with respect to each real dataset and tried different parameter settings of IGP-M and IGP-C to report their best average quality. The recommended parameter settings of IGP-M and IGP-C are shown in Table 21. Evaluation results on test sets of the four real datasets (in terms of modularity and NCut) are depicted in Table 22, where 'Real' and 'Syn.' represent the results with respect to (i) training on training set of the original real datasets (same as the results in Table 8 and

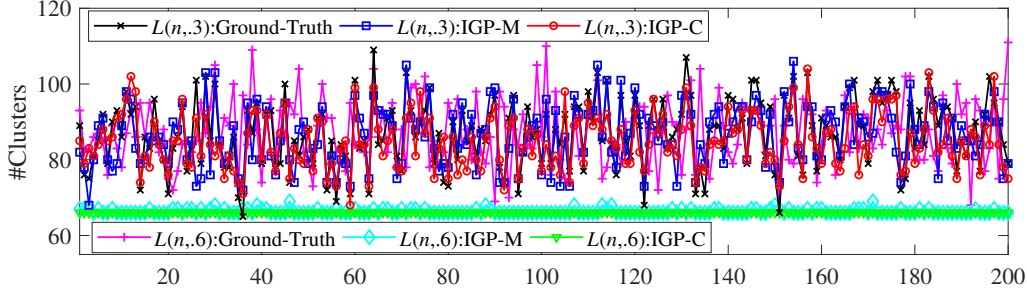

Figure 9: Model selection results of IGP-M and IGP-C on the test set of $L(n,0.3)$ and $L(n,0.6)$.

Table 21: Parameter setting of IGP in the transferring test.

| | IGP-M | | IGP-C | |
|---|---|---|---|---|
| | $(\alpha, \beta, m, p)$ | $(\eta_G, \eta_D)$ | $(\alpha, \beta, m, p)$ | $(\eta_G, \eta_D)$ |
| Taxi | (1, 1, 1, 1000) | (1e-4, 1e-4) | (1, 0.1, 1, 1000) | (1e-4, 1e-4) |
| Reddit | (1, 0.1, 1000) | (1e-4, 1e-4) | (1, 10, 1, 1000) | (5e-5, 5e-5) |
| AS | (1, 0.1, 1, 1000) | (1e-4, 1e-4) | (1, 5, 1, 1000) | (1e-4, 1e-4) |
| Enron | (1, 1, 1, 1000) | (1e-4, 1e-4) | (1, 0.1, 1, 1000) | (5e-5, 5e-5) |

Table 10) and (ii) training on the aforementioned synthetic snapshots, respectively. According to Table 22, we have the following key observations.

- In some cases (e.g., on *Taxi* and *Reddit*), the quality metrics of training on synthetic graphs are close to those of training on the original real datasets for both IGP-M and IGP-C, which are also competitive to the strong baselines as illustrated in Table 8 and Table 10. It preliminarily validates the potential of IGP to be transferred from completely synthetic graphs to real datasets.

- However, in other cases, there are still significant quality declines for both the variants of IGP, which indicate that the transferring capability of IGP is largely affected by the domain similarity between the synthetic training set and real test set. The negative transferring can be alleviated by carefully generating or selecting synthetic snapshots with the assistant of validation information from the target real datasets. We leave the study of this application in our future work.

- Compared with IGP-C, IGP-M has better transferring quality metrics in most cases. The quality declines of IGP-C are also more significant than those of IGP-M. It further validates that the modularity matrix $\mathbf{Q}_t$ used in IGP-M is more informative than the normalized adjacency matrix $\mathbf{M}_t$ used in IGP-C.

Note that IGP has similar motivations with SOTA pre-training methods for graphs, e.g., Pretrain-GNN (Hu et al., 2020) and GCC (Qiu et al., 2020), in which we pre-train a GNN-based model on historical graphs (in an *offline* way) and generalize it to new snapshots. The major difference between IGP and SOTA pre-training methods is that a *fine-tuning* procedure with another optimization algorithm (based on a certain fine-tuning loss) on new test graphs is required for most pre-training methods after the *offline* (pre-)training. In particular, the *fine-tuning* procedure may still be time-consuming for new test graphs, where several iterative optimization algorithms (e.g., gradient descent) are applied.

In this study, we focus on the NP-hard challenge of GP and try to achieve a better trade-off between quality and efficiency. To some extent, the proposed IGP framework sacrifices some opportunities to fine-tune the (pre-)trained model, which ensures the low runtime on new test graphs. In our future research, we intend to consider a promising application that we pre-train a high-quality GP model on completely synthetic graphs and fast fine-tune the (pre-)trained model on new graphs (via some efficient fine-tuning strategies), with the guarantee of high partitioning quality and low runtime.

Table 22: Transferring test results on real datasets.

| | | Modularity (%) (↑) | | | | NCut (↓) | | | |
|---|---|---|---|---|---|---|---|---|---|
| | | Taxi | Reddit | AS | Enron | Taxi | Reddit | AS | Enron |
| **IGP-M** | Real | 74.01 (0.51) | 69.38 (5.49) | 56.82 (2.71) | 74.33 (13.51) | 0.77 (0.12) | 4.74 (2.19) | 7.26 (1.27) | 4.53 (16.94) |
| | Syn. | 74.04 (0.46) | 69.31 (6.04) | 54.71 (2.28) | 73.96 (17.58) | 0.77 (0.12) | 4.14 (1.82) | 7.24 (1.21) | 18.47 (98.84) |
| **IGP-C** | Real. | 74.00 (0.42) | 69.37 (5.47) | 55.40 (2.26) | 74.15 (11.78) | 0.77 (0.13) | 4.71 (2.15) | 9.66 (3.46) | 4.3581 (16.50) |
| | Syn. | 73.78 (0.51) | 68.20 (5.17) | 49.65 (1.00) | 71.62 (10.90) | 0.79 (0.12) | 5.20 (2.35) | 46.55 (56.27) | 2.0651 (0.82) |

# E  POSSIBLE FUTURE WORK

We summarize several promising future research directions as follows.

**Further Reducing the Runtime.** According to the results in Table 11 and Table 12, the downstream clustering module (i.e., $K$Means with 10 independent runs) consumes most of the runtime of IGP. Although replacing the clustering module with an FC output layer (trained in an E2E way) to directly output the partitioning results can further reduce the total runtime, the downstream clustering module is a key component enables IGP to tackle *online* GP with non-fixed $K$. In contrast, existing E2E GP methods still need to be trained from scratch for new graphs with non-fixed $K$. In our future research, we intend to further reduce the runtime of IGP by exploring a *generic E2E scheme* that can directly derive the GP results (e.g., via an output layer) and tackle the *online* GP with non-fixed $K$. One possible solution is to simultaneously train multiple output layers (with different output dimensionality) to derive multiple partitioning results with respect to all the possible values of $K$. Another possible strategy is to output partitioning results in a hierarchical structure (that covers the results with respect to all the possible values of $K$) based on the idea of hierarchical clustering.

**Dealing with Distribution Shifts.** As discussed in Appendix D.2, our IGP framework has similar motivations with SOTA pre-training methods for graphs (Hu et al., 2020; Qiu et al., 2020). Namely, we conduct the *offline* (pre-)training of *inductive* GNNs (regardless of time cost) but sacrifice some opportunities to fine-tune IGP on new test graphs for a better trade-off between quality and efficiency of the *online* GP. The fine-tuning on new graphs of targeted scenarios can avoid the negative transferring by utilizing validation information from new targeted graphs. Moreover, to select the proper set of pre-training datasets is still an open issue in recent researches regarding the pre-training and fine-tuning scheme. In our future work, we intend to conduct the *offline* (pre-)training of IGP on completely synthetic graphs (that covers properties of various real network systems) and explore an efficient strategy to fine-tune the model on real new system snapshots (of other scenarios) with the guarantee of high partitioning quality and low runtime.

**Scaling Up to Large Graphs.** Scaling GNNs up to graphs with large number of nodes is a significant direction in recent researches, where several sampling strategies are used to split a large graph into multiple subgraphs (Chen et al., 2018; Huang et al., 2018; Zeng et al., 2020), i.e., mini-batches with small number of nodes. Note that IGP is an *inductive* framework across graphs, which can also be trained on and generalized to multiple subgraphs sampled from a large graph (in terms of large number of nodes). Most of existing GNNs that can be scaled up to large graphs usually focus on (semi-)supervised node-level tasks (e.g., node classification) with available node attributes, while we consider challenging *inductive* unsupervised node-level task (i.e., GP) across graphs without graph attributes in this study. In our future work, we intend to further explore efficient (i) feature extraction and (ii) subgraph sampling strategies for IGP on large graphs without available attributes. In particular, we can first split a large graph into multiple small subgraphs via a sampling module and extract the neigh-induced features $\{\mathbf{X}_t\}$ for each sampled graph. IGP can then be applied to multiple sampled graphs and derive graph embedding $\{\mathbf{U}_t\}$ with respect to each subgraph. One possible challenge in this application is to ensure the consistency of embeddings $\{\mathbf{U}_t\}$ of multiple subgraphs sampled from a common large graph (e.g., embeddings of all the subgraphs are mapped into a common latent space) without any auxiliary information given by the node attributes.

**Theoretical Bound Analysis.** Existing theoretical analysis regarding the combinatorial nature of GP (Newman, 2006; Von Luxburg, 2007) is usually based on one single graph. However, we considered the GP across multiple graphs using the *inductive* nature of GNNs. The GP quality on a test graph is not only related to the expressive ability of an inductive GNN, but also related to the similarity between the training and test graphs. To give strict theoretical bound analysis of IGP (e.g., for modularity maximization and NCut minimization) is also a significant direction of in feature work.

We intend to extend the theoretical analysis of GP on one single graph to multiple associated graphs with the consideration of (i) expressive ability of GNNs and (ii) similarity between graphs in terms of their underlying distributions.

**Application to Attributed Graphs.** In this study, we considered *inductive* GP without attributes. Although GP is originally defined only based on graph toplogy (e.g., NCut minimization and modularity maximization), our IGP method can be easily extended to include node attributes by concatenating the reduced neighbor-induced features $\mathbf{Z}_t$ and the attribute matrix of an attributed graph. However, prior research (Qin et al., 2018; Chunaev et al., 2020; Qin & Lei, 2021) has demonstrated that there exist complicated correlations between graph topology and attributes. On the one hand, attributes may provide some complementary information beyond topology that can further improve the quality of some graph inference tasks including GP. On the other hand, *attributes may also carry some noise and mismatched characteristics, which result in unexpected quality declines*, e.g., in terms of modularity and NCut for GP. In our feature work, we intend to consider an *adaptive incorporation scheme* between these two heterogeneous sources for *inductive* GP. Concretely, when attributes carry consistent characteristics with topology, we can fully utilize attributes to improve the GP quality. In contrast, when topology 'mismatches' with attributes, we need to control the contribution of attributes to avoid unexpected quality decline for GP.

