# OpenReview forum: "Trading Quality for Efficiency of Graph Partitioning: An Inductive Method across Graphs"
_ICLR.cc/2022/Conference — ICLR 2022 Submitted_

### Official Review · Reviewer_FSan · 2021-11-02

**Correctness:** 4
**Technical Novelty And Significance:** 3
**Empirical Novelty And Significance:** 3
**Recommendation:** 6
**Confidence:** 3

**Main Review:**

**Strengths**

Both the inductive graph partitioning problem which the paper attempts to solve and the dual GNN adversarial approach are interesting. The authors conduct comprehensive experiments and showcase the usefulness of their method. Even though there are a few notable limitations of the current work (which I will detail below), the authors explicitly state the assumptions and carry out a number of experiments that cover a reasonable range of inductive scenarios.

**Weaknesses / Limitations**

1. The paper does not provide a theoretical guarantee on the generalization to new unseen graphs. It would be nice if a guarantee on modularity or NCut is provided.

2. The proposed framework does not work with attributed graphs. For graph partitioning this might be the usual setting, but the authors should comment on if incorporating node features could be helpful.

3. The authors assume that the set of graphs, which includes both offline training and online testing graphs, follow similar distributions. This assumption easily breaks in practice for graphs that come from different domains. Table 21 in the appendix seems to indicate that, in general, the proposed inductive method may not generalize well due to distribution shifts. On the other hand, I appreciate the authors conduct additional transferring tests, even though the results are mixed, this is a plus since it provides the reader with additional information. For the experiments with LFR synthetic graphs in the main text, what would happen if we increase the range of N, e.g., from 500 to 10,000?

4. The method requires knowing the number of ground-truth clusters K. It may not really be a limitation, as many graph partitioning methods also require K as a parameter.

**Minor issues**

1. In order for (1) and (2) to be equivalent, do we need to perform some rounding algorithm on an optimal solution H_t of (2)?

2. Theorem 1 is trivial, it should be stated as a fact with detailed arguments left to the appendix.

3. The font size in Figure 2 is too small. I cannot read it without zooming in on a screen.

**Summary Of The Paper:**

This paper proposes an inductive GNN-based framework that partitions unattributed graphs with possibly different numbers of nodes and ground-truth clusters. Extensive empirical evaluations using both synthetic and real datasets demonstrate that the new method achieves a better trade-off between quality and efficiency when compared with 15 baselines from 4 categories.

**Summary Of The Review:**

The paper proposes a new GNN-based framework for inductive graph partitioning. The authors conduct extensive experiments that cover a reasonable range of scenarios that showcase the usefulness of the proposed method. My concerns are (1) there is no generalization guarantee, and (2) the method may fail due to distribution shifts, as shown in the experiments.

---

> ### Author Response · Authors · 2021-11-23
> **Responses to Anonymous Reviewer FSan (Cont-3)**
>
> **(6) Organization of 'Theorem 1'**
> >Q: Theorem 1 is trivial, it should be stated as a fact with detailed arguments left to the appendix.
>
> Thank you for your comments regarding 'Theorem 1'. During the revision, we adopted your suggestion to rename 'Theorem 1' as 'Fact 1'. Some arguments regarding 'Fact 1' are given in Appendix 1.
> ***
> **(7) Small font size in some figures**
> >Q: The font size in Figure 2 is too small. I cannot read it without zooming in on a screen.
>
> Thank you for your comments regarding the presentation of some figures. During the revision, we redrew Fig. 1 and Fig. 2, where we highlighted some text in bold and enlarged the font size of most text to be as large as that in the main paper. We hope our revision can help the readers better read the figures.

---

> ### Author Response · Authors · 2021-11-23
> **Responses to Anonymous Reviewer FSan (Cont-2)**
>
> **(3) Hypothesis of similar distributions among multiple graphs**
> >Q: The authors assume that the set of graphs, which includes both offline training and online testing graphs, follow similar distributions. This assumption easily breaks in practice for graphs that come from different domains. Table 21 in the appendix seems to indicate that, in general, the proposed inductive method may not generalize well due to distribution shifts. On the other hand, I appreciate the authors conduct additional transferring tests, even though the results are mixed, this is a plus since it provides the reader with additional information. For the experiments with LFR synthetic graphs in the main text, what would happen if we increase the range of N, e.g., from 500 to 10,000?
>
> We agree with you that our basic assumption (i.e., graphs used in the offline training and online generalization follow similar underlying distributions of a common system or scenario) may break when we relax our constraints to allow graphs come from different domains. From our perspective, although most inductive GNNs do not consider the unsupervised inductive GP across graphs without attributes like our method, they also have some inherent assumptions regarding the domain of graphs for inductive tasks. For instance, GraphSAGE [1] and GAT [2] consider node classification across PPI graphs, where each node has a feature vector with fixed dimensionality. Since each dimension in the feature vector has a clear definition, **most existing inductive GNNs may still inherently assume that all the graphs should be from the same domain** (i.e., PPI of some human tissues), but there are no discussions regarding how to generalize the trained GNNs to other domains (e.g., social networks with different definitions of node attributes from PPI graphs).
>
> As highlighted in Appendix D.2 and E, some recent works regarding pre-training GNNs [3-4] consider how to avoid the negative transferring from one (pre-)training domain to other domains using the pre-training and fine-tuning framework. Different from our IGP framework that focuses on achieving a better trade-off between quality and efficiency of online GP on new graphs, **a fine-tuning procedure with additional optimization algorithms (e.g., gradient descent) should be applied to new graphs in existing pre-training GNNs**, which is time-consuming for the online GP. We gave some discussions regarding the relationship between IGP and pre-training GNNs as well as possible solutions to deal with the negative transferring in Appendix E.
>
> Thank you for your another suggestion to increasing the range of $N$ in LFR-Net. During our revisions, we also **conducted additional robustness analysis for IGP using the LFR-Net** following your suggestions, where we set $N \in [500, 10000]$, $k \in [10, 20]$, $k_{\rm{min}} \in [100, 200]$, $c_{\rm{min}} \in [10, 20]$, $c_{\rm{max}} \in [100, 200]$, and $\mu \in [0.3, 0.6]$. Corresponding results are reported in Appendix C.5.3 of the revised version (see Table 17) where our method still has significant results.
>
> [1] Hamilton, William L., Rex Ying, and Jure Leskovec. "Inductive Representation Learning on Large Graphs." Proc. of NIPS. 2017.
>
> [2] Veličković, Petar, et al. "Graph Attention Networks." Proc. of ICLR. 2017.
>
> [3] Hu, Weihua, et al. "Strategies for Pre-Training Graph Neural Networks." Proc. of ICLR. 2020.
>
> [4] Qiu, Jiezhong, et al. "GCC: Graph Contrastive Coding for Graph Neural Network Pre-Training." Proc of ACM SIGKDD. 2020.
> ***
> **(4) Setting of the number of clusters $K$**
> >Q: The method requires knowing the number of ground-truth clusters K. It may not really be a limitation, as many graph partitioning methods also require K as a parameter.
>
> We agree with you that some GP methods assume the number of clusters $K$ is known. In the main paper, we assume $K$ is given by a real application since we need to ensure the fairness of comparison, where each test graph should be assigned with a common $K$ for all the methods to be evaluated. Please note that we also consider the case when $K$ is not given in Appendix D.1, where our IGP method can still estimate $K$ for a given test graph. It is also the advantage of IGP over some baselines that require the known $K$.
> ***
> **(5) Equivalence of (1) and (2)**
> >Q: In order for (1) and (2) to be equivalent, do we need to perform some rounding algorithm on an optimal solution H_t of (2)?
>
> In (2), the indicator ${\bf{H}}_t$ has the discrete constraint that it can only be assigned with a positive value $[d_i^t vol(C_t^t)^{-1}]^{0.5}$ or 0, otherwise it is not a feasible solution. For a feasible solution to (2), only one entry in each row of ${\bf{H}}_t$ can be positive with all the rest entries as 0. From our perspective, we do not need to round ${\bf{H}}_t$. In the $i$-th row of a feasible indicator ${\bf{H}}_t$, if only the $r$-th entry is positive with all the rest entries in the same row as 0, then the cluster label of node $v_i^t$ is $r$.

---

> ### Author Response · Authors · 2021-11-23
> **Responses to Anonymous Reviewer FSan**
>
> Thank you for your valuable comments and suggestions, which help us further improve the presentation of our paper. We have made some revisions by comprehensively considering comments of all the anonymous reviewers. A summary of the revision can be found in another comment block we have posted. We hope the revised paper and the following explanation can help you better understand the problem we studied and some details of our method. Our responses to your comments are as follows.
> ***
> **(1) Theoretical guarantee of modularity and NCut**
> >Q: The paper does not provide a theoretical guarantee on the generalization to new unseen graphs. It would be nice if a guarantee on modularity or NCut is provided.
>
> We agree with you that the theoretical analysis regarding the bound that our IGP method can achieve will make this work more solid. When we designed our method and wrote the paper, we tried our best to analyze the theoretical bound of IGP for both NCut minimization and modularity maximization. However, we found that to give a strict theoretical guarantee for these two NP-hard combinatorial optimization problems remains a challenge. To the best of our knowledge, existing theoretical analysis regarding the combinatorial nature of GP is usually based on one single graph. In this study, we consider inductive GP across multiple graphs. The GP quality on a graph is not only related to the expressive ability of inductive GNNs, but also related to the similarity between multiple graphs.
>
> Due to the aforementioned reasons, we leave the analysis of theoretical bounds as an important direction of our feature work. During our revision, we **added some discussion regarding this extension in Appendix E**, e.g., extend the theoretical analysis of GP on one single graph to multiple associated graphs with the consideration of (i) the expressive ability of GNN and (ii) similarity between graphs in terms of their underlying distributions.
> ***
> **(2) Application to attributed graphs**
> >Q: The proposed framework does not work with attributed graphs. For graph partitioning, this might be the usual setting, but the authors should comment on if incorporating node features could be helpful.
>
> Our IGP method can be easily extended to include node attributes by concatenating the reduced neighbor-induced features ${\bf{Z}}_t$ with the attribute matrix of an attributed graph. However, **we do not recommend directly incorporating graph attributes using such a naïve strategy**.
>
> GP is a typical unsupervised task that is originally defined only based on graph topology. Some metrics (e.g., modularity and NCut) are also defined only based on graph topology. Some hybrid methods that combine topology and attributes simply assume that attributes can provide complementary information beyond topology which can further improve the quality of some tasks including GP, but **they inherently ignore another possibility that attributes may also carry some noise and mismatched characteristics**. Prior works [1-6] have demonstrated that the simple incorporation of graph attributes (e.g., node features) (i) can sometimes lead to quality improvement but (ii) **may also result in unexpected quality declines** (e.g., in terms of modularity and NCut for GP), which is defined as the **'mismatch' effect** between topology and attributes in some papers [2,3,6].
>
> One can try to **tackle the 'mismatch' effect by adaptively adjusting the contributions of graph topology and attributes**. For instance, when topology 'matches' well with attributes, attributes can be fully utilized to improve the GP quality. In contrast, when topology 'mismatches' with attributes, we need to carefully control the contribution of attributes to avoid unexpected quality decline. We leave this extended application in our future work.
>
> During the revision, we **added some discussions regarding the incorporation of graph attributes in Appendix E**, where we highlighted the aforementioned discussions regarding the 'mismatch' effect.
>
> [1] Newman, Mark EJ, and Aaron Clauset. "Structure and Inference in Annotated Networks." Nature Communications 7.1 (2016): 1-11.
>
> [2] He, Dongxiao, et al. "Joint Identification of Network Communities and Semantics via Integrative Modeling of Network Topologies and Node Contents." Proceedings of AAAI. 2017.
>
> [3] Qin, Meng, et al. "Adaptive Community Detection Incorporating Topology and Content in Social Networks." Knowledge-Based Systems 161 (2018): 342-356.
>
> [4] Chunaev, Petr, Timofey Gradov, and Klavdiya Bochenina. "Community Detection in Node-Attributed Social Networks: How Structure-Attributes Correlation Affects Clustering Quality." Procedia Computer Science 178 (2020): 355-364.
>
> [5] Wang, Xiao, et al. "AM-GCN: Adaptive Multi-Channel Graph Convolutional Networks." Proceedings of ACM SIGKDD. 2020.
>
> [6] Qin, Meng, and Kai Lei. "Dual-Channel Hybrid Community Detection in Attributed Networks." Information Sciences 551 (2021): 146-167.

---

> ### Author Response · Authors · 2021-11-29
> **Authors' follow-up on comments by Anonymous Reviewer FSan**
>
> Dear Reviewer:
>
> Thank you again for your valuable comments! Since the discussion period will end soon, could you please kindly check our responses and revisions? We believe that our responses and revisions have addressed some of your concerns.

---

### Official Review · Reviewer_njQS · 2021-11-02

**Correctness:** 3
**Technical Novelty And Significance:** 2
**Empirical Novelty And Significance:** 2
**Recommendation:** 3
**Confidence:** 3

**Main Review:**

Strengths:

- Performs graph partitioning on multiple graph snapshots while not requiring any form of correspondence of nodes between snapshots.
- Proposed architecture includes a few novel elements.

Weaknesses:

- Presentation of results is extremely poor. Figure 3 is trying to show way too much! It took me several minutes of staring at the figure, zooming in and out, and examining multiple legends to understand what is being shown. Be selective in the results that you show in the main body, and provide the rest in the supplementary. Text in figures and tables is too small.
- ~~Proposed problem setting specifically assumes that there is no node correspondence between snapshots, but then all of the real data experiments involve snapshots over time, which do have node correspondence. Methods that use node correspondence (e.g. incremental and evolutionary clustering) could possibly do better on these datasets.~~
- No motivating application for the no node correspondence setting. Since there is no dataset targeting the proposed setting, a motivating application without data could also provide some justification as to why one should care about this setting. But no such application is present, so it feels more like the authors invented a problem to solve.
- On real data sets, the number of clusters $K$ is not known, so the authors run the Louvain algorithm to choose $K$. But the Louvain algorithm also outputs a set of clusters, which appears to be discarded. ~~Furthermore, the computation time of the Louvain algorithm does not appear to be included, although I can't be sure because the presentation of the results is such a mess.~~

Theorem 1 is trivial, and I'm not sure why it is highlighted as a theorem.

*After discussion period:* I have added a strike through the incorrect portions of my review that have been clarified by the authors.

**Summary Of The Paper:**

The authors consider the problem of inductive graph partitioning, which they formulate as clustering or partitioning multiple snapshots of a time-evolving graph for which we have no node correspondence. In other words, we cannot link the nodes in snapshot $t$ to those in snapshot $t-1$, which prevents incremental or evolutionary clustering algorithms from being applied. The propose a complicated dual graph neural network (GNN) architecture for this problem setting and demonstrate potentially good clustering accuracy with low computation time on simulated and real networks.

**Summary Of The Review:**

The presentation of the results is extremely poor, making empirical results very hard to interpret. I also have concerns regarding the problem setting and evaluation, so I do not view this paper as ready for publication at this time.

---

> ### Author Response · Authors · 2021-11-23
> **Responses to Anonymous Reviewer njQS (Cont-3)**
>
> **(3) Settings of the number of clusters $K$**
> >Q: On real data sets, the number of clusters $K$ is not known, so the authors run the Louvain algorithm to choose $K$. But the Louvain algorithm also outputs a set of clusters, which appears to be discarded. Furthermore, the computation time of the Louvain algorithm does not appear to be included, although I can't be sure because the presentation of the results is such a mess.
>
> In the main paper, we consider GP where the number of clusters $K$ is known. Namely, **$K$ is assumed to be already given by a real application on each graph and we do not need to consider how $K$ is estimated**. Only in this setting, we can include some of the baselines (e.g., SNMF) in our experiments.The anonymous reviewer FSan also agrees with us that some GP methods have this assumption regarding the known $K$. In Appendix D.1, we further demonstrate that **our IGP method can still estimate $K$ (i.e., model selection) and give corresponding GP results when $K$ is not given**.
>
> Please note that we use Louvain to estimate $K$ for each real graph in the main paper due to the following reasons.
>
> First, the real datasets in our experiments do not provide the ground-truth of $K$. We follow prior works [1-3] to use Louvain to estimate $K$ for each graph. **Using an auxiliary model selection method (e.g., Louvain, hierarchical clustering, etc.) to estimate $K$**, while **evaluating the GP results of other baselines w.r.t. the estimated $K$** (i.e., even with GP results of the model selection method being discarded) is **a common strategy used in some prior works [1-3] for the case when $K$ is unknown**.
>
> Second, for some baselines in our experiments (e.g., SNMF, GAP, ClusNet, etc.), $K$ must be given for each graph, otherwise they cannot derive the GP results for evaluation. As the real datasets do not provide ground-truth of $K$, we have to use some auxiliary model selection methods (e.g., Louvain) to determine $K$ for these baselines.
>
> Third, we must ensure the fairness of the quantitative comparison where **each graph should be assigned with a common $K$ for all the methods to be evaluated**. Although some of the methods can automatically estimate $K$ (e.g., our IGP method as demonstrated in Appendix D.1), different methods may have different estimated $K$ values, which undermines the fairness of comparison for some metrics.
>
> On a test graph, based on the common $K$ given by Louvain, each method that is being evaluated can output its own GP result. In this case, Louvain is applied to all the methods in our experiments. Hence, **discarding the GP results of Louvain and whether including the runtime of Louvain will not undermine the fairness of comparison**.
>
> In summary, using an auxiliary model selection method (e.g., Louvain) to estimate $K$ when it is unknown is a common strategy in some prior works. The usage of Louvain will not undermine the fairness of comparison. In the main paper, we also assume $K$ is already given by the application, where we do not need to consider how $K$ is estimated. Some GP methods also has the similar assumption.
>
> [1] Wang, Xiao, et al. "Semantic Community Identification in Large Attribute Networks." Proc. of AAAI. Vol. 30. No. 1. 2016.
>
> [2] He, Dongxiao, et al. "Joint Identification of Network Communities and Semantics via Integrative Modeling of Network Topologies and Node Contents." Proc. of AAAI. 2017.
>
> [3] Qin, Meng, et al. "Adaptive Community Detection Incorporating Topology and Content in Social Networks." Knowledge-Based Systems 161 (2018): 342-356.
>
> ***
> **(4) Organization of ‘Theorem 1’**
> >Q: Theorem 1 is trivial, and I'm not sure why it is highlighted as a theorem.
>
> Thank you for your comments regarding 'Theorem 1'. During our revision, we adopted the suggestion of another reviewer to rename 'Theorem 1' as 'Fact 1'. Some arguments (e.g., the proof) regarding 'Fact 1' are given in Appendix 1.

---

> ### Author Response · Authors · 2021-11-23
> **Responses to Anonymous Reviewer njQS (Cont-2)**
>
> **(2) Assumption of node correspondence**
> >Q: Proposed problem setting specifically assumes that there is no node correspondence between snapshots, but then all of the real data experiments involve snapshots over time, which do have node correspondence. Methods that use node correspondence (e.g. incremental and evolutionary clustering) could possibly do better on these datasets.
>
> >No motivating application for the no node correspondence setting. Since there is no dataset targeting the proposed setting, a motivating application without data could also provide some justification as to why one should care about this setting. But no such application is present, so it feels more like the authors invented a problem to solve.
>
> Please note that **NOT all the real datasets in our experiments have node correspondence**. As introduced in Appendix C.2, the Reddit dataset is extracted from multiple online discussion threads of Reddit, with each graph describing the interaction between users in a thread. It is very common that two discussion threads in social media have different sets of users (nodes), i.e., there is no node correspondence between two graphs. In addition to snapshots evolving over time, the multiple ‘graphs’ defined in Section 2 can also be **a set of independent (sub)graphs of a large system or a scenario** (e.g., discussion threads from a social media and multiple ego-nets in a social network) that are disconnected and have no node correspondence.
>
> The Reddit dataset in our experiment provides a motivating application for the no node correspondence setting. We also find some other motivating applications regarding your concerns. For instance, authors of [1] train a GNN-based GP model on a set of TensorFlow neural networks (e.g., VGG) and generalize the model to other neural networks (e.g., ResNet), where there is no node correspondence between different neural networks (each operation in a neural network is abstracted as a node). In summary, the multiple graphs from a system or scenario can be (i) **snapshots evolving over time** or (ii) **independent (sub)graphs of a scenario without temporal dependency** (e.g., multiple ego-nets in social media). Since there may be no temporal dependency in some cases, we do not consider the node correspondence in this study. Hence, **our IGP method can be applied to both cases with and without node correspondence, which is more generic than conventional incremental GP methods**.
>
> Moreover, to **enable IGP to be transferred from one scenario to other different scenarios** is also the reason why we do not consider node correspondence. As demonstrated in Appendix D.2, we conducted the offline training of IGP on synthetic graphs and generalize it to real graphs, where there is no node correspondence between a synthetic graph and a real graph. Some pre-training GNN methods [2-3] (with offline pre-training some scenarios and additional fine-tuning on other different scenarios) have similar motivations with such an extended application in Appendix D.2. We also discuss the relationship and difference between IGP and SOTA pre-training GNN methods in Appendix D.2 and Appendix E.
>
> During our revision, we adopted your suggestions to **add some discussions and examples regarding our no node correspondence settings in the very beginning of Section 1 and 2**. Moreover, we also **evaluated the quality and efficiency of some incremental GP and graph embedding methods (e.g., IncNSA [4] and TIMERS [5]) on some real datasets with node correspondence** (e.g., Taxi and AS). The evaluation results and trade-off analysis are shown in Appendix C.5 (see Tables 8, 10, 12, 15, and 16), where our IGP method can still achieve a significant trade-off between quality and efficiency beyond these incremental GP baselines.
>
> [1] Nazi, Azade, et al. "A Deep Learning Framework for Graph Partitioning." Proc. of ICLR. 2019.
>
> [2] Hu, Weihua, et al. "Strategies for Pre-Training Graph Neural Networks." Proc. of ICLR. 2020.
>
> [3] Qiu, Jiezhong, et al. "GCC: Graph Contrastive Coding for Graph Neural Network Pre-Training." Proc. of ACM SIGKDD. 2020.
>
> [4] Su, Xing, et al. "IncNSA: Detecting Communities Incrementally from Time-Evolving Networks Based on Node Similarity." International Journal of Modern Physics C 31.07 (2020): 2050094.
>
> [5] Zhang, Ziwei, et al. "TIMERS: Error-bounded SVD Restart on Dynamic Networks." Proc. of AAAI. 2018.

---

> ### Author Response · Authors · 2021-11-23
> **Responses to Anonymous Reviewer njQS**
>
> Thank you for your valuable comments and suggestions that help improve the presentation of our paper. We have made some revisions based on the comments of all the anonymous reviewers. A summary of the revision can be found in another comment block we have posted. We really hope that our revision and the following explanation can help you better read the paper and understand the assumption as well as constraints of our method. Our responses to your comments are as follows.
> ***
> **(1) Presentation of experiment results**
> >Q: Presentation of results is extremely poor. Figure 3 is trying to show way too much! It took me several minutes of staring at the figure, zooming in and out, and examining multiple legends to understand what is being shown. Be selective in the results that you show in the main body, and provide the rest in the supplementary. Text in figures and tables is too small.
>
> Thank you for your comments regarding the presentation of some figures and tables. During our revision, we **redrew some of the figures**, where we **enlarged the font size and highlighted the text in bold**. Especially, we followed the informative visualization presentation in [1] that presents the trade-off between two aspects to redraw Fig. 3, in which we **removed all the legends and directly denoted each data point using text**. Moreover, we also adopted your suggestions to **show some of the visualization results in Fig. 3** (e.g., the trade-off between NMI/modularity and runtime) and **put the rest in Appendix C.5**. In addition, we also **enlarged the tables** and tried to **make the font size as large as that in the main paper**. We hope that our revision can help the readers better read the figures and tables.
>
> [1] Chen, Ting, et al. "A Simple Framework for Contrastive Learning of Visual Representations." Proc. of ICML. PMLR, 2020.

---

> ### Author Response · Authors · 2021-11-29
> **Authors' follow-up on comments by Anonymous Reviewer njQS**
>
> Dear Reviewer:
>
> Thank you again for your valuable comments! Since the discussion period will end soon, could you please kindly check our responses and revisions? We believe that our responses and revisions have addressed some of your concerns.

---

> > ### Comment · Reviewer_njQS · 2021-11-29
> > **Thanks for the clarifications!**
> >
> > I have updated my review to strike out the incorrect portions that the authors have addressed. The presentation is slightly improved also.

---

> > > ### Author Response · Authors · 2021-11-29
> > > **Further response to the update of Reviewer njQS**
> > >
> > > Thank you for your update.
> > >
> > > For your 3rd concerns regarding the node correspondence, on the premise of obeying the page limit (i.e, 9 pages for the main paper), we have tried our best to **add a simple motivating application** for the no node correspondence setting **in 5th paragraph of Section 1**, i.e., conducting offline training on known ego-nets of a social network and generalize the model to new unseen ego-nets for fast online GP, where different ego-nets usually do not have node correspondence. Such a motivating example is also highlighted **in the 1st paragraph of Section 2**. In our experiment, **Reddit** (with multiple independent discussion threads of social media) **is a dataset without node correspondence**. To some extent, the **transferring test** (i.e., offline training on synthetic graphs and online generalization to real graphs) **demonstrated in Appendix D.2 can also be a motivating application** for this setting.
> > >
> > > In the 1st paragraph of Section 4, we have tried our best to add some reasons for why we use Louvain to estimate $K$.
> > >
> > > We also plan to further improve the presentation of experiment results in Fig. 1 and Fig. 4, which can be updated in our final version of the manuscript. Concretely, we intend to only present our methods (i.e., IGP-M and IGP-C) in bold, which can make it much easier to distinguish our methods from other baselines. Moreover, we also intend to remove some overlapping data points to obtain a clearer presentation.
> > >
> > > **Thanks again for your insight comments. We hope that you could update your evaluation if we have addressed your concerns.**

---

### Official Review · Reviewer_BgbP · 2021-11-06

**Correctness:** 4
**Technical Novelty And Significance:** 3
**Empirical Novelty And Significance:** 3
**Recommendation:** 5
**Confidence:** 3

**Main Review:**

The paper has various strong points, including an extensive experimental evaluation with numerous other baselines,

- What is the advantage that the dual GNN offers over other choices in prior work (Pan et al. 2018, Nazi et al. 2019)?

- Under what conditions on the training dataset does the framework work well? Is it easy to test those conditions? E.g., from equation (11) that encodes a "k-cut size" it seems that you implicitly assume small changes from snapshot to snapshot?  Is this what you mean by saying that the graphs are snapshots of a given complex system?

- Can you please elaborate on how the uncoarsening is done to get the actual community participation of the query graph G?

- Having an axis with >T secs is not very informative for the proposed method. Can you please be more specific about the runtimes?

- Can you test your method on communities with community groundtruth?

- For the synthetic experiments using the stochastic blockmodel, it would have been nice to see a comparison with the paper  "Supervised community detection with line graph neural networks" by Chen et al. that provides state-of-the-art results.



**Summary Of The Paper:**

In this work the authors focus on an important problem, building an inductive framework for graph partitioning. This is a major problem with the potential of improving the performance of various classic transductive  algorithms for graph partitioning, both in terms of output quality, but also in terms of speed when a new snapshot  of a system needs to be partitioned into communities. This is a recent line of research, see, e.g., Nazi et al. 2019 "GAP: Generalizable Approximate Graph Partitioning Framework".  The proposed framework can address snapshots with differing number of nodes, by projecting them down to coarsened versions of the network. There are two versions of the framework, based on normalized cut and modularity objectives respectively. The framework can be potentially used with other measures as well. A nice idea is to leverage the unsupervised communities extracted from modularity optimization, or spectral algorithms through a dual GNN structure, that can then be used to partition fast unseen instances.

**Summary Of The Review:**

The paper has various strong points, but the write-up can be improved. I found overall the paper hard to read; e.g., the choice of the dual GNN is not well justified (what issues is it resolving from GAP), reading the theorem 1 was kind of confusing, as the statement feels more as an observation derived from the definition of a partition etc.  I also found the term of permutation invariant for labels (0,1,1) vs (1,0,0) a bit confusing, as up to that point, I was expecting labels to be indicator vectors of which community a node participates in.

---

> ### Author Response · Authors · 2021-11-23
> **Responses to Anonymous Reviewer BgbP (Cont-4)**
>
> **(6) Comparison with (Chen et al. 2019) ‘Supervised community detection with line graph neural networks’**
> >Q: For the synthetic experiments using the stochastic blockmodel, it would have been nice to see a comparison with the paper "Supervised community detection with line graph neural networks" by Chen et al. that provides state-of-the-art results.
>
> Thank you for your suggestions. We also found that (Chen et al. 2019) can be a good baseline. It is a typical E2E GP method based on inductive GNNs. Although it follows a similar training and generalization paradigm to our IGP method, **it can only tackle the inductive GP across graphs with fixed number of clusters $K$**. Hence, we can only apply this baseline to datasets with fixed $K$ (i.e., on the GN-Net as you suggested).
>
> Although this method can also utilize the permutation invariant GP results (‘ground-truth’) of training data to train the model (like our IGP method), its training loss (i.e., Eq. (3) in (Chen et al. 2019)) should select the ‘best’ mapping from its derived GP result to the ‘ground-truth’ among all the possible $O(K!)$ cases, which can only be used on graphs with small $K$, e.g., $K=\{2, 5\}$ in (Chen et al. 2019). **For the GN-Net in our experiments with $K=250$, this baseline should select the ‘best’ mapping among $O(250!)$ cases, which is extremely time-consuming!** Although the authors of (Chen et al. 2019) give some discussion about how to deal with large $K$ (e.g., randomly partition $K$ clusters into several groups), they do not validate its effectiveness in their experiments and do not provide the code of such extension in their open-source implementation. Moreover, **the topology input of this baseline also includes ${\bf{A}}^{J}$ and ${\bf{B}}^{J}$ with $J={0, 1, 2}$, which are also time-consuming to derive**.
>
> In contrast, **our IGP method can tackle the inductive GP with non-fixed $N$ and $K$**. **The feature extraction procedure is also much more efficient than (Chen et al. 2019)**. More importantly, our IGP method **can incorporate the permutation invariant training ‘ground-truth’ by combining its dual GNN structure with some classic GP objectives, which is also much more efficient than (Chen et al. 2019)**, i.e., we do not need to select the ‘best’ mapping from all $O(K!)$ cases.
>
> In our current experiments, directly applying the open-source implementation of (Chen et al. 2019) to the GN-Net is very easy to result in the out-of-memory exception and is so time-consuming for the offline training that we cannot give the final evaluation results by the revision period. We can include the evaluation results in our final version once we get the evaluation results of (Chen et al. 2019).
> ***
> **(7) Examples of the permutation invariant label assignments**
> >Q: I also found the term of permutation invariant for labels (0,1,1) vs (1,0,0) a bit confusing, as up to that point, I was expecting labels to be indicator vectors of which community a node participates in.
>
> Thanks for your suggestions. From our perspective, different GP objectives may have different definitions of the indicator vectors, e.g., NCut minimization in (2) and modularity maximization in (4). To make examples regarding the permutation invariance clearer, we have replaced the label sequences ‘(0, 1, 1) and (1, 0, 0)’ in Section 1 with the clustering membership assignments $(l_1, l_2, l_3)=(1, 2, 2)$ and $(l_1, l_2, l_3)=(2, 1, 1)$, where $l_i$ is defined as the label assignment of node $v_i$. The permutation invariance between the two assignments indicates that there exists a ‘perfect’ label mapping from $(l_1, l_2, l_3)=(1, 2, 2)$ to $(l_1, l_2, l_3)=(2, 1, 1)$, e.g., label ‘1’ in the former case is mapped to ‘2’ in the latter case. Namely, $(l_1, l_2, l_3)=(1, 2, 2)$ and $(l_1, l_2, l_3)=(2, 1, 1)$ are the same clustering membership assignment for GP.

---

> ### Author Response · Authors · 2021-11-23
> **Responses to Anonymous Reviewer BgbP (Cont-3)**
>
> **(4) Presentation of the runtime**
> >Q: Having an axis with >T secs is not very informative for the proposed method. Can you please be more specific about the runtimes?
>
> We agree with you that an axis with '>T secs' is not very informative but it is a compromise between informative presentation and the limit of 9 pages for main paper. In our experiments, results of 17 methods are reported. The runtime of some baselines is very large compared with other baselines and our methods. If we exactly present the distribution of the runtime of all the method (without using ‘>T secs’), we must significantly enlarge each subfigure in Fig. 3 and 4, otherwise it will make Fig. 3 and 4 hard to read (e.g., data points of IGP-M and IGP-C overlaps with other points) and cannot highlight the fast runtime of our method beyond other baselines. If we further enlarge the figures, the main paper will be over-length. **Please note that we also gave the concrete number records of runtime for each method in Appendix C.5 (see Tables 11 and 12)**. We hope that you can understand our difficulty in making the informative presentation while obeying the page limit.
>
> **The ‘runtime’ in our experiments refers to the total time of a method to derive its final GP result in terms of seconds**. The total runtime of IGP includes the time of (i) efficient feature extraction to derive ${\bf{Z}}_t$, (ii) one feedforward propagation through the GNN to derive graph embedding ${\bf{U}}_t$, and (iii) 10 independent runs of KMeans. We also reported the runtime of these three parts in Tables 11 and 12.
> ***
> **(5) Test on communities with ground-truth**
> >Q: Can you test your method on communities with community ground-truth?
>
> Please note that the LFR-Net (with 4 datasets) and GN-Net (with 3 datasets) in our experiments are widely-used synthetic benchmarks with ground-truth for community detection and GP. We also use metrics of NMI and AC on these synthetic benchmarks. **To some extent, we have tested our method on some benchmarks with ground-truth.**
>
> We also tried to include some real datasets with ground-truth. Since we consider inductive GP across graphs, we should **select real datasets that have multiple associated graphs from a common system or scenario**. Moreover, as we consider the case without attributes, we should **include real datasets that do not have node attributes**.
>
> Please note that we do not consider GP on attributed graphs because some prior work [1-6] has demonstrated that there may be complicated correlations between graph topology and attributes. The incorporation of attributes may further improve GP quality in some cases, but may also result in quality decline due to the mismatched characteristics and noise of attributes. For some real attributed graph datasets with ground-truth (e.g., PPI in [7]), **it is unclear whether the ground-truth is dominated by topology or attributes**.
>
> Unfortunately, we have not found some suitable open-source real datasets with ground-truth due to the aforementioned constraints. To the best of our knowledge, some related work [8-9] has similar settings to our experiments, where they also use real datasets without ground-truth and evaluate quality via some unsupervised metrics (e.g., NCut). Although [10] uses real datasets with their ground-truth derived from the overlapping communities of some large datasets in SNAP, all the snapshots are very small (e.g., with average 91 nodes) and we cannot find the pre-processed dataset from the official open-source implementation of [10].
>
> [1] Newman, Mark EJ, and Aaron Clauset. "Structure and Inference in Annotated Networks." Nature Communications 7.1 (2016): 1-11.
>
> [2] He, Dongxiao, et al. "Joint Identification of Network Communities and Semantics via Integrative Modeling of Network Topologies and Node Contents." Proceedings of AAAI. 2017.
>
> [3] Qin, Meng, et al. "Adaptive Community Detection Incorporating Topology and Content in Social Networks." Knowledge-Based Systems 161 (2018): 342-356.
>
> [4] Chunaev, Petr, Timofey Gradov, and Klavdiya Bochenina. "Community Detection in Node-Attributed Social Networks: How Structure-Attributes Correlation Affects Clustering Quality." Procedia Computer Science 178 (2020): 355-364.
>
> [5] Wang, Xiao, et al. "AM-GCN: Adaptive Multi-Channel Graph Convolutional Networks." Proc. ACM SIGKDD. 2020.
>
> [6] Qin, Meng, and Kai Lei. "Dual-Channel Hybrid Community Detection in Attributed Networks." Information Sciences 551 (2021): 146-167.
>
> [7] Hamilton, William L., Rex Ying, and Jure Leskovec. "Inductive representation learning on large graphs." Proc. of NIPS. 2017.
>
> [8] Nazi, Azade, et al. "GAP: Generalizable Approximate Graph Partitioning Framework." arXiv preprint arXiv:1903.00614 (2019).
>
> [9] Nazi, Azade, et al. "A Deep Learning Framework for Graph Partitioning." (2019).
>
> [10] Chen, Zhengdao, Xiang Li, and Joan Bruna. "Supervised Community Detection with Line Graph Neural Networks." Proc. of ICLR. 2017.

---

> ### Author Response · Authors · 2021-11-23
> **Responses to Anonymous Reviewer BgbP (Cont-2)**
>
> **(2) Conditions of the proposed method**
> >Q: Under what conditions on the training dataset does the framework work well? Is it easy to test those conditions? E.g., from equation (11) that encodes a "k-cut size" it seems that you implicitly assume small changes from snapshot to snapshot? Is this what you mean by saying that the graphs are snapshots of a given complex system?
>
> Please note that we highlighted our basic hypothesis in Section 2. Namely, we assume that **snapshots in the training, validation, and test sets are independently generated via several underlying distributions of a common system or scenario**. Since we adopt a similar learning paradigm with supervised learning (i.e., train and validate a model on historical known graphs and generalize the model to new graphs), **the condition that whether the given snapshots from a system or scenario share similar properties (e.g., in terms of underlying distributions) and thus deriving high-quality GP results can be tested using the validation set** as we usually do in supervised learning.
>
> The assumption of ‘small changes from snapshot to snapshot’ is usually adopted in some incremental GP methods (but not in our method), where there should be only small changes in the node sets and edges sets of multiple snapshots. In contrast, the hypothesis of our IGP method, i.e., graphs are independently generated via several underlying distributions, indicates that **different graphs can have entirely different node sets and thus different edge sets**. Moreover, most incremental GP methods assume that there must be node correspondence among multiple graphs. In this study, the multiple graphs can be (i) associated snapshots evolving over time with node correspondence or (ii) independent (sub)graphs of a scenario without node dependency e.g., multiple ego-nets in a social network. In particular, we do not consider the node correspondence among graphs, so **our IGP method can be applied to both cases with and without node correspondence**, which is more generic than existing incremental GP methods.
>
> IGP works well when multiple graphs (e.g., from a system or scenario) share some similar underlying distributions. To some extent, we assume that **there are small differences among the underlying distributions of different graphs**, but there are no specific constraints on the changes of their node sets and edge sets. In summary, we do not simply assume ‘small changes from snapshot to snapshot’ but consider a more generic hypothesis than existing incremental GP methods.
>
> In addition to the aforementioned hypothesis regarding similar underlying distributions of a system or scenario, we also test the potential of IGP to transfer from one training scenario to other different scenarios. In Appendix D.2, we consider the application that we conducted offline training of IGP on synthetic graphs and generalized the model to real graphs. We also give some discussion regarding how to deal with the negative transferring in Appendix E.
>
> ***
> **(3) How the uncoarsening is done?**
> >Q: Can you please elaborate on how the uncoarsening is done to get the actual community participation of the query graph G?
>
> Please note that our IGP method does not follow the conventional coarsening-uncoarsening framework of prior GP methods, e.g., Metis, hMetis, GraClus, MILE, etc. In contrast, **there is no uncoarsening step in IGP** and we only use the efficient HEM coarsening to extract informative feature input of the dual GNN structure. As IGP follows a novel inductive graph embedding across graphs, we can **get the final GP result by applying a downstream clustering algorithm (e.g., KMeans in our experiments) to the derived embedding** (i.e., low-dimensional vector representations). You can also check the overall procedures of the offline training and online generalization of IGP in Appendix B (see Algorithm 1, 2, 3, and 4).

---

> ### Author Response · Authors · 2021-11-23
> **Responses to Anonymous Reviewer BgbP**
>
> Thank you for your valuable comments and suggestions that help us achieve a better presentation of our paper. By comprehensively considering the comments of all the anonymous reviewers, we have made some revisions. A summary of the revision can be found in another comment block we have posted. We hope that our revision and the following explanation can help you better understand some details of this paper. Our responses to your comments are as follows.
> ***
> **(1) Advantages of the proposed method beyond prior work (Pan et al. 2018, Nazi et al. 2019)**
> >Q: What is the advantage that the dual GNN offers over other choices in prior work (Pan et al. 2018, Nazi et al. 2019)?
>
> In (Pan et al. 2018), the authors proposed two graph embedding methods (i.e., ARGA and ARVGA) based on GCN and AAE. They use the adversarial process between graph embedding (derived by GNN) and pre-define probabilistic distribution to regularize the embedding inference. Different from ARGA and ARVGA, our dual GNN structure is not a simple application of AAE. In particular, the dual GNN structure is introduced to **capture the permutation invariant training ‘ground-truth’** via the **adversarial process** between (1) graph embeddings w.r.t. **original graphs** and (2) auxiliary label-induced embeddings w.r.t. **label-induced graphs**, where the **label-induced graph is our original design**. ARGA and ARVGA cannot capture the permutation invariant training ‘ground-truth’ of historical graphs.
>
> Moreover, (Pan et al. 2018) only demonstrates some transductive applications but does not consider inductive GP across graphs. The transductiveness of ARGA and ARVGA indicates that they need to be optimized from scratch for each input graph, which is time-consuming for the online GP on new test graphs. In addition, ARGA and ARVGA are originally designed for attributed graphs, where each node has a feature vector. There are no discussions in (Pan et al. 2018) about how to apply ARGA and ARVGA to graphs without attributes. Our experiments indicate that some standard settings of GNN for graphs without attributes (e.g., use a constant matrix as the feature input of GNN) may still suffer from poor GP quality. In contrast, **our IGP method follows an inductive graph embedding scheme, which is more efficient than ARGA and ARVGA, to tackle GP on multiple graphs without attributes**.
>
> GAP (Nazi et al. 2019) is a typical end-to-end GP method based on inductive GNNs, with the final GP results derived by an FC output layer. Although it can tackle the inductive GP across graphs like our IGP method, it can only be generalized to new graphs with fixed number of clusters $K$, due to the fixed dimensionality of its output layer. For new test graphs with non-fixed $K$, we still need to optimize GAP from scratch (i.e., we cannot use its fast generalization), which is time-consuming for the online GP on new test graphs. Moreover, as recommended in (Nazi et al. 2019), for graphs without attributes, one needs to use PCA to map each adjacency matrix (with non-fixed number of nodes $N$) to a latent feature space with fixed dimensionality, but PCA is usually time-consuming.
>
> In contrast, our IGP method follows an inductive graph embedding scheme, with GP results derived via a downstream clustering module. **It enables IGP to tackle the inductive GP with non-fixed $K$**. In addition, we also introduce a novel feature extraction module based on HEM coarsening, which **enables IGP to tackle the inductive GP with non-fixed $N$ and is much more efficient than the PCA used in GAP**. Our experiment results also demonstrate that the online generalization of IGP is more robust than that of GAP on datasets with fixed $K$. In particular, **IGP can capture the permutation invariant ‘ground-truth’ of the training data**, while (Nazi et al. 2019) does not consider the utilization of such ‘ground-truth’.
>
> In summary, **our IGP is more generic, more efficient, and more robust than GAP**.

---

> ### Author Response · Authors · 2021-11-29
> **Authors' follow-up on comments by Anonymous Reviewer BgbP**
>
> Dear Reviewer:
>
> Thank you again for your valuable comments! Since the discussion period will end soon, could you please kindly check our responses and revisions? We believe that our responses and revisions have addressed some of your concerns.

---

### Official Review · Reviewer_7xoz · 2021-11-07

**Correctness:** 3
**Technical Novelty And Significance:** 3
**Empirical Novelty And Significance:** 3
**Recommendation:** 5
**Confidence:** 3

**Main Review:**

The strengths of this paper are as follows.

+ The problem consider is important and practical. Graph partitioning is an important pre-processing step in many applications. Moreover the model of being sampled i.i.d. has practical relevance and also considered in many prominent works in the graph networks literature.

+ For the most part, the paper is written well, with detailed explanantion of the methodology and the results. The experiments are also exhaustive and sound. They considered an exhaustive list of baselines, and show that they perform nearly well or better than most methods on the range of tasks considered.

+ They use publicly available datasets and the results can largely be reproduced.

The weakness of this paper is as follows.

- The paper does not do a good job of crisply characterizing how their methods differ from prior works. The reasons why their method is better, that the authors state, seem somewhat shallow/arbitrary. They claim that computing gradients etc are expensive/time-consuming. Yet their method is based on adversarial auto-encoders, which need to be trained via gradient based optimization methods. So it seems like this is not a real difference. The only difference I can see is that, this paper considers a different NN architecture and training method compared to some of the other GNN based methods (e.g., Hamilton et al.,). I would like the author's to be a bit more precise and scientific & also credit prior works better.

- Along the lines of (1) it is not entirely clear to me why this method works. Adding intuition about some of the choices would make it much better. At the moment it seems like, they created a new method, threw it at the datasets, and it happened to work. Some more principled study on why it works would greatly help the paper.

- The paper does not report CIs in the experiment. Some of the points are so close to each other, that without CIs its unclear if its real improvement or just noise. This is important, since the thrust of the claims made by this paper is in the experiments, and having rigoros statistical analysis gives more confidence in the results.

**Summary Of The Paper:**

This paper considers the problem of solving the graph partitioning problem repeatedly over many different graphs. Graphs are sampled i.i.d. from an unknown but fixed distribution and the goal of the algorithm is to solve the GP problem on each of the graphs. There are two quantities that are opposing; efficiency (on one hand we can solve an NP-hard problem each time but would be prohibitively time consuming) and quality (on the other hand we can generalize from the learnings on the other graph since they are related via the i.i.d. distribution). This paper proposes a NN architecture that uses a subset of the graphs in the family to learn an embedding, and then derives the solution to the other instances by using this embedding via a matrix multiplication. Using a number of simulated and real-world datasets they show that this method works well empirically.

**Summary Of The Review:**

My main review is based on the weaknesses I stated above. I am not totally confident about the importance/merit of the approach provided in this paper. Thus, I am giving a weak reject.

---

> ### Author Response · Authors · 2021-11-23
> **Responses to Anonymous Reviewer 7xoz (Cont-3)**
>
> ***
> **(3) Report of CIs**
> >Q: The paper does not report CIs in the experiment. Some of the points are so close to each other, that without CIs it’s unclear if it’s the real improvement or just noise. This is important, since the thrust of the claims made by this paper is in the experiments, and having rigorous statistical analysis gives more confidence in the results.
>
>   We agree with you that CIs can better validate the effectiveness of our method. We found that directly presenting CIs in the visualization results in Fig. 3 and 4 may make the figures hard to read, e.g., overlapping of some intervals and data points. Hence, in Appendix C.5, we reported the mean and standard deviation of each metric (i.e., NMI, AC, modularity, NCut, and runtime) on the test set of each dataset (see Tables 5-12). The test set of each dataset consists of multiple graphs, with each graph having its corresponding quality and efficiency metrics (e.g., NMI, AC, modularity, NCut, and runtime). The evaluation results in Fig. 3 and Tables 5-12 correspond to one independent run of a method on the test set, with all the methods validated by the validation set. In summary, **in each independent run of a method on a test set, we have a mean and a standard deviation for each metric over multiple snapshots in this test set**.
>
> The report of CI needs multiple independent runs of a method, in which we need to record the mean and standard deviation of the ‘average’ metrics on the test set over multiple runs. Thus, **CIs in our settings have much more complicated definitions than we usually report in other tasks** (e.g., graph classification). Moreover, we also found that it is extremely time-consuming to run all the methods (i.e., more than 20 methods in our experiments) multiple times on all the 11 datasets and compute CIs. Hence, we cannot report CIs for all the methods before the revision period. To some extent, **we believe that the mean and standard deviation values reported in Tables 5-12 can provide some information for your concerns**.

---

> ### Author Response · Authors · 2021-11-23
> **Responses to Anonymous Reviewer 7xoz (Cont-2)**
>
> **(2) Why does the proposed method work?**
> >Q: Along the lines of (1) it is not entirely clear to me why this method works. Adding intuition about some of the choices would make it much better. At the moment it seems like, they created a new method, threw it at the datasets, and it happened to work. Some more principled study on why it works would greatly help the paper.
>
> Please note that Eq. (1) is preliminary regarding the combinatorial nature of GP and NCut metric. **It is not directly related to why our IGP method works**, since we do not directly use the original NCut minimization objective in our method. If you are interested in NCut minimization, you can refer to [1], which gives some details regarding the intuition of using NCut minimization to tackle GP and why some relaxed approximation algorithms for NCut minimization (e.g, spectral clustering) work.
>
> Please also note that our IGP method follows an **inductive** unsupervised node-level framework across graphs. Since **inductive** node classification, a supervised node-level task, is widely studied in a series of works regarding inductive GNNs [2-3] (i.e., why they work well for inductive node classification has been validated by some prior works), **the basic intuition of our IGP method is to generalize inductive framework from supervised node-level tasks (e.g., inductive node classification in [2-3]) to unsupervised node-level tasks (e.g., inductive GP across graphs in this paper)**. There are also related works regarding the inductive GP across graphs [4-6] (i.e., with similar motivation to our IGP method), which **preliminarily verify the effectiveness of the direction considered in our paper and demonstrate why it works**.
>
> **Some of our original designs enable IGP to have several advantages beyond these SOTA inductive methods [4-6] with clear intuitions**. For instance, based on the **intuition of coarsening a graph with non-fixed number of nodes $N$ into a supergraph with fixed number of supernodes**, the HEM feature extraction module enables IGP to tackle GP on graphs with non-fixed $N$. Furthermore, **Fact 1 (page 4) also gives the intuition of the dual GNN structure**, which help IGP capture the permutation invariant label information of training data by controlling the message passing of GNN in a set of **auxiliary label-induced graphs*.*
>
> Moreover, conducting the ablation study is a widely-used strategy to validate the effectiveness of each component of a method, i.e., **illustrating why a method works**. We also conducted ablation studies regarding some components of IGP in Section 4 and Appendix C.5.4. The ablution studies also demonstrate that all the considered components are essential to ensure the high GP quality of IGP.
>
> During our revision, we comprehensively adopted suggestions from all the reviewers to **add some brief discussions and examples regarding our intuitions, assumptions and constraints in Section 1, 2 and 3, while obeying the limit of 9 pages of the main paper**. We hope that the aforementioned explanation and our revision can help you better understand why our method works.
>
> [1] Von Luxburg, Ulrike. "A Tutorial on Spectral Clustering." Statistics and Computing 17.4 (2007): 395-416.
>
> [2] Hamilton, William L., Rex Ying, and Jure Leskovec. "Inductive representation learning on large graphs." Proc. of NIPS. 2017.
>
> [3] Veličković, Petar, et al. "Graph Attention Networks." Proc. of ICLR. 2017.
>
> [4] Nazi, Azade, et al. "GAP: Generalizable Approximate Graph Partitioning Framework." arXiv preprint arXiv:1903.00614 (2019).
>
> [5] Nazi, Azade, et al. "A Deep Learning Framework for Graph Partitioning." (2019).
>
> [6] Chen, Zhengdao, Xiang Li, and Joan Bruna. "Supervised Community Detection with Line Graph Neural Networks." arXiv preprint arXiv:1705.08415 (2017).

---

> ### Author Response · Authors · 2021-11-23
> **Responses to Anonymous Reviewer 7xoz**
>
> Thank you for your valuable comments and suggestions that point out some presentation problems of our paper. We have made some revisions by comprehensively considering the comments from all the anonymous reviewers. A summary of the revisions can be found in another comment block we have posted. We hope that our revision and the following explanation can help you better understand the problem we studied and some details of our method. Our responses to your comments are as follows.
> ***
> **(1) Advantages of the proposed method beyond prior works**
> >Q: The paper does not do a good job of crisply characterizing how their methods differ from prior works. The reasons why their method is better, that the authors state, seem somewhat shallow/arbitrary. They claim that computing gradients etc are expensive/time-consuming. Yet their method is based on adversarial auto-encoders, which need to be trained via gradient-based optimization methods. So it seems like this is not a real difference. The only difference I can see is that this paper considers a different NN architecture and training method compared to some of the other GNN based methods (e.g., Hamilton et al.,). I would like the author's to be a bit more precise and scientific & also credit prior works better.
>
>   Please note that one major contribution of this paper is to use an **inductive** framework across multiple associated graphs to achieve a better trade-off between quality and efficiency over conventional **transductive** GP methods. Namely, **transductive and inductive graph inferences [1,2] are key concepts in this paper**.
>
> Most conventional GP methods are **transductive**, which are only optimized on one single graph (e.g., using gradient descent) and can only tackle GP on such a unique graph. Given a new graph, these **transductive** methods must be optimized from scratch, which may be time-consuming as we highlighted in paragraph 5 of Section 1.
>
> In contrast, as illustrated in Fig. 1, our IGP framework includes (i) the **offline** training on historical known graphs and (ii) **online** generalization to new graphs. The **inductiveness** of IGP implies that we **let model parameters be shared by all the training and test graphs**. *We still need to optimize the dual GNN structure in the **offline** training (e.g., using gradient descent)*, but *there is no additional optimization in the **online** generalization*. Since there is no additional optimization in the **online** generalization and IGP is believed to capture the key characteristics shared by multiple graphs after its **offline** training, **a better trade-off between quality and efficiency can be achieved in the online GP on new graphs**. In summary, the **fast high-quality online generalization** is the major advantage of IGP beyond existing **transductive** methods.
>
> Please also note that **the dual GNN structure in IGP is not a simple application of AAE**. Different from AAE that regularizes the learned embedding with a pre-define probability distribution, there are no probability distributions in IGP as illustrated in Fig. 2. In contrast, we introduce **a novel adversarial process between original graphs and auxiliary label-induced graphs to incorporate the permutation invariant label information (e.g., GP ground-truth) of training data to the offline training**, which is our original design. Some other GP methods based on inductive GNNs [1-5] cannot capture the permutation invariant label information of training data.
>
> During the revision, we comprehensively considered the comments of all the reviewers and added some concrete examples in Section 1 and 2. According to your concerns, we also highlighted the brief discussions regarding the difference between **transductive** and **inductive** graph inference in paragraph 5 of Section 1. We hope the aforementioned explanation and our revision can help you better understand how our method differs from prior works.
>
> [1] Hamilton, William L., Rex Ying, and Jure Leskovec. "Inductive representation learning on large graphs." Proc. of NIPS. 2017.
>
>   [2] Veličković, Petar, et al. "Graph Attention Networks." Proc. of ICLR. 2017.
>
>   [3] Nazi, Azade, et al. "GAP: Generalizable Approximate Graph Partitioning Framework." arXiv preprint arXiv:1903.00614 (2019).
>
>   [4] Nazi, Azade, et al. "A Deep Learning Framework for Graph Partitioning." (2019).
>
>   [5] Wilder, Bryan, et al. "End to End Learning and Optimization on Graphs." Proc. of NIPS 32 (2019): 4672-4683.

---

> ### Author Response · Authors · 2021-11-29
> **Authors' follow-up on comments by Anonymous Reviewer 7xoz**
>
> Dear Reviewer:
>
> Thank you again for your valuable comments! Since the discussion period will end soon, could you please kindly check our responses and revisions? We believe that our responses and revisions have addressed some of your concerns.

---

### Official Review · Reviewer_r3jt · 2021-11-07

**Correctness:** 3
**Technical Novelty And Significance:** 3
**Empirical Novelty And Significance:** 3
**Recommendation:** 6
**Confidence:** 1

**Main Review:**

Strong points:

Overall, the paper is well written and the experimental part seems comprehensive.

A few comments.

1) Theorem 1 is stated in a very informal way. It seems not good to add discussion on one concrete example in Figure 2 in a statement of a theorem. I would suggest that please move that outside and discuss the potential meanings and implications of theorem 1 on examples separately.

2) Figures 1 and 2 both are too small and it is hard to read details there.





**Summary Of The Paper:**

The authors propose an inductive graph partitioning framework across multiple evolving graph snapshots to alleviate the NP-hard challenge. It first conducts the offline training of a dual graph neural network on historical snapshots to capture the structural properties of a system. The trained model is then generalized to newly generated snapshots for fast high-quality online GP without additional optimization, where a better trade-off between quality and efficiency is achieved.

**Summary Of The Review:**

The paper seems well written. Here are a few typos.

1) The first line on the first paragraph of section 2: $S=...$, a comma is missing after the symbol $\mathcal{G}_2$.

2) On page 6, in the sentence just above equation (12), *we drive* should be *we derive*...

---

> ### Author Response · Authors · 2021-11-23
> **Responses to Anonymous Reviewer r3jt**
>
> Thank you for your valuable comments and suggestions. We have made some revisions based on the comments of all the reviewers. A summary of the revisions can be found in another comment block we have posted. We hope that our revision can help you better understand the problem we study and increase your confidence to assess the quality of our paper. Our responses to your comments are as follows.
> ***
> **(1) Organization of Theorem 1**
> >Q: Theorem 1 is stated in a very informal way. It seems not good to add discussion on one concrete example in Figure 2 in a statement of a theorem. I would suggest that please move that outside and discuss the potential meanings and implications of theorem 1 on examples separately.
>
> Thank you for your comments regarding Theorem 1. During our revision, we adopted the suggestion of another reviewer to rename 'Theorem 1' as 'Fact 1'. Moreover, we also adopted your suggestion to separate the statements and examples of 'Fact 1' into 2 different paragraphs.
> ***
> **(2) Small fonts size in some figures**
> >Q: Figures 1 and 2 both are too small and it is hard to read details there.
>
>   During the revision, we redrew Fig. 1, Fig. 2, and Fig. 3, in which we tried to make the font size of most text as large as that in the main paper. We hope our revision can help the readers better read the figures.
> ***
> **(3) Some typos**
> >Q: The first line on the first paragraph of section 2: S={} , a comma is missing after the symbol. On page 6, in the sentence just above equation (12), we drive should be we derive...
>
>   Thank you for pointing out some typos. During the revision, we carefully checked and revised some grammar errors and typos.

---

> ### Author Response · Authors · 2021-11-29
> **Authors' follow-up on comments by Anonymous Reviewer r3jt**
>
> Dear Reviewer:
>
> Thank you again for your valuable comments! Since the discussion period will end soon, could you please kindly check our responses and revisions? We believe that our responses and revisions have addressed some of your concerns.

---

> > ### Comment · Reviewer_r3jt · 2021-11-29
> > **I believe authors well addressed my reviews. Thanks.**
> >
> > I read carefully the response and I believe my concerns and comments are carefully addressed.

---

### Official Review · Reviewer_CKG9 · 2021-11-08

**Correctness:** 3
**Technical Novelty And Significance:** 3
**Empirical Novelty And Significance:** 2
**Recommendation:** 3
**Confidence:** 3

**Main Review:**

The paper proposes an interesting framework which is resistant to permutations of the training set used by the IGP framework.  Specifically, the framework derives graph embeddings based on the input graph (as an adjacency matrix) and feature matrix which encodes neighbor similarity.  The embeddings are then used in the assignment of the node-disjoint cluster labels.

The ablation studies clearly show the effect of different aspects of the proposed architecture.

The reporting of the trade-off scores is slightly unclear.  What is the efficiency measure that is being traded-off?  Clarifying this in the main body of the results will make them clear.  Finally, there is literature - both in terms of discrete algorithms and learning representations - that addressed the problem of incremental partitioning of an evolving graph.  See, for example, https://rlgm.github.io/papers/41.pdf and https://www.vldb.org/pvldb/vol13/p1261-fan.pdf (and references within).  Comparison with the prior art in terms of the trade-off achievable will make the empirical analysis stronger.

Other drawbacks of the work include the availability of training data that *already* computes an 'optimal' partition of a snapshot.  It's not clear how this can be achieved.  Secondly, the comparisons in the experiments are against other ML models whose performance cannot be verified in terms optimality loss, i.e., how far are those solutions from the optimal solution.   In this context, comparison to an algorithm with known approximation factors would help.

**Summary Of The Paper:**

The authors propose an inductive graph partitioning (IGP) framework across multiple snapshots of a dynamic graph to produce an effective partitioning of the graph. IGP is based on training on the snapshots of the graph using a dual GNN architecture and is used to generalize to subsequent snapshots of the graph.

**Summary Of The Review:**

I feel the paper in its current form can be significantly strengthened in terms of the presentation of the empirical results by clearly explaining the trade-off achieved as well comparing with more algorithms from the literature as baselines.

---

> ### Author Response · Authors · 2021-11-22
> **Responses to Anonymous Reviewer CKG9 (Cont-2)**
>
> **(3) ‘Optimal’ partition of training graphs and ‘optimal’ solutions for evaluation**
> >Q: Other drawbacks of the work include the availability of training data that already computes an 'optimal' partition of a snapshot. It's not clear how this can be achieved. Secondly, the comparisons in the experiments are against other ML models whose performance cannot be verified in terms of optimality loss, i.e., how far are those solutions from the optimal solution. In this context, comparison to an algorithm with known approximation factors would help.
>
> Please note that GP can be formulated as some NP-hard combinatorial optimization problems. The NP-hardness implies that there are no polynomial-time algorithms so far that can obtain the optimal solution. In some real applications, **we do not expect to get the optimal partition for graphs even with several hundreds of nodes** since it is extremely time-consuming. In contrast, **we can still evaluate GP quality by comparing the derived objective values of some optimization problems**, e.g., modularity and NCut in our experiments. Moreover, **we can also evaluate quality by measuring the correspondence between GP results and ‘ground-truth’**, e.g., using NMI and AC in our experiments. The ‘ground-truth’ can be from real-world groups or organizations in a application, e.g., cells in wireless cellular networks (WCN). **The aforementioned evaluation protocols (as used in our experiments) are also widely used in some prior work regarding GP [1-5]**. [1-3] have similar settings and evaluation protocols for inductive GP across graphs with our experiments.
>
> Please also note that **we do not assume the training data already computes the ‘optimal’ partitions**, due to the NP-hardness as discussed above. In Section 2, we have discussed two possible sources of the GP results of training data (used as training ‘ground-truth’).
>
> First, the ‘ground-truth’ can be given by some real applications (e.g., decomposing of a WCN) corresponding to real-world groups or organizations in a system (e.g., cells in a WCN). The available ‘ground-truth’ may not correspond to the ‘optimal’ partition but is highly related to real applications.
>
> Second, when the ‘ground-truth’ information from real applications is not available, we can still use GP results of some strong (but usually time-consuming) baselines (e.g., SNMF, node2vec, etc.) as a special source of ‘ground-truth’ to regularize the offline training of IGP (i.e., enhance the learned embedding). In particular, we can select some strong baselines with high GP quality using unsupervised quality metrics of some optimization objectives, e.g., modularity as discussed above.
>
> [5] has a similar problem statement to our method (i.e., inductive GP across graphs), which also tries to capture the ‘ground-truth’ of training data. As we can check in [5], the training ‘ground-truth’ of all the datasets in this paper are not ensured to be optimal. Moreover, different objectives (e.g., modularity maximization and NCut minimization) may have different optimal partitions. Some high-quality GP results for real applications (measured by the correspondence between GP results and application ‘ground-truth’) may also not be the optimal partition of an objective.
>
> For your another concern, we note that there is also analysis regarding the approximation of (i) spectral clustering (SC) for NCut minimization [6] and (ii) degree-corrected stochastic blockmodel (DCSBM) [7] for modularity maximization. SC and DCSBM are already included in our experiments.
>
> In summary, we **adopted the widely-used experiment settings and evaluation protocols [1-5] that do not rely on the ‘optimal’ partition**. There are also **two sources of the ‘ground-truth’ for the offline training of IGP without relying on the ‘optimal’ partition**. We hope the aforementioned explanation can help you better understand the training 'ground-truth' of our method.
>
>   [1] Nazi, Azade, et al. "GAP: Generalizable Approximate Graph Partitioning Framework." arXiv preprint arXiv:1903.00614 (2019).
>
>   [2] Nazi, Azade, et al. "A Deep Learning Framework for Graph Partitioning." (2019).
>
>   [3] Wilder, Bryan, et al. "End to End Learning and Optimization on Graphs." Proc. of NIPS 32 (2019): 4672-4683.
>
>   [4] Dhillon, Inderjit S., Yuqiang Guan, and Brian Kulis. "Weighted Graph Cuts without Eigenvectors a Multilevel Approach." IEEE TPAMI 29.11 (2007): 1944-1957.
>
>   [5] Chen, Zhengdao, Xiang Li, and Joan Bruna. "Supervised Community Detection with Line Graph Neural Networks." Proc. of ICLR (2018).
>
>   [6] Von Luxburg, Ulrike. "A Tutorial on Spectral Clustering." Statistics and Computing 17.4 (2007): 395-416.
>
>   [7] Karrer, Brian, and Mark EJ Newman. "Stochastic Blockmodels and Community Structure in Networks." Physical Review E 83.1 (2011): 016107.

---

> ### Author Response · Authors · 2021-11-23
> **Responses to Anonymous Reviewer CKG9**
>
> Thank you for your valuable comments and suggestions that help us further improve the presentation of our paper. We have made some revisions based on the comments of all the anonymous reviewers. A summary of the revision can be found in another comment block we have posted. Our responses to your comments are as follows. We hope our revision and the following explanation can help you better understand some details of our work.
> ***
> **(1) Presentation of the trade-off between quality and efficiency**
> >Q: The reporting of the trade-off scores is slightly unclear. What is the efficiency measure that is being traded off? Clarifying this in the main body of the results will make them clear.
>
> The efficiency measure in our experiment is the **overall runtime** of a method to derive its final GP result for a graph. Please note that we use **four quality metrics** (i.e., NMI and AC for datasets with ground-truth as well as modularity and NCut for all the datasets) and **only one single efficiency metric** (i.e., runtime). Hence, we consider the **trade-off between each quality metric (from the four possible metrics) and the single efficiency metric (i.e., runtime)** w.r.t. each subfigure in Fig. 3 and 4. During the revision, we adopted your suggestions to **add clear statements regarding the efficiency measure in Section 4** (page 7).
>
> Since our original design of trade-off score (TOS) is based on the normalized area covered by a data point in the visualization results w.r.t. subfigures of Fig. 3, it would be better to give Fig. 3 before introducing TOS, from our perspective. In Fig. 3, we believe that one can qualitatively check the trade-off (between quality and efficiency) of each method by **comparing the relative positions of different data points**. [1] also uses a similar presentation to visualize the trade-off between two aspects. When we wrote the paper, we also tried to give definitions and results of TOS in the main paper but the length of the main paper exceeds the limit of 9 pages. Hence, we had to put details of TOS (e.g., Eq.(15)-(10) and Tables 13-16) in appendix. Despite such difficulty, we still tried our best to revise the paper according to your suggestions, where we **highlighted the basic idea of TOS (e.g., its relationship to Fig. 3) in Section 4.3 while obeying the page limit**.
>
> [1] Chen, Ting, et al. "A simple framework for contrastive learning of visual representations." International conference on machine learning. Proc. of ICML, 2020.
>
> ***
> **(2) Other related baselines**
> >Q: Finally, there is literature - both in terms of discrete algorithms and learning representations - that addressed the problem of incremental partitioning of an evolving graph. See, for example, https://rlgm.github.io/papers/41.pdf and https://www.vldb.org/pvldb/vol13/p1261-fan.pdf (and references within). Comparison with the prior art in terms of the trade-off achievable will make the empirical analysis stronger.
>
> Please note that the first method GAP (from your link https://rlgm.github.io/papers/41.pdf) and some baselines in this paper (e.g., hMetis, GraSAGE, etc.) have been already included in our original submission. You can check results of GAP and related baselines in Section 4 and Appendix C.5.
>
> In this study, we consider inductive GP across multiple graphs that can be (i) associated snapshots evolving over time or (ii) independent (sub)graphs of a scenario without temporal dependency e.g., multiple ego-nets in social media. Namely, **not all the datasets in our experiments have node correspondence among graphs**, e.g., multiple online discussion threads (interaction graphs) of Reddit. **We can only apply incremental GP baselines to some of the datasets with node correspondence** (e.g., Taxi and AS). In contrast, since we do not consider node correspondence among graphs, our IGP methods can be applied to both cases with and without node correspondence, which **is more generic than incremental GP methods**.
>
> During the revision, we tried to find the official open-source code of the incremental GP method you recommend (from your link https://www.vldb.org/pvldb/vol13/p1261-fan.pdf) but we cannot find any open-source implementation of this work. Despite this difficulty, we still tried to include additional incremental GP and graph embedding baselines with official open-source implementations. After the revision, we **have evaluated the GP quality and runtime of IncNSA [1] and TIMERS [2] on the Taxi and AS datasets that have node correspondence with evaluation results and trade-off analysis shown in Appendix C.5** (see Tables 10, 12, 15, and 16). Our IGP method still significantly outperforms these new baselines.
>
>   [1] Su, Xing, et al. "IncNSA: Detecting Communities Incrementally from Time-Evolving Networks Based on Node Similarity." International Journal of Modern Physics C 31.07 (2020): 2050094.
>
>   [2] Zhang, Ziwei, et al. "Timers: Error-bounded SVD Restart on Dynamic Networks." Proc. of AAAI. 2018.

---

> ### Author Response · Authors · 2021-11-29
> **Authors' follow-up on comments by Anonymous Reviewer CKG9**
>
> Dear Reviewer:
>
> Thank you again for your valuable comments! Since the discussion period will end soon, could you please kindly check our responses and revisions? We believe that our responses and revisions have addressed some of your concerns.

---

### Author Response · Authors · 2021-11-23
**Summary of Author Responses and Revisions**

Dear Reviewers:

Thank you for your insightful comments and constructive suggestions that help us achieve a better presentation of our paper. Most of the concerns are about (i) the presentation of figures and tables, (ii) basic assumption of our method, as well as (iii) some detailed settings of our experiments. For each concern, we have provided responses and tried our best to make major revisions of our paper. We hope that you could update your evaluation if we have addressed some of your concerns. For convenience, we summarize our major revisions as follow. All the major revisions have also been highlighted with red text in the new submitted version of our paper.

* We have redrawn some of the figures, where we have highlighted most text in bold and enlarged the font size.

* We have enlarged the font size of all the tables.

* In Section 1 (page 2), we have given some brief discussions regarding the transductiveness of conventional GP methods and their major limitations, which can highlight advantages of our IGP method with an inductive framework across graphs.

* In Section 1 and 2 (page 2 and 3), we have added (i) some brief examples regarding the GP application without node correspondence as well as (ii) reasons why we do not consider node correspondence.

* In Section 1 (page 2), we have replaced the original example of permutation invariant GP labels (e.g., label sequences (0, 1, 1) and (1, 0, 0)) by a more concrete example with node label assignments.

* In Section 3 (page 4), we have renamed ‘Theorem 1’ as ‘Fact 1’ and separated its statements and examples in two paragraphs.

* In Section 4 (page 7), we have added some more brief discussions regarding the setting of $K$ and the usage of Louvain to estimate $K$.

* In Section 4 (page 7), we have highlighted the efficiency measure (i.e., the total runtime) used in our experiment.

* In Section 4 (page 8), we have highlighted the basic idea of the trade-off score and its relationship to the visualization results of Fig. 3.

* In Section 4 (page 9), we have redrawn Fig. 3, where we have removed all the legends, directly denoted data points using text, and selected some of the results to be shown in Fig. 3, with the rest results shown in Appendix C.5.1. (see Fig. 4).

* **[New Experiments]** In our quantitative evaluation, we have included two more incremental GP and embedding baselines (i.e., IncNSA and TIMERS). The brief introduction of these baselines has been added in Appendix C.3 (page 18). New evaluation results of these baselines are reported in Tables 8, 10, 12, 15, and 16.

* **[New Experiments]** In Appendix C.5.3 (page 26), we have added the robustness analysis, where we set $N \in [500, 10000]$, $k \in [10, 20]$, $k_{\rm{min}} \in [100, 200]$, $c_{\rm{min}} \in [10, 20]$, $c_{\rm{max}} \in [100, 200]$, and $\mu \in [0.3, 0.6]$ for the LFR-Net. Evaluation results have been shown in Table 17.

* In Appendix E (page 32), we have added some discussions regarding (i) theoretical bound analysis and (ii) application to attributed graphs.

---

### Decision · Program_Chairs · 2022-01-20

**Decision:**

Reject

**Comment:**

This paper considers an important problem, graph partitioning, from a transductive viewpoint: assuming that the graphs are generated by independent draws from an unknown distribution, learn some parameters in an ``offline” phase, and use these in the ``online” phase (much as in PAC learning). The authors have also answered many of the reviewer questions. In particular, the comparison with existing work is substantial.

While I laud the positives of this work and the importance of the transductive approach, I see an issue: as a reviewer points out and as agreed by the authors, the paper does not provide a theoretical guarantee of the quality of the generalization to unseen graphs. It would have been useful, e.g., to consider this on Erdos-Renyi G(n,p) models, stochastic block models etc.